# Stepsize Anything: A Unified Learning Rate Schedule for Budgeted-Iteration Training

**Anda Tang[1]** **Yiming Dong[1]*** **Yutao Zeng[2]** **Xun Zhou[2]** **Zhouchen Lin[1,3,4]†**

[1]State Key Lab of General AI, School of Intelligence Science and Technology, Peking University
[2]ByteDance Seed
[3]Institute for Artificial Intelligence, Peking University
[4]Pazhou Laboratory (Huangpu), Guangzhou, Guangdong, China
tanganda@pku.edu.cn, yimingdong_ml@outlook.com,
yutao.zeng@outlook.com, zhouxun@bytedance.com, zlin@pku.edu.cn

## Abstract

The expanding computational costs and limited resources underscore the critical need for budgeted-iteration training, which aims to achieve optimal learning within predetermined iteration budgets. While learning rate schedules fundamentally govern the performance of different networks and tasks, particularly in budgeted-iteration scenarios, their design remains largely heuristic, lacking theoretical foundations. In addition, the optimal learning rate schedule requires extensive trial-and-error selection, making the training process inefficient. In this work, we propose the Unified Budget-Aware (UBA) schedule, a theoretically grounded learning rate schedule that consistently outperforms commonly-used schedules among diverse architectures and tasks under different constrained training budgets. First, we bridge the gap by constructing a novel training budget-aware optimization framework, which explicitly accounts for the robustness to landscape curvature variations. From this framework, we derive the UBA schedule, controlled by a single hyper-parameter $\varphi$ that provides a trade-off between flexibility and simplicity, eliminating the need for per-network numerical optimization. Moreover, we establish a theoretical connection between $\varphi$ and the condition number, adding interpretation and justification to our approach. Besides, we prove the convergence for different values of $\varphi$. We offer practical guidelines for its selection via theoretical analysis and empirical results. Extensive experimental results show that UBA *consistently surpasses* the commonly-used schedules across diverse vision and language tasks, spanning network architectures (e.g., ResNet, OLMo) and scales, under different training-iteration budgets.

## 1 Introduction

Deep learning has achieved remarkable success across a wide range of domains, including computer vision and natural language processing. However, despite continual advancements in hardware technologies [45, 53], the training cost of neural networks has increased dramatically due to the growing scale of models and datasets [4, 8, 47, 48]. As a result, resource constraints, including computational power, memory, energy consumption, and time budgets, are emerging as significant bottlenecks in the training process [41, 58]. These challenges highlight the pressing need for budgeted training, which aims to achieve optimal model performance under fixed hardware and limited time.

---

*Work was done during an internship at ByteDance Seed.
†Corresponding author.

39th Conference on Neural Information Processing Systems (NeurIPS 2025).

While existing budgeted training studies broadly address resource efficiency, a critical yet under-explored direction within budgeted training is achieving the best possible model performance under strictly fixed iteration constraints. This scenario is common and practically significant where practitioners work under limited computational or time budgets [30], and in extreme cases, models have to be completed within a few training iterations due to resource exhaustion. To formalize this specific research problem, we introduce the term 'budgeted-iteration training', distinguishing it from the broader scope of budgeted training.

Budgeted-iteration training has received growing attention in research, given its significant real-world applicability. Several studies have developed relevant techniques that align with its goals. Smith et al. [43] propose the cyclical learning rate schedule (CLR), improving accuracy in fewer iterations without tuning [44]. Li et al. [30] introduce budget-aware adaptations for existing learning rate schedules. Chen et al. [5] propose a novel learning rate schedule called Reflected Exponential (REX). These approaches are primarily based on learning rate designing. Learning rate scheduling highlights key advantages: (i) It plays a critical role in general training of diverse neural architectures across tasks. (ii) It has demonstrated suitable and competitive in fixed training iteration budgets [20]. (iii) It is plug-and-play, requiring minimal adjustments to the underlying model architecture, which makes them easily adaptable to various deep learning frameworks. Leveraging these advantages, we adopt the learning rate design approach for budgeted-iteration training.

Despite their advantages, most learning rate schedules, whether tailored for budgeted-iteration training or standard training performance, are still heuristic and lack rigorous theoretical grounding. In addition, existing schedules typically rely on manually designed rules or empirical tuning. Consequently, selecting an optimal schedule often involves extensive trial-and-error, incurring substantial cost in terms of time and computation [34].

*A natural question arises: Does there exist a theoretically grounded, unified schedule that eliminates heuristic selection while maintaining robust performance across tasks, networks, scales and training budgets?*

In this paper, we provide an affirmative answer by proposing a theoretically grounded schedule. The proposed learning rate schedule should consistently outperforms existing schedules among diverse architectures and tasks under different constrained training budgets. By doing so, it avoids choosing suitable learning rate schedule after multiple trials for network training.

To achieve this, we first bridge the gap by constructing a unified budget-aware training optimization framework, which incorporates the robustness to landscape curvature variations induced by data distribution, sampling, network architectures and optimization. The framework employs a min-max optimization approach to guarantee performance under worst-case conditions, since the minimization of learning rate is operated on an upper bound. Then we obtain numerical solutions by gradient projection methods. To eliminate the need for repeated numerical optimization when applying our method to different networks, we propose a universal parametric function that approximates numerical solutions. We nominate the resulting schedule Unified Budget-Aware (UBA) schedule. It requires tuning only a single hyper-parameter $\varphi$, reducing the overhead of per-network numerical optimization. Moreover, we establish a theoretical connection between $\varphi$ and the condition number, adding interpretation and optimization difficulty-aware theoretical grounding. Besides, we prove the convergence for different values of $\varphi$. These theoretical analysis along with empirical results offer practical guidelines for $\varphi$ selection. We evaluate UBA through comprehensive experiments across vision and language tasks, spanning diverse architectures and iteration budgets. Specifically, for vision tasks, the UBA schedule demonstrates *consistent superiority* over baselines across all evaluated datasets and model scales under varying training iterations. For language tasks, we validate the effectiveness of UBA through extensive benchmarks with OLMo model(36M, 73M, 150M and 300M parameters). Results show that UBA achieves state-of-the-art performance across on approximately half of the benchmarks, and consistently outperforms baselines on the average scores. Moreover, our performance improvements come with negligible computational overhead, which constitutes one of its key advantages.

**Main contributions**:

1. We construct a unified budget-aware training optimization framework that inherently adapts to landscape curvature variations, enhancing training robustness of leaning rates.

2. We propose the Unified Budget-Aware (UBA) schedule from our constructed optimization problem, controlled by a single hyper-parameter $\varphi$ that provides a trade-off between flexibility and simplicity, i.e. adaptive curvature adjustment and minimal tuning cost.

3. We prove the convergence under $\varphi$ and derive practical guidelines for its selection through analysis and experiments. Besides, theoretical analysis and empirical results show that $\varphi$ is related to optimization difficulty such as condition number.

4. We perform experiments and demonstrate that UBA surpasses the commonly-used schedules across diverse vision and language tasks, spanning network architectures (e.g., ResNet, OLMo) and scales, under different training-iteration budgets. Moreover, our performance improvements come with negligible computational overhead, which constitutes one of its key advantages.

## 2 Related work

**Budgeted training**  Researchers face significant challenges in achieving optimal model performance under fixed hardware and limited time. To address these challenges, the concept of budgeted training has gained increasing attention, exploring techniques including computation efficiency, model compression, training stability and convergence improvement [41]. It focuses on: (i) emphasizing the allocation of resources, such as the balance between the model size and the amount of data [2, 6, 24, 28]. (ii) finding optimal configurations or improving performance within the given compute or time budget, such as memory efficiency and computation reduction [17, 26, 31, 38], optimization learning rate schedules [5, 30, 43, 44], batch size [17] and other weight averaging method [26, 27].

Within the context of budgeted training, a key area that remains under-explored is achieving the best possible model performance within a fixed number of iterations, i.e. 'budgeted-iteration training'. In this domain, learning rate scheduling based methods are particularly aligned with the objectives of budgeted-iteration training. Smith et al. [43] propose a new learning rate schedule, named cyclical learning rates (CLR). It improves accuracy in fewer iterations without tuning and is relevant to super-convergence phenomenon [44]. Li et al. [30] introduce an alternative setting of existing learning rate schedules for budgeted training. Chen et al. [5] propose the reflected exponential schedule (REX) via a profile and sampling fashion. Zhang et al. [57] introduce the finite horizon optimization framework and apply it to linear programming, which optimizes the algorithm performance under a strict iteration budget. Learning rate based approaches achieve robust and high-performing results under various lengths of training iterations, which corresponds to our purpose in this work, i.e. budgeted-iteration training [34]. Besides, this approach is plug-and-play, requiring no substantial alterations to the underlying model structure, making it readily adaptable to various deep network frameworks. Therefore, we explore the proper learning rate schedule to achieve budgeted-iteration training.

**Learning rate schedule**  The learning rate plays a pivotal role in controlling the optimization process during network training. The common scheme is the step decay schedule. A typical instance decreases the learning rate by a decaying scalar 0.1 after $50\%$ epochs and by a decaying scalar 0.01 after $75\%$ epochs [21]. Then Loshchilov et al. [32] observe that sharp decreases may prevent models from escaping local minima and propose the cosine schedule function, which is the most popular schedule for language model pretraining.

Pan et al. [36] propose Eigencurve, and demonstrate that minimax-optimal convergence can be achieved for quadratic objectives when the Hessian's eigenvalue distribution exhibits substantial skewness. Defazio et al. [11] developed the theoretically-grounded framework for learning rate scheduling, establishing optimal linear decay schedules and refinements through rigorous mathematical derivation. Their work additionally provides the most extensive empirical evaluation of scheduling approaches to date. Although some schedules include the CLR [43], REX [5], Warmup-Stable-Decay (WSD) [25] and schedule from multi-power law [34], there is no consensus on the optimal choice. In addition, the detail can be found in Appendix F, including some works focus on adaptive learning rate methods.

Learning rate design is not only significant in general training, but also critical in budgeted-iteration training which still remains a topic of debate. Some analyses advocate for small, constant learning rates to ensure stability and convergence [12]. On the contrast, one prevailing hypothesis suggests that

large learning rates may facilitate crossing over sharp local minima in the optimization landscape [56]. Despite the lack of comprehensive theoretical explanations, a range of learning rate schedules inspired by the above analyses as heuristic guidelines has been widely adopted in practice, using variable learning rates to budgeted-iteration training [5, 30]. In this work, we explore learning rate schedule from optimization problem tailored to budgeted-iteration training, aiming to balance iteration budget constraints and generalization.

# 3 Budgeted-iteration training

## 3.1 Finite optimization under limited training iterations

To design a one-size-fits-all learning rate schedule, we construct a robust optimization model of learning rates across varying training conditions (see more conceptual illustration in Appendix A). Specifically, we aim to guarantee minimal loss within constrained training iterations under the worst-case conditions, proposing a budget-iteration-aware framework for learning rate optimization.

**Definition (Finite optimization)**: Let $f(W, D)$ denote the function parameterized by a given neural network with parameters $W \in \mathbb{R}^N$ on the dataset $D$. Let $\xi$ denote the data sampling on the dataset $D$, and let $\mathcal{F}$ be a function class. Let $\mathcal{L}$ be the loss function. Let $\eta_t$ be the learning rate at the $t$-th iteration, $T$ be a maximum number of training iterations, and $t$ be the current learning step. A finite optimization is

$$
\begin{aligned}
\min_{\eta_1, \eta_2, \cdots, \eta_{T-1}} \max_{f \in \mathcal{F}} \quad & \mathcal{L}(f(W_T, \xi)), \\
s.t. \quad & W_{t+1} = W_t - \eta_t \nabla \mathcal{L}(f(W_t, \xi)), \\
& t = 0, 1, 2, \cdots, T-1.
\end{aligned} \tag{1}
$$

In the optimization model 1, the constraint represents the stochastic gradient descent process. The maximizing of $\mathcal{L}(f(W_T, \xi))$ represents the worst-case among the training process on the network $f$. Then it minimizes the worst-case loss within given iterations, embodying its budget-aware property. By formulating the problem as a min-max optimization, we identify learning rates $\eta_t (t = 1, 2, \cdots, T-1)$ that are resilient to the uncertainties introduced by different training configuration, uniformly throughout the optimization trajectory.

The challenge lies in characterizing the $f$ within the optimization process. $f$ is primarily determined by variations in parameter configuration, datasets characteristics, batch ordering and network architecture. From an optimization standpoint, we assume that the characteristics of $f$ shaped by these factors can be captured by the loss landscape of $f$. Therefore, we can analyze $f$ by approximating its loss using a quadratic expansion around nearby strict local optima. Specifically, during the optimization process, the loss surface in the vicinity of the optimization trajectory can be approximated by sequence of strict local optima (or at least by one optimum), denoted as $\bar{W}^{(k)} \quad (k = 1, 2, \cdots, K, \quad K \in \mathbb{Z}^+)$, with the final optimum represented as $\bar{W}^K$. This approach enables us to capture the key features of the loss surface near these points. Therefore, the trajectory of optimization is impacted by the characteristics of the nearby strict local optima, since the optimization process is inherently shaped by the loss landscape and the key features of the surface are captured by these optima. Consequently, we can derive the learning rate within the optimization model by the information of these nearby strict local optima.

According to the second-order necessary condition for the strict minimum $\bar{W}^{(k)}$, this approximation is achieved through the positive semi-definite Hessian matrix $H_f^{(k)}(\xi) \in \mathbb{R}^{N \times N}$. In addition, the optimization problem (1) will be programmed sequentially. Then objective function of the optimization problem (1) can be reformulated as $\mathcal{L}(f(\bar{W}^{(k)}, \xi)) + \frac{1}{2} (W_{T_{k+1}} - \bar{W}^{(k)})^\top H_f^{(k)}(\xi) (W_{T_{k+1}} - \bar{W}^{(k)}) \quad (k = 1, 2, \cdots, K)$. Given that the networks under consideration possess sufficient capacity to fit the data, the loss at the optimal point for different $f$ can be made small enough. Thus the term $\mathcal{L}(f(\bar{W}^{(k)}, \xi))$ can be reasonably neglected in our analysis. We obtain the following sequential optimization problem.

$$\min_{\eta_{1+T_k},\eta_{2+T_k},...,\eta_{T_{k+1}}} \max_{f(\xi)} \quad \frac{1}{2}\left(W_{T_{k+1}} - \bar{W}^{(k)}\right)^\top H_f^{(k)}(\xi)\left(W_{T_{k+1}} - \bar{W}^{(k)}\right),$$

$$s.t. \quad W_{t+1} = W_t - \eta_t H_f^{(k)}(\xi)\left(W_t - \bar{W}^{(k)}\right), \tag{2}$$

$$t = T_k + 1, T_k + 2, ..., T_{k+1} \quad (k = 1, 2, \cdots, K-1),$$

where $T_1 = 0$ and $T_K = T - 1$.

By the first constraint $W_{t+1} = W_t - \eta_t H_f^{(k)}(\xi)\left(W_t - \bar{W}^{(k)}\right)$, we can obtain (see Appendix A for derivation)

$$\left\|W_{T_{k+1}} - \bar{W}^{(k)}\right\|_2^2 \leq \max_{\lambda_l^{(k)} \leq \lambda_i^{(k)} \leq \lambda_u^{(k)}} \left[\prod_{t=1+T_k}^{T_{k+1}}\left(1 - \eta_t \lambda_i^{(k)}\right)\right]^2 \left\|W_{T_k} - \bar{W}^{(k)}\right\|_2^2, \tag{3}$$

where $\lambda_i^{(k)}$ $(i = 1, 2, \cdots, N)$ are the eigenvalues of $H_f^{(k)}(\xi)$ around the $k$-th optimum, which satisfy $0 < \lambda_l^{(k)} \leq \lambda_i^{(k)}(f(\xi)) \leq \lambda_u^{(k)}$ for all $i$, $u_i^{(k)}$ $(i = 1, 2, \cdots, N)$ denote $N$ linearly independent eigenvectors and $s_i^{(k)}$ are the coefficients corresponding to the eigenvector components. Since the term $\left\|W_{T_k} - \bar{W}^{(k)}\right\|_2^2$ is a certain constant, the optimization process of weights $W_{T_{k+1}}$ is equivalent to the least upper bound of the $\prod_{t=1+T_k}^{T_{k+1}}\left(1 - \eta_t \lambda_i^{(k)}\right)$. Thus, the optimization of learning rate schedule can be formulated as follow,

$$\min_{\eta_{1+T_k},\eta_{2+T_k},...,\eta_{T_{k+1}}} \max_{\lambda_l^{(k)} \leq \lambda_i^{(k)} \leq \lambda_u^{(k)}} \prod_{t=1+T_k}^{T_{k+1}}\left(1 - \eta_t \lambda_i^{(k)}\right)^2,$$

$$s.t. \quad \eta_{1+T_k}, \eta_{2+T_k}, ..., \eta_{T_{k+1}} \in [\eta_{\min}, \eta_{\max}], \tag{4}$$

$$k = 1, 2, \cdots, K,$$

To solve the constrained min-max problem (4), we adopt an iterative projected gradient method that alternates between minimizing over the variables $\eta_t$ and maximizing over the parameters $\lambda_i^{(k)}$. To avoid repeated numerical optimization, we fit the solutions with a parametric function. Details regarding the numerical solution and curve fitting process are provided in Appendix B. Then, we obtain the $\eta_t$ within the interval $t \in [1 + T_k, T_{k+1}]$ as follows

$$\eta_t = (\eta_{\max} - \eta_{\min}) \frac{2\left[1 + \cos\left(\frac{[2(t-T_k)-1]\pi}{2(T_{k+1}-T_k)} + (k-1)\pi\right)\right]}{2\varphi + (2-\varphi)\left[1 + \cos\left(\frac{[2(t-T_k)-1]\pi}{2(T_{k+1}-T_k)} + (k-1)\pi\right)\right]} + \eta_{\min}, \tag{5}$$

where $\varphi$ is the hyper-parameter controlling the variation speed of the learning rate $\eta_t$.

When setting $(K > 2)$, UBA extends to a multi-phase formulation, which offers three potential advantages. (i) Hierarchical optimization: by partitioning the budget-aware optimization into multiple phases, each near a different local minima, it improves approximation accuracy and captures the dynamic features of the loss surface. (ii) It helps escape saddle points, (iii) It generalizes the single-phase method, allowing for future extensions. Notably, a single-phase approach $(K = 2)$ remains effective, as the robust budget-aware model inherently derives the optimal schedule function, as shown in Figure 1(a). For consistency with common practice and to better isolate scheduling effects, this paper focuses primarily on the single-phase implementation. The multi-phase approach is also compared in Appendix E.9, with its schedule shown in Figure 1(b).

We name the proposed schedule Unified Budget-Aware (UBA) schedule for the following reasons. The robust budget-aware model minimizes the loss function **uniformly** across the optimization trajectory, resulting in stable performance. UBA provides a reliable, **unified** choice for practitioners, eliminating the need for case-by-case baseline comparisons and delivering consistent superiority across datasets, architectures, and training budgets. Lastly, UBA can **uniformly** approximate the behavior of existing schedules through simple parameter adjustments.

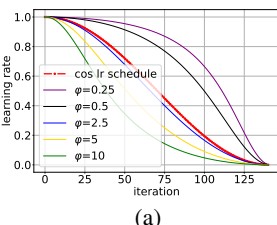 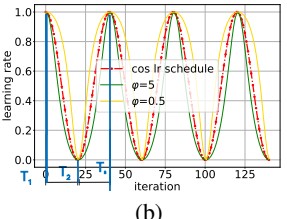

(a)                 (b)

Figure 1: Evolution of the learning rate in UBA schedule across training iterations.

## 3.2 Theoretical analysis

**Proposition 1.** *The fit function* (5) *is the exact closed-form solution to the min-max optimization problem:*

$$\min_{\eta_{1+T_k},\eta_{2+T_k},\cdots,\eta_{T_{k+1}}} \max_{\lambda_l^{(k)} \leq \lambda_i^{(k)} \leq \lambda_u^{(k)}} \prod_{t=1+T_k}^{T_{k+1}} \left[1 - \left(\left(\frac{1}{\lambda_l^{(k)}} - \frac{1}{\lambda_u^{(k)}}\right)\eta_t + \frac{1}{\lambda_u^{(k)}}\right)\lambda_i^{(k)}\right]^2, \quad (6)$$

*when the hyper-parameter $\varphi$ are determined by $\lambda_l^{(k)}$ and $\lambda_u^{(k)}$ through the relation $\varphi = 2\frac{\lambda_u^{(k)}}{\lambda_l^{(k)}}$ and $\eta_{\max} = 1, \eta_{\min} = 0$.*

The min-max model in Proposition 1 represents a special case of our generalized optimization framework. We show that UBA is the exact solution to this special case optimization problem when the learning rate is scaled by $\left(\frac{1}{\lambda_l^{(k)}} - \frac{1}{\lambda_u^{(k)}}\right)\eta_t + \frac{1}{\lambda_u^{(k)}}$. It provides a theoretical foundation for our choice of learning rate instead of choosing it heuristically or empirically, adding rigor and justification to our approach. Moreover, in this case, $\varphi$ is linked condition number. It indicates how the learning rate is shaped by the local curvature of the model, which could guide the optimization process more effectively. The relationship between $\varphi$ and condition number suggests an adaptive nature for the learning rate. In regions with sharp curvatures (large condition number), $\varphi$ reduces the learning rate more rapidly to avoid overshooting. Conversely, in flatter regions, $\varphi$ allows the learning rate to remain large for several iterations, facilitating faster convergence. The transformation of the learning rate trend is shown in Figure 1. *This is a step toward establishing a principled connection between learning rate and local loss landscape geometry along with optimization difficulty.* By this special case, we generalize that $\varphi$ is related to optimization difficulty. The empirical results support these conclusions, with further details in Section 4.3 and Appendix E.6.

**Proposition 2.** *Consider the training process within the interval $t \in [1 + T_k, T_{k+1}]$. When the hyper-parameter $\varphi$ is set sufficiently close to 2, the proposed learning rate scheduling formula reduces to the cosine learning rate schedule.*

By Proposition 1, the hyper-parameter $\varphi$ is related to the optimization difficulty. In that case, the optimization difficulty is explicitly quantified as the condition number and $\varphi = 2\kappa$, where $\kappa = \frac{\lambda_u^{(k)}}{\lambda_l^{(k)}}$. Furthermore, Proposition 2 shows that when $\varphi$ approaches 2, the proposed learning rate schedule converges to the standard cosine schedule. It means that, in regions where the optimization difficulty is manageable (i.e., the curvature is relatively flat), setting $\varphi \approx 2$ allows the learning rate schedule to naturally reduce to the cosine form. Besides, our schedule can approximate the behavior of existing schedules (e.g.step decay, cosine annealing, cyclic schedule or Rex schedule) through simple parameter adjustments, shown in Table 1. The detail can be found in Appendix C. More importantly, it outperforms these schedules by training convergence and final accuracy, supporting results can be found in experiment Sections 4.1 and 4.2 and Appendix E.5.

**Theorem 1.** *Let $n_t = H_f^{(k)}\left(W_t - \bar{W}^{(k)}\right) - H_f^{(k)}(\xi)\left(W_t - \bar{W}^{(k)}\right)$ be the stochastic curvature noise introduced by sampling at iteration $t$. Assume that the sampling Hessian satisfies: $\mathbb{E}_\xi\left[n_t n_t^\top\right] \preceq \sigma^2 H_f^{(k)}$ for some constant $\sigma$. Denote $\tau := \frac{4\lambda_l^{(k)}(\eta_{\max}-\eta_{\min})\left[1+\cos\left(\frac{[2(t-T_k)-1]\pi}{2(T_{k+1}-T_k)}\right)\right](T_{k+1}-T_k)}{(\varphi-2)\pi}$. If we*

Table 1: The adaptive simulation of existing schedules.

| Schedule | Parameter adjustments | Schedule | Parameter adjustments |
|---|---|---|---|
| Cosine | $\varphi = 2$ | Step | $\varphi = 0, \eta_{\max} = 0.5^k, T_{k+1} - T_k =$decaying step,$k = 1, 2, ...$ |
| Exponential | $\varphi = 30$ | Cyclic | $\varphi = 2, k \leftarrow k+1, T_{k+1} - T_k =$cyclic step ,$k = 1, 2, ...$ |
| Rex | $\varphi = 0.8$ | OneCycle | $\varphi = 2, k \leftarrow k+1, T_2 - T_1 =$ pct_start step |

*set the learning rate as the proposed form* (5)*, the loss uncertainty introduced by stochastic gradient method within the interval* $t \in [1 + T_k, T_{k+1}]$ *can be bounded by two terms,*

**For $\varphi > 2$:**

$$\mathbb{E}\left[\mathcal{L}(f(W_t, \xi)) - \mathcal{L}(f(\bar{W}^{(k)}, \xi))\right]$$
$$\leq \left[\frac{4(T_{k+1} - T_k) + (\varphi - 2)\pi}{4(T_{k+1} - T_k) + (\varphi - 2)\pi(t - T_k)}\right]^\tau \cdot \exp\left(-2\lambda_i^{(k)}\eta_{\min}(t - T_k)\right)\lambda_u^{(k)}\|W_{1+T_k} - \bar{W}^{(k)}\|_2^2$$
$$+\sigma^2 \sum_{i=1+T_k}^{t} \eta_i^2 \sum_{j=1}^{N} (\lambda_j^{(k)})^2 \exp\left(-2\lambda_j^{(k)}\eta_{\min}(t - i)\right) \cdot \left[\frac{4(T_{k+1} - T_k) + (\varphi - 2)\pi(i - T_k)}{4(T_{k+1} - T_k) + (\varphi - 2)\pi(t - T_k)}\right]^\tau . \quad (7)$$

**For $\varphi < 2$:**

$$\mathbb{E}\left[\mathcal{L}(f(W_t, \xi)) - \mathcal{L}(f(\bar{W}^{(k)}, \xi))\right]$$
$$\leq \left[\frac{(2\varphi + 2\pi - \varphi\pi) - \frac{(2-\varphi)\pi}{(T_{k+1}-T_k)}(t - T_k - 0.5)}{(2\varphi + 2\pi - \varphi\pi) + \frac{(2-\varphi)\pi}{2(T_{k+1}-T_k)}}\right]^{-\tau} \cdot \exp\left(-2\lambda_i^{(k)}\eta_{\min}(t - T_k)\right)\lambda_u^{(k)}\|W_{1+T_k} - \bar{W}^{(k)}\|_2^2$$
$$+\sigma^2 \sum_{i=1+T_k}^{t} \eta_i^2 \sum_{j=1}^{N} \left(\lambda_j^{(k)}\right)^2 \exp\left(-2\lambda_j^{(k)}\eta_{\min}(t - i)\right) \cdot \left[\frac{(2\varphi + 2\pi - \varphi\pi) - \frac{(2-\varphi)\pi}{(T_{k+1}-T_k)}(t - T_k - 0.5)}{(2\varphi + 2\pi - \varphi\pi) - \frac{(2-\varphi)\pi}{(T_{k+1}-T_k)}(i - T_k - 0.5)}\right]^{-\tau} . \quad (8)$$

## 4 Experiment results

We conduct comprehensive evaluations along three dimensions: (i) Modality diversity, including vision and language tasks; (ii) Training budget, including model scales (small to large) and training iterations (short to long); (iii) Ablation studies, such as parameter sensitivity and cross-optimizer performance. This evaluation strategy ensures the robustness and generalization of our findings.

Our implementation strictly follows the PyTorch official API, i.e. it is inherited from the base class `torch.optim.lr_scheduler.LRScheduler` (Appendix E.3). It maintains full compatibility with all `torch.optim` optimizers. The complete production-ready code is publicly available (see Appendix E.3 for link).

A key advantage of UBA lies in its critical parameter $\varphi$. While an optimal $\varphi$ can further improve model performance, we intentionally fix its value across all tasks and architectures to ensure fair evaluation. We fix $\varphi$ for SGD and AdamW respectively (see learning rate variations in Figure 1(a)). The rationale for these choices is systematically analyzed in our ablation study 4.3. Since each schedule has a different optimal learning rate in practice, to thoroughly address this concern, we conduct additional experiments to evaluate the performance of all schedules with different learning rates in Appendix E.8.

**Baselines** Research on budgeted-iteration training remains limited, with most existing approaches focusing on learning rate scheduling strategies [5, 30, 44]. To provide a fair comparison, we adopt several widely used and empirically effective learning rate schedules as baselines: **Step(SS)** [16], **Cosine(CS)** [32], **Cyclical (CLR) and OneCycle (1C)** [43, 44], **Budgeted training(BT, mathematically equivalent to linear decay)** [30] and **Reflected Exponential (REX)** [5]. Details of these baselines are provided in Appendix E.1.

Table 2: Validation accuracy for vision classification tasks. We present validation accuracy on vision benchmarks (e.g., CIFAR10/100 and ImageNet) using different architectures (ResNet18, ResNet34, ResNet50) under fixed epoch budgets of 25%, 50%, and 100% of the maximum training epochs.

| Dataset | Network | Method | training budget (epoch(%)) | | |
|---|---|---|---|---|---|
| | | | 75 (25%) | 150(50%) | 300(100%) |
| CIFAR10 | ResNet18 | SS | 92.71±0.4243 | 93.45±0.4822 | 93.61±0.3035 |
| | | CS | 94.21±0.5116 | 94.24±0.2419 | 95.52±0.2433 |
| | | CLR | 92.13±0.2610 | 92.99±0.5459 | 94.92±0.3811 |
| | | 1C | 94.23±0.1838 | 95.03±0.1353 | 95.35±0.2984 |
| | | BT | 94.18±0.3546 | 94.79±0.4605 | **95.68±0.1168** |
| | | REX | 94.49±0.3225 | 95.10±0.1929 | 95.45±0.1350 |
| | | UBA(ours) | **94.57±0.3281** | **95.26±0.2150** | 95.65±0.1589 |
| | | | 75 (25%) | 150(50%) | 300(100%) |
| CIFAR100 | ResNet34 | SS | 71.09±0.2676 | 75.34±0.1802 | 77.24±0.1808 |
| | | CS | 64.12±0.1808 | 69.84±0.7580 | 74.29±0.5958 |
| | | CLR | 73.33±0.5635 | 73.41±0.2835 | 74.74±0.2266 |
| | | 1C | 73.38±0.3227 | 75.04±0.6382 | 77.86±0.2239 |
| | | BT | 72.75±0.2611 | 75.35±0.2423 | 78.58±0.2705 |
| | | REX | 73.64±0.5635 | 75.14±0.3110 | 78.04±0.1430 |
| | | UBA(ours) | **74.60±0.1508** | **76.63±0.2678** | **78.95±0.2818** |
| | | | 75 (25%) | 150(50%) | 300(100%) |
| ImageNet | ResNet50 | SS | 74.74±0.1386 | 77.24±0.3861 | 78.56±0.6435 |
| | | CS | 75.84±0.2220 | 77.95±0.2164 | 79.07±0.0438 |
| | | CLR | 73.90±1.4609 | 76.04±1.6419 | 77.84±1.3053 |
| | | 1C | 75.83±0.7792 | 77.74±0.5218 | 78.90±0.2956 |
| | | BT | 75.90±0.2786 | 77.81±0.4624 | 78.86±0.2814 |
| | | REX | 75.85±0.8047 | 77.66±0.8853 | 78.66 ±0.2828 |
| | | UBA(ours) | **76.29±0.4031** | **78.26±0.1640** | **79.39±0.0170** |

## 4.1 Experiments for vision classification tasks

We evaluate our proposed UBA schedule on vision benchmarks (e.g., CIFAR10/100 and ImageNet) using different architectures (ResNet18, ResNet34, ResNet50). In addition, we independently train models using fixed epoch budgets of 25%, 50%, and 100% of the maximum training epochs, without reusing or interpolating results from longer training runs. This setup ensures that each budget setting is evaluated in isolation, thereby preserving the integrity of comparisons across low and high training budgets. Table 2 presents the validation accuracy across different training budgets, comparing UBA against six baseline schedules (see detailed results in Appendix E.4).

We find that: (i) UBA demonstrates the strongest performance across all training budgets, achieving superior results on both small-scale (CIFAR10/100 with ResNet18/34) and large-scale (ImageNet with ResNet50) benchmarks. This consistent improvement highlights the *generalizability across model and dataset scales*. (ii) UBA outperforms baselines not only at 100% training budget but also at 25% and 50% iteration budgets, demonstrating its *budget efficiency*, i.e. effectiveness in computation-constrained scenarios. (iii)Notably, UBA shows robust performance even while the second-best schedule varies depending on both the datasets, architectures and training budgets. For practitioners seeking reliable schedules without extensive method selection, UBA provides a default-strong choice. UBA eliminates the need for case-by-case baseline comparison and delivers *stable superiority and reliability* regardless of datasets, architectures, training budgets and scales.

## 4.2 Experiments for language models

We evaluate UBA schedule on the OLMo[19], a truly open language model based on decoder-only transformer architecture, across diverse benchmarks in language model evaluation. Since large language models commonly provide multiple scales to accommodate different compute constraints. We adjust both model size and training steps to explore budgets. We evaluate UBA on OLMo networks

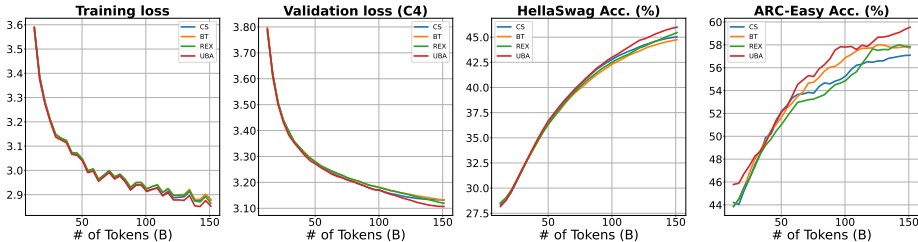

Figure 2: Training dynamics and performance for language tasks under 150B tokens on 300M OLMo. We present the training loss, validation loss, and downstream performance on HSWAG and ARC-E, demonstrating that UBA schedule achieves superior performance.

Table 3: Performance comparison between UBA and the best-performing baseline schedules on OLMo. We report the results of top-performing baseline schedule for the corresponding benchmark and model scale. The names of the top-performing baseline schedules are listed below the results. Bold values indicate UBA's superiority (See overall results of each schedule in Appendix E.5).

| Size | Sched. | Benchmark(accuracy %) | | | | | | | | | | |
| | | PIQA | HSWAG | OBQA | SciQ | ARC-E | ARC-C | COPA | SIQA | SOC | OTH | Avg. |
|---|---|---|---|---|---|---|---|---|---|---|---|---|
| 36M | Best baseline | 61.15 (CLR) | 27.91 (1C) | 27.80 (REX) | 68.10 (REX) | 45.26 (SS) | 23.41 (1C) | 63.00 (SS) | 41.45 (CLR) | 25.18 (REX) | 29.86 (1C) | 40.92 (REX) |
| | UBA | 60.39 | **27.98** | 27.40 | **68.20** | **45.79** | 21.74 | **63.00** | 40.69 | 24.33 | **30.14** | **40.97** |
| 73M | Best baseline | 62.46 (REX) | 30.22 (REX) | 28.80 (CS) | 72.60 (CS) | 47.54 (1C) | 26.42 (1C) | 66.00 (BT) | 41.97 (BT) | 26.32 (SS) | 29.34 (REX) | 42.53 (CS) |
| | UBA | **63.17** | 30.09 | **28.80** | **74.30** | 45.79 | 22.74 | 65.00 | 41.45 | **27.13** | 28.85 | **42.73** |
| 150M | Best baseline | 66.00 (CS) | 35.72 (REX) | 32.80 (1C) | 78.20 (BT) | 53.16 (1C) | 26.42 (BT) | 69.00 (REX) | 43.71 (CLR) | 27.68 (REX) | 32.64 (REX) | 45.91 (REX) |
| | UBA | 65.23 | 35.50 | 29.80 | **78.30** | 50.35 | **27.42** | 67.00 | 43.50 | **29.29** | **32.72** | 45.91 |
| 300M | Best baseline | 70.18 (REX) | 46.30 (REX) | 33.40 (CS) | 84.40 (REX) | 57.72 (REX) | 28.43 (REX) | 72.00 (BT) | 44.58 (CS) | 29.50 (REX) | 36.74 (REX) | 49.70 (REX) |
| | UBA | 69.48 | **46.44** | **34.60** | 83.90 | **60.35** | **29.10** | **72.00** | 44.42 | 28.53 | 35.41 | **50.42** |

spanning four parameter scales: 36M, 73M, 151M, and 300M, covering normal to large-scale models. Due to space limitation, we defer details such as benchmarks introduction, experimental setting and overall results in Appendix E.5).

From Table 3, UBA achieves state-of-the-art performance on approximately 50% of the benchmarks across all scales while the second-best schedule varies. Furthermore, it achieves *consistent superior average performance* among all baselines. It highlights the stable superiority and reliability of UBA, providing a *default-strong choice*. Moreover, it demonstrates significant improvements in SciQ-73M(+1.7) and ARC-E-300M(+2.63), highlighting its ability to enhance generalization across diverse benchmarks. Besides, in Figure 2, UBA consistently achieves lower training loss and validation loss throughout the training, indicating the *efficient training ability* and downstream performance enhancement. Notably, while task-specific tuning of $\varphi$ can further improve model performance, we intentionally fix its value across all tasks and architectures to ensure fair evaluation. Remarkably, even with this universal $\varphi$ setting, UBA consistently outperforms baselines on diverse benchmarks, demonstrating *inherent robustness* to varying benchmarks and model scales.

### 4.3 Ablation study

**Performance across different optimizers**   UBA originates from the optimization problem under gradient descent dynamics. While modern optimizers (e.g., AdamW [33]) introduce momentum and adaptive mechanisms, they maintain the fundamental property of gradient-based updates. To verify the cross-optimizer performance, we conduct ablation studies between SGD and AdamW

optimizers(see detailed results in Appendix E.6). The results show that UBA achieves SOTAs on both SGD and AdamW optimizers, demonstrating the *cross-optimizer robustness* of UBA. This highlights its *broad applicability* beyond standard gradient descent.

**Parameter analysis of** $\varphi$    We perform a sensitivity analysis of $\varphi \in \{0.25, 0.5, 1.0, 2.5, 5, 10\}$ in equation (5) , which controls the variation speed of learning rate (see detailed setting, results and analyses in Appendix E.7). Figure 11 shows that AdamW prefers smaller $\varphi$ while SGD requires larger $\varphi$. We attribute this phenomenon to the preconditioning effect of AdamW. As the relationship formalized in Proposition 1), smaller $\varphi$ is favored for low optimization difficult, while larger $\varphi$ is beneficial for difficult optimization. AdamW adapts the learning rate by scaling gradients with the second moment estimate $\sqrt{v_t}$, implicitly reducing the condition number of the optimization landscape. Thus the schedule with smaller $\varphi$ is preferred. In contrast, SGD lacks such adaptive mechanisms, thus a larger $\varphi$ is more suitable for this situation. This alignment between theory and experiment underscores the importance of tailoring $\varphi$ to the optimizer's characteristics. *Overall, $\varphi$ is related to a generalized optimization difficulty, where optimization precondition effect, datasets distribution and network architecture are all related to optimization difficulty.*

**Performance across different periods**    Our schedule has a periodic phase-based learning rate adjustment setting, where the learning rate at the $k$-th phase is dynamically determined by the $k$-th local minimum of the loss landscape. To validate our scheduling strategy, we conduct experiments by varying $K$ (see detailed results in Appendix E.9). The results suggests that multi-phase scheduling captures the dynamic features of the loss surface more finely, but it needs careful selection for $\varphi$, where $\varphi$ reflects optimization difficulty. However, selecting optimal $\varphi$ values per phase for multi-phase remains non-trivial, which motivates future work on automated landscape-aware $\varphi$ tuning.

## 5    Conclusion

In this paper, we construct a unified budget-aware training optimization framework that inherently adapts to landscape curvature variations, enhancing training robustness. From this optimization framework, we propose the Unified Budget-Aware (UBA) schedule. Extensive experiments demonstrate UBA *consistently surpasses* the commonly-used schedules across diverse vision and language tasks, spanning network architectures (e.g., ResNet, OLMo) and scales, under different training-iteration budgets. Moreover, our performance improvements come with negligible computational overhead, which constitutes one of its key advantages.

Theoretically and empirically, we observe that the parameter $\varphi$ correlates with the optimization difficulty of the training process, influences the optimal choice of $\varphi$ and UBA's performance. However, the explicit relationship between $\varphi$ and optimization difficulty remains unexplored and no established evaluation metric exists to quantify the optimization difficulty. These limitations motivate future work on optimization difficulty-aware $\varphi$ tuning.

Despite these open questions, the effectiveness, implementation simplicity and ease of tuning make the UBA a practical, must-try schedule for deep learning practitioners. We hope this work could motivate more studies on learning rate scheduling.

## 6    Acknowledgments and disclosure of funding

Z. Lin was supported by National Key R&D Program of China (2022ZD0160300), the NSF China (No. 62276004) and the State Key Laboratory of General Artificial Intelligence.

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

Figure 3: Conceptual illustration: visualization of the motivations behind the proposed modeling approaches. (a) shows the rationale for the modeling approach in (1), while (b) explains the modeling approach for (2). The solid line represents the optimization trajectory, while the dashed circles indicate the local approximation around corresponding minima.

## Appendix

## A    Methodology

**Conceptual illustration**   The optimization landscape of neural networks is highly dynamic due to data distribution and sampling, and sensitive to various factors, such as network architecture, optimization and hyper-parameters. Figure 3(a) illustrates the inherent uncertainties during the training process. The black-and-white surfaces reflect the loss landscape processing one batch, while the colored surface represents the landscape following the next batch. The substantial variations in the landscape between consecutive batches significantly affect the training trajectory, making it challenging to design a one-size-fits-all learning rate strategy that can effectively adapt to such fluctuations. Therefore, we aim to develop a strategy a robust approach to be scheduled leaning rate across diverse optimization landscapes. We construct a budget-iteration-aware framework for learning rate optimization model that guarantees minimal loss within constrained training iterations under the worst-case conditions.

We analyze loss landscape by a quadratic approximation around nearby strict local optima. This approach enables us to capture the key features of the loss surface near these points. Therefore, the trajectory of optimization is impacted by the characteristics of the nearby strict local optima, since the optimization process is inherently shaped by the loss landscape and the key features of the surface are captured by these optima, as illustrated in Figure 3(b), Consequently, we can derive the learning rate within the optimization model by the information of these nearby strict local optima.

**Assumptions**   The optimization behavior of a neural network is fundamentally governed by the geometry of its loss landscape, which emerges from the interaction of parameter configuration, datasets characteristics, sampling, network architecture, optimization preconditioning effect. In the vicinity of any strict local minimum, the loss surface can be approximated by a quadratic method whose curvature is determined by the local Hessian matrix. Thus the optimization trajectory is consequently dominated by the geometric properties (e.g., Hessian features, curvature profiles) of the nearest strict local minima.

Many tasks train models using epochs, where one epoch represents a full pass through the datasets. Under this circumstance, an iteration occurs when the model processes a single batch. While the number of iterations per epoch varies with batch size, training with a fixed epoch budget becomes equivalent to using a fixed iteration count when batch size remains constant. For consistency, this paper uses iterations as our primary unit of optimization model construction and networks training.

The sampling Hessian satisfies: $\mathbb{E}_\xi \left[ n_t n_t^\top \right] \preceq \sigma^2 H_f^{(k)}$ for some constant $\sigma$. This assumption is followed by the work [16, 36]. Here, $\sigma^2$ measures the degree of Hessian inconsistency across data samples, i.e., the variance in the per-sample Hessian matrices. Besides, a smaller $\sigma^2$ indicates a more stable curvature landscape between each data sampling, leading to lower noise amplification during optimization.

**Optimization problem derivation for Section 3.1**   Considering the first constraint $W_{t+1} = W_t - \eta_t H_f^{(k)}(\xi) \left( W_t - \bar{W}^{(k)} \right)$ in min-max optimization (2), we can reformulate it as

$$W_{t+1} - \bar{W}^{(k)} = \left( I - \eta_t H_f^{(k)}(\xi) \right) \left( W_t - \bar{W}^{(k)} \right), \tag{9}$$

where $I$ denotes the identity matrix. By iteratively applying this equation (9), we obtain

$$
\begin{aligned}
&\left( W_{T_{k+1}} - \bar{W}^{(k)} \right) \\
&= \left( I - \eta_{T_{k+1}-1} H_f^{(k)}(\xi) \right) \cdots \left( I - \eta_{T_k+1} H_f^{(k)}(\xi) \right) \left( W_{T_k+1} - \bar{W}^{(k)} \right).
\end{aligned} \tag{10}
$$

Since the matrix $H_f^{(k)}(\xi)$ is positive semi-definite, it possesses $N$ eigenvalues $\lambda_1^{(k)}(f(\xi)), \lambda_2^{(k)}(f(\xi)), \cdots, \lambda_N^{(k)}(f(\xi))$ abbreviated as $\lambda_i^{(k)}$, which satisfy $0 < \lambda_l^{(k)} \leq \lambda_i^{(k)}(f(\xi)) \leq \lambda_u^{(k)}$ for all $i$. Here, $\lambda_l^{(k)}$ and $\lambda_u^{(k)}$ denote the bounds on the non-zero eigenvalues of the Hessian around $k$-th optimum. It also satisfies $\lambda_u^{(k)} \leq \frac{2}{\eta}$ where $\eta$ is the learning rate to guarantee the convergence [56]. Moreover, there exist $N$ linearly independent eigenvectors $u_1^{(k)}(f(\xi)), u_2^{(k)}(f(\xi)), \cdots, u_N^{(k)}(f(\xi))$ abbreviated as $u_i^{(k)}$, which can form the basis for the $N$ dimension vector space. Then we have $H_f^{(k)}(\xi) u_i^{(k)} = \lambda_i^{(k)} u_i^{(k)}$   $(i = 1, 2, \cdots, N)$ and the initial deviation from the optimal solution can be expressed as $W_{T_k} - \bar{W}^{(k)} = \sum_{i=1}^{N} s_i^{(k)} u_i^{(k)}$ where $s_i^{(k)}$ are the coefficients corresponding to the eigenvector components. Thus, we obtain

$$
\begin{aligned}
\left\| W_{T_{k+1}} - \bar{W}^{(k)} \right\|_2^2 &= \sum_{i=1}^{N} (s_i^{(k)})^2 \| u_i^{(k)} \|_2^2 \left[ \prod_{t=1+T_k}^{T_{k+1}} \left( 1 - \eta_t \lambda_i^{(k)} \right) \right]^2 \\
&\leq \sum_{i=1}^{N} (s_i^{(k)})^2 \| u_i^{(k)} \|_2^2 \max_{\lambda_l^{(k)} \leq \lambda_i^{(k)} \leq \lambda_u^{(k)}} \left[ \prod_{t=1+T_k}^{T_{k+1}} \left( 1 - \eta_t \lambda_i^{(k)} \right) \right]^2 \\
&= \max_{\lambda_l^{(k)} \leq \lambda_i^{(k)} \leq \lambda_u^{(k)}} \left[ \prod_{t=1+T_k}^{T_{k+1}} \left( 1 - \eta_t \lambda_i^{(k)} \right) \right]^2 \left\| W_{T_k} - \bar{W}^{(k)} \right\|_2^2
\end{aligned} \tag{11}
$$

# B   Numerical solution and curve fitting

To solve the constrained min-max problem (4) as follow,

$$\min_{\eta_{1+T_k},\eta_{2+T_k},...,\eta_{T_{k+1}}} \max_{\lambda_l^{(k)} \leq \lambda_i^{(k)} \leq \lambda_u^{(k)}} \prod_{t=1+T_k}^{T_{k+1}} \left(1 - \eta_t \lambda_i^{(k)}\right)^2$$

$$s.t. \quad \eta_{1+T_k},\eta_{2+T_k},...,\eta_{T_{k+1}} \in [\eta_{\min}, \eta_{\max}]$$

$$k = 1, 2, \cdots, K$$

we adopt an iterative projected gradient method that alternates between minimizing over the variables $\eta_t$ and maximizing over the parameters $\lambda_i^{(k)}$. Specifically, at each iteration, we update $\eta_t$ with fixed $\lambda_i^{(k)}$ as follows

$$\eta_t \leftarrow \mathcal{P}_{[\eta_{\min},\eta_{\max}]}\left(\eta_t - \alpha \nabla_{\eta_j} \mathcal{L}(\eta_t, \lambda_i^{(k)})\right).$$

For fixed $\eta_t$, we update $\lambda_i^{(k)}$ via

$$\lambda_i^{(k)} \leftarrow \mathcal{P}_{[\lambda_l^{(k)},\lambda_u^{(k)}]}\left(\lambda_i^{(k)} + \beta \nabla_{\lambda_i^{(k)}} \mathcal{L}(\eta_t, \lambda_i^{(k)})\right),$$

where $\mathcal{P}$ is the projection over box constraints, and $\alpha$ and $\beta$ are step sizes.

This approach requires per-network numerical optimization to determine optimal learning rates case-by-case. To circumvent repeated numerical optimization for learning rate scheduling whenever a new network is trained, we propose to fit these solutions with a parametric function universally across networks.

We note that the solution of optimization model is an unordered set $\{\eta_{1+T_k},\eta_{2+T_k},...,\eta_{T_{k+1}}\}$, meaning that any permutation of these values yields the same objective value. To facilitate function approximation and improve the training stability of the network, we sort the solution set in both descending and ascending orders before fitting them with a smooth function. This pre-processing step preserves the optimality of the solution while enhancing the stability and interpretability of the fitted function.

Due to the symmetry between the descending and ascending orders, and without loss of generality, we first fit a function to the descending order solution. The corresponding function for the ascending order can then be easily obtained via a simple transformation.

Our key observation is that the numerical solutions under varying $\lambda_l, \lambda_u$ configurations exhibit two characteristic decay modes. As shown in Figure 4, these transformations manifest through two distinct mechanisms: (i) gradual-to-accelerated decay: gentle initial decrease transitioning to rapid drop (as shown in Figures 4(d),4(e) and 4(f)); (ii) rapid-to-gradual decay: Initial steep decline followed by slow decay (as shown in Figures 4(g),4(h) and 4(i)). These pattern can be formulated as transformations of a base function, which resembles the curvature variation of the transformed $\ell_1$ norm modulated by a parameter [42].

We first isolate the base function by examining the limiting case where $\lambda_l \approx \lambda_u$. In this regime, the numerical solution within each interval $[1 + T_k, T_{k+1}]$ manifests a cosine-like profile (Figures 4(a),4(b) and 4(c)), suggesting the base form:

$$1 + \cos\left(\frac{2\left[(t - T_k) - 1\right]\pi}{2\left(T_{k+1} - T_k\right)}\right).$$

Then we construct the fitting function as:

$$(\eta_{\max} - \eta_{\min}) \frac{a\left[1 + \cos\left(\frac{2[(t-T_k)-1]\pi}{2(T_{k+1}-T_k)}\right)\right]}{b + c\left[1 + \cos\left(\frac{2[(t-T_k)-1]\pi}{2(T_{k+1}-T_k)}\right)\right]} + \eta_{\min}, \tag{12}$$

where $\{a, b, c\}$ control learning rate decay modes.

In addition, through systematic fitting across nine configurations (Table 4), we discover a universal scaling relationship among the parameters:

$$\frac{b + 2c}{a} \approx 2. \tag{13}$$

Table 4: The parameter details of curve fitting.

| hyper-parameters | | | | fitted parameters | | | | objective value(log)(4) | |
|---|---|---|---|---|---|---|---|---|---|
| $\eta_{\min}$ | $\eta_{\max}$ | $\lambda_l^{(k)}$ | $\lambda_u^{(k)}$ | a | b | c | $\varphi$ | by numerical solution | by fitting function |
| 0 | 10 | 1.89 | 2.60 | 0.26 | 0.51 | 0.00 | 1.99 | 792.67 | 750.83 |
| 0 | 10 | 2.39 | 2.61 | 4.16 | 7.35 | 0.53 | 1.80 | 798.38 | 770.38 |
| 0 | 10 | 5.29 | 5.61 | 4.84 | 7.32 | 1.28 | 1.56 | 1118.79 | 1105.80 |
| 0 | 10 | 18.89 | 260.11 | 1124.67 | 212.28 | 1011.94 | 0.19 | 2933.34 | 2933.62 |
| 0 | 5 | 28.89 | 200.11 | 755.91 | 85.01 | 719.40 | 0.12 | 2607.11 | 2591.51 |
| 0 | 2 | 50.89 | 150.11 | 0.27 | 0.05 | 0.24 | 0.17 | 2031.66 | 2075.50 |
| 1 | 10 | 10.01 | 150.11 | 77.45 | 684.15 | $-268.20$ | 8.40 | 2373.58 | 2431.42 |
| 2 | 10 | 18.89 | 50.11 | 0.11 | 2.52 | $-1.15$ | 21.02 | 2007.68 | 2044.84 |
| 3 | 10 | 20.01 | 180.11 | 0.07 | 3.07 | $-1.47$ | 38.34 | 2604.98 | 2632.21 |

Table 4 lists the fitted parameters $\{a, b, c\}$, where the mean absolute error is 0.04. Figure 5 visualizes the variability of the $\frac{b+2c}{a}$ by shading the region within $\pm 1$ standard deviation from the theoretical value in Figure 5. The empirical results falls within $\pm 1\sigma$ of the predicted value $(2.0 \pm 0.04)$ in most of cases, demonstrating strong agreement with the assumption relation (13).

This empirical relationship (13) allows us to reduce the original 3-parameter system to 2 degrees of freedom as follows,

$$(\eta_{\max} - \eta_{\min}) \frac{2a \left[1 + \cos\left(\frac{2[(t-T_k)-1]\pi}{2(T_{k+1}-T_k)}\right)\right]}{2b + (2a-b)\left[1 + \cos\left(\frac{2[(t-T_k)-1]\pi}{2(T_{k+1}-T_k)}\right)\right]} + \eta_{\min}. \tag{14}$$

Actually, it is a 1-parameter system

$$(\eta_{\max} - \eta_{\min}) \frac{2 \left[1 + \cos\left(\frac{2[(t-T_k)-1]\pi}{2(T_{k+1}-T_k)}\right)\right]}{2\varphi + (2-\varphi)\left[1 + \cos\left(\frac{2[(t-T_k)-1]\pi}{2(T_{k+1}-T_k)}\right)\right]} + \eta_{\min}, \tag{15}$$

where $\varphi := b/a$. We list the value of fitted parameter $\varphi$ in Table 4. Since the equation (14) and the equation (15) are mathematically equivalent, we adopt the equation (15) as the canonical representation throughout this work.

Then the corresponding function for the ascending order can then be easily obtained via a phase shift of $\pi$ as follow,

$$(\eta_{\max} - \eta_{\min}) \frac{2 \left[1 + \cos\left(\frac{2[(t-T_k)-1]\pi}{2(T_{k+1}-T_k)} + \pi\right)\right]}{2\varphi + (2-\varphi)\left[1 + \cos\left(\frac{2[(t-T_k)-1]\pi}{2(T_{k+1}-T_k)} + \pi\right)\right]} + \eta_{\min}, \tag{16}$$

In summary, we obtain the $\eta_t$ within the interval $t \in [1 + T_k, T_{k+1}]$ as follows

$$\eta_t = (\eta_{\max} - \eta_{\min}) \frac{2 \left[1 + \cos\left(\frac{[2(t-T_k)-1]\pi}{2(T_{k+1}-T_k)} + (k-1)\pi\right)\right]}{2\varphi + (2-\varphi)\left[1 + \cos\left(\frac{[2(t-T_k)-1]\pi}{2(T_{k+1}-T_k)} + (k-1)\pi\right)\right]} + \eta_{\min}, \tag{17}$$

where $\varphi$ is the hyper-parameter controlling the variation speed of $\eta_t$.

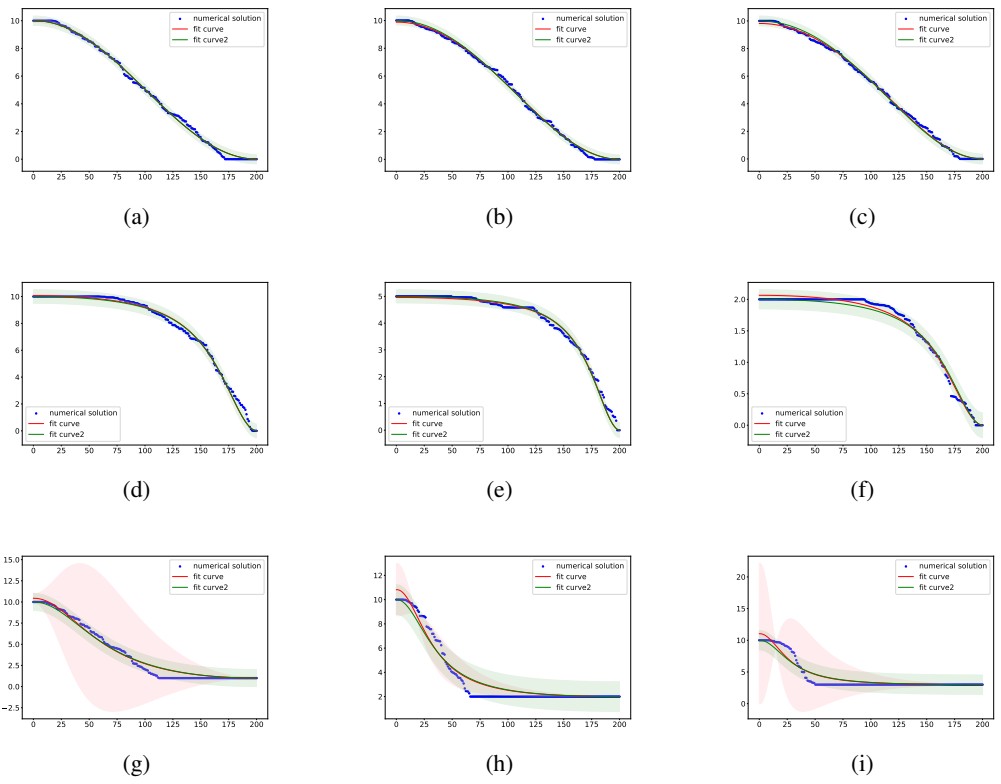

Figure 4: The curve fitting of numerical solutions.

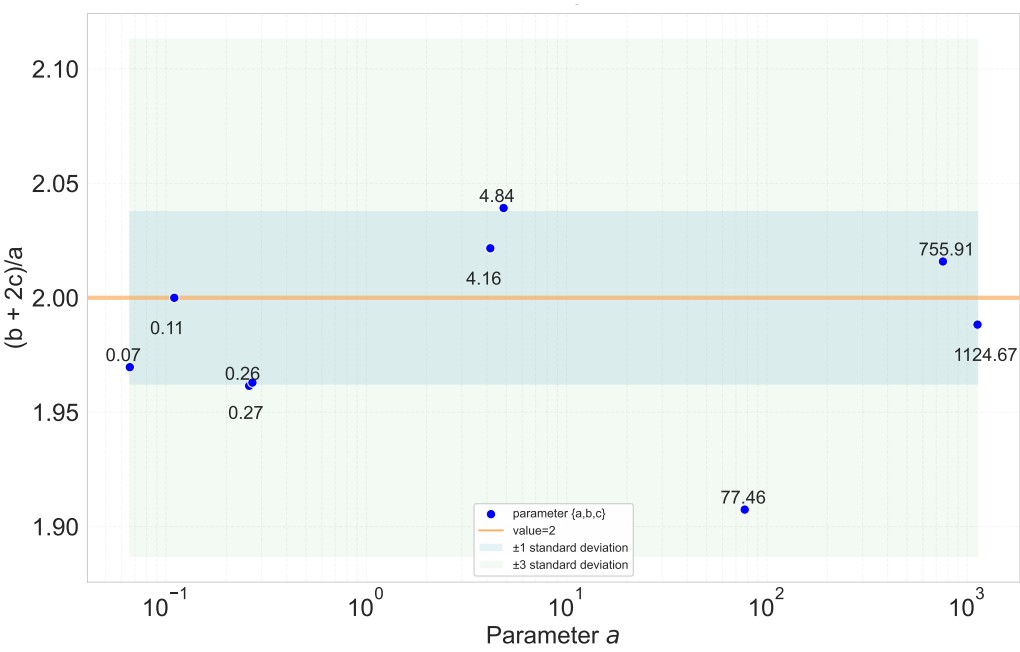

Figure 5: Visualization of relation between parameter $a$, $b$ and $c$.

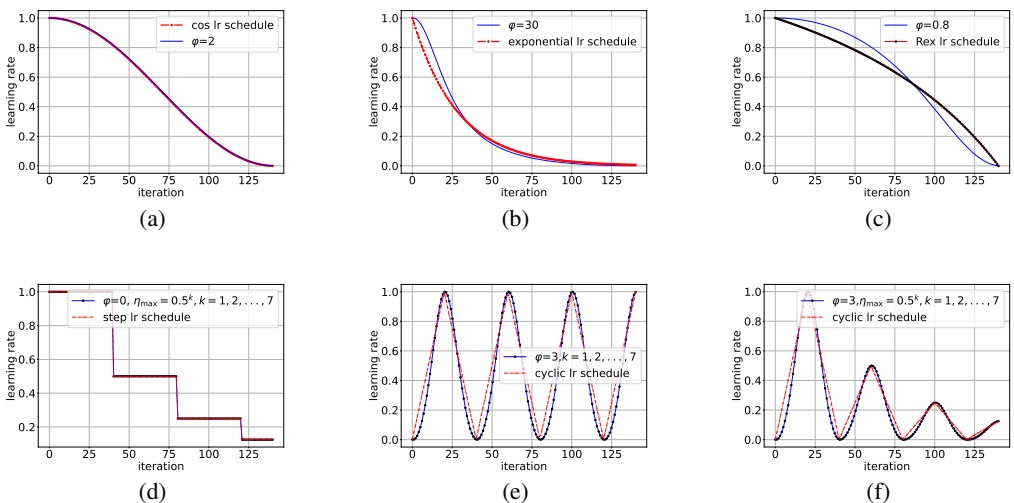

Figure 6: Adaptive simulation by UBA schedule.

## C  Adaptive approximation of existing schedules

Our schedule has adaptability owing to the parameter $\varphi$, enabling it to approximate the behavior of existing schedules (e.g.step decay, cosine annealing, cyclic schedule or Rex schedule) through simple parameter adjustments. More importantly, it outperforms these schedules by jointly optimizing training stability and final accuracy.

We visualize the adaptive simulation of in Figure 6. By setting our parameter $\varphi$ to 2, the UBA schedule degenerates to a cosine schedule (Figure 6(a)). By setting $\varphi$ to large value such as 30, the UBA schedule mimics exponential schedule (Figure 6(b)). By setting $\varphi$ to small value such as 0.8, the UBA schedule mimics Rex schedule (Figure 6(c)). By setting $\varphi$ to 0 , the proposed schedule degenerates to a constant. Therefore, we can control $\eta_{\max}$ as piece-wise value depending on iteration, and the UBA schedule degenerates to a step schedule (Figure 6(d)). By setting $k$ to any integer larger than 1, the UBA schedule possesses cyclic characters, thus it mimics cyclic schedule (Figure 6(e) 6(f)). Since the OneCycle learning rate schedule can be viewed as a single-period version of cyclical learning rates, the UBA schedule mimics the OneCycle approach by emulating the cyclical schedule.

# D    Propositions and lemmas

## D.1    Proof of Proposition 1

**Proposition    1**    *The    fitted    function    (5)    is    the    exact    closed-form    solution*
$\frac{2\left[1+\cos\left(\frac{2[(t-T_k)-1]\pi}{2(T_{k+1}-T_k)}\right)\right]}{2\varphi+(2-\varphi)\left[1+\cos\left(\frac{2[(t-T_k)-1]\pi}{2(T_{k+1}-T_k)}\right)\right]}$ *to the min-max optimization problem:*

$$\min_{\eta_{1+T_k},\eta_{2+T_k},\ldots,\eta_{T_{k+1}}} \max_{\lambda_l^{(k)}\leq\lambda_i^{(k)}\leq\lambda_u^{(k)}} \prod_{t=1+T_k}^{T_{k+1}} \left[\left(1-\left((\frac{1}{\lambda_l^{(k)}}-\frac{1}{\lambda_u^{(k)}})\eta_t+\frac{1}{\lambda_u^{(k)}}\right)\lambda_i^{(k)}\right)\right]^2, \quad (18)$$

*when the hyper-parameter $\varphi$ are determined by $\lambda_l^{(k)}$ and $\lambda_u^{(k)}$ through the relation $\varphi = 2\frac{\lambda_u^{(k)}}{\lambda_l^{(k)}}$ and*
$\eta_{\max} = 1, \quad \eta_{\min} = 0.$

*Proof.* According to the theorem in [14], among all polynomials of degree $n$ in $\mu$ that take the value $+1$ at $\mu = d > 1$, the Chebyshev polynomial $S_n(\mu) = \frac{C_n(\mu)}{C_n(d)}$ is the unique solution that minimizes the maximum absolute value over the interval $(-1, +1)$. Here, $C_n(\mu) = \cos(n\arccos(\mu))$ is the $n$-th order Chebyshev polynomial.

We define the polynomial $Q(\lambda_i^{(k)})$ as:

$$Q(\lambda_i^{(k)}) := \prod_{t=T_k+1}^{T_{k+1}} \left[1-\left(\left(\frac{1}{\lambda_l^{(k)}}-\frac{1}{\lambda_u^{(k)}}\right)\eta_t+\frac{1}{\lambda_u^{(k)}}\right)\lambda_i^{(k)}\right],$$

which is a degree $n = T_{k+1} - T_k$ polynomial in $\lambda_i^{(k)}$.

Next, we introduce the variable transformation:

$$\gamma = \frac{-2\lambda_i^{(k)} + \lambda_l^{(k)} + \lambda_u^{(k)}}{\lambda_u^{(k)} - \lambda_l^{(k)}},$$

which maps the interval $\lambda_l^{(k)} \leq \lambda_i^{(k)} \leq \lambda_u^{(k)}$ to the interval $-1 \leq \gamma \leq 1$, such that $\lambda_i^{(k)} = \lambda_u^{(k)}$ corresponds to $\gamma = -1$ and $\lambda_i^{(k)} = \lambda_l^{(k)}$ corresponds to $\gamma = 1$.

The polynomial $P(\gamma) = Q(\lambda_i^{(k)})$ now satisfies the condition in the theorem from [14], i.e.,

$$P\left(\frac{\lambda_l^{(k)} + \lambda_u^{(k)}}{\lambda_u^{(k)} - \lambda_l^{(k)}}\right) = 1 \quad \text{since} \quad Q(0) = 1.$$

Therefore, the original min-max problem

$$\min_{\eta_{T_k+1},\eta_{T_k+2},\ldots,\eta_{T_{k+1}}} \max_{\lambda_l^{(k)}\leq\lambda_i^{(k)}\leq\lambda_u^{(k)}} \prod_{t=T_k+1}^{T_{k+1}} \left[1-\left(\left(\frac{1}{\lambda_l^{(k)}}-\frac{1}{\lambda_u^{(k)}}\right)\eta_t+\frac{1}{\lambda_u^{(k)}}\right)\lambda_i^{(k)}\right]^2,$$

is equivalent to the problem of finding a polynomial in $\gamma$ of degree $T_{k+1} - T_k$ that minimizes the maximum absolute value in the interval $-1 \leq \gamma \leq 1$.

By the theorem in [14], the desired solution to the problem is the Chebyshev polynomial:

$$S_{T_{k+1}-T_k}(\gamma) = \frac{C_{T_{k+1}-T_k}(\gamma)}{C_{T_{k+1}-T_k}\left(\frac{\lambda_l^{(k)}+\lambda_u^{(k)}}{\lambda_u^{(k)}-\lambda_l^{(k)}}\right)}.$$

To match $P(\gamma) = S_{T_{k+1}-T_k}(\gamma)$, we equate the corresponding zeros. The roots of $Q(\lambda_i^{(k)}) = 0$ are given by:

$$\lambda_i^{(k)} = \frac{1}{\left(\frac{1}{\lambda_l^{(k)}}-\frac{1}{\lambda_u^{(k)}}\right)\eta_t+\frac{1}{\lambda_u^{(k)}}} \quad \text{for} \quad t = 1, 2, \ldots, T_{k+1}-T_k.$$

On the other hand, the zeros of $P(\gamma)$ correspond to the values of $\gamma$ where the polynomial $P(\gamma)$ vanishes, and these zeros are given by:

$$\gamma_t = \frac{\lambda_l^{(k)} + \lambda_u^{(k)}}{\lambda_u^{(k)} - \lambda_l^{(k)}} - \frac{2}{\left[ \left( \frac{1}{\lambda_l^{(k)}} - \frac{1}{\lambda_u^{(k)}} \right) \eta_t + \frac{1}{\lambda_u^{(k)}} \right] (\lambda_u^{(k)} - \lambda_l^{(k)})} \quad \text{for} \quad t = 1, 2, \ldots, T_{k+1} - T_k.$$

The zeros of the Chebyshev polynomial $S_{T_{k+1} - T_k}(\gamma)$ are given by the values:

$$\cos\left( \frac{(2(t - T_k) - 1)\pi}{2(T_{k+1} - T_k)} \right) \quad \text{for} \quad t = 1, 2, \ldots, T_{k+1} - T_k.$$

Equating the zeros of $S_{T_{k+1} - T_k}(\gamma)$ and $P(\gamma)$, we have:

$$\frac{\lambda_l^{(k)} + \lambda_u^{(k)}}{\lambda_u^{(k)} - \lambda_l^{(k)}} - \frac{2}{\left( \left( \frac{1}{\lambda_l^{(k)}} - \frac{1}{\lambda_u^{(k)}} \right) \eta_t + \frac{1}{\lambda_u^{(k)}} \right) (\lambda_u^{(k)} - \lambda_l^{(k)})} = \cos\left( \frac{(2(t - T_k) - 1)\pi}{2(T_{k+1} - T_k)} \right).$$

Solving this equation for $\eta_t$, we obtain:

$$\eta_t = \frac{\left(1 + \cos\left( \frac{(2(t-T_k)-1)\pi}{2(T_{k+1}-T_k)} \right)\right)}{2\frac{\lambda_u^{(k)}}{\lambda_l^{(k)}} - \left( \frac{\lambda_u^{(k)}}{\lambda_l^{(k)}} - 1 \right) \left( 1 + \cos\left( \frac{(2(t-T_k)-1)\pi}{2(T_{k+1}-T_k)} \right) \right)}.$$

$\square$

## D.2 Proof of Proposition 2

**Proposition 2** *Consider the training process within the interval $t \in [1 + T_k, T_{k+1}]$. When the hyper-parameter $\varphi$ is set sufficiently close to 2, the proposed learning rate scheduling formula reduces to the cosine learning rate schedule.*

*Proof.*

$$\begin{aligned}
& \lim_{\varphi \to 2} (\eta_{\max} - \eta_{\min}) \frac{2 \left[ 1 + \cos\left( \frac{2[(t-T_k)-1]\pi}{2(T_{k+1}-T_k)} \right) \right]}{2\varphi + (2 - \varphi) \left[ 1 + \cos\left( \frac{2[(t-T_k)-1]\pi}{2(T_{k+1}-T_k)} \right) \right]} + \eta_{\min} \\
&= (\eta_{\max} - \eta_{\min}) \left( \frac{1}{2} + \frac{1}{2} \cos\left( \frac{(2(t - T_k) - 1)\pi}{2(T_{k+1} - T_k)} \right) \right) + \eta_{\min} \\
&= \frac{1}{2} (\eta_{\max} - \eta_{\min}) \left( 1 + \cos\left( \frac{(t - T_k)\pi}{T_{k+1} - T_k} \right) \cos\left( \frac{\pi}{2(T_{k+1} - T_k)} \right) \right. \\
& \quad \left. + \sin\left( \frac{(t - T_k)\pi}{T_{k+1} - T_k} \right) \sin\left( \frac{\pi}{2(T_{k+1} - T_k)} \right) \right) + \eta_{\min} \\
&\approx \frac{1}{2} (\eta_{\max} - \eta_{\min}) \left( 1 + \cos\left( \frac{(t - T_k)\pi}{T_{k+1} - T_k} \right) \right) + \eta_{\min}.
\end{aligned} \tag{19}$$

The final approximation holds because each training phase typically contains many iterations, i.e., making $\frac{\pi}{T_{k+1}-T_k} \approx 0$. Consequently, $\sin\left(\frac{\pi}{2(T_{k+1}-T_k)}\right) \approx 0$ and $\cos\left(\frac{\pi}{2(T_{k+1}-T_k)}\right) \approx 1$. $\square$

## D.3 Proof of Theorem 1

**Lemma 1.** *Let $\eta_j$ be the learning rate at the iteration $j \in [1 + T_k, t]$, defined by the proposed form (5). Let $\lambda_i^{(k)}$ be the eigenvalues of the Hessian matrix $H_f^{(k)}(\xi)$ at the $k$-th strict minimum $\bar{W}^{(k)}$. Denote $\tau := \frac{4\lambda_l^{(k)}(\eta_{\max}-\eta_{\min})\left[1+\cos\left(\frac{[2(t-T_k)-1]\pi}{2(T_{k+1}-T_k)}\right)\right](T_{k+1}-T_k)}{(\varphi-2)\pi}$. Then for any $t \in [1 + T_k, T_{k+1}]$, the following inequalities hold,*

*Case 1 ($\varphi > 2$):*

$$\prod_{j=1+T_k}^{t} \left[(1 - \eta_j \lambda_i^{(k)})\right]^2$$

$$\leq \exp\left(-2\lambda_i^{(k)} \eta_{\min}(t - T_k)\right) \cdot \left[\frac{4(T_{k+1} - T_k) + (\varphi - 2)\pi}{4(T_{k+1} - T_k) + (\varphi - 2)\pi(t - T_k)}\right]^{\tau}. \tag{20}$$

*Case 2 ($\varphi < 2$):*

$$\prod_{j=1+T_k}^{t} \left[(1 - \eta_j \lambda_i^{(k)})\right]^2$$

$$\leq \exp\left(-2\lambda_i^{(k)} \eta_{\min}(t - T_k)\right) \cdot \left[\frac{(2\varphi + 2\pi - \varphi\pi) - \frac{(2-\varphi)\pi}{(T_{k+1}-T_k)}(t - T_k - 0.5)}{(2\varphi + 2\pi - \varphi\pi) + \frac{(2-\varphi)\pi}{2(T_{k+1}-T_k)}}\right]^{-\tau}. \tag{21}$$

*Proof.* By the fact that $1 - x \leq \exp(-x)$, we have

$$\prod_{j=1+T_k}^{t} \left[(1 - \eta_j \lambda_i^{(k)})\right]^2 \leq \exp\left(-2 \sum_{j=1+T_k}^{t} \eta_j \lambda_i^{(k)}\right). \tag{22}$$

Substituting the expression for $\eta_j$ by the proposed form (5) yields:

$$\exp\left(-2 \sum_{j=1+T_k}^{t} \eta_j \lambda_i^{(k)}\right)$$

$$= \exp\left(-2\lambda_i^{(k)} \sum_{j=1+T_k}^{t} (\eta_{\max} - \eta_{\min}) \frac{2(1 + \cos\left(\frac{(2(j-T_k)-1)\pi}{2(T_{k+1}-T_k)}\right))}{2\varphi + (2-\varphi)(1 + \cos\left(\frac{(2(j-T_k)-1)\pi}{2(T_{k+1}-T_k)}\right)} + \eta_{\min}\right) \tag{23}$$

$$= \exp\left(-2\lambda_i^{(k)} \eta_{\min}(t - T_k)\right)$$

$$\cdot \exp\left(-2\lambda_i^{(k)} \sum_{j=1+T_k}^{t} (\eta_{\max} - \eta_{\min}) \frac{2\left(1 + \cos\left(\frac{(2(j-T_k)-1)\pi}{2(T_{k+1}-T_k)}\right)\right)}{2\varphi + (2-\varphi)\left(1 + \cos\left(\frac{(2(j-T_k)-1)\pi}{2(T_{k+1}-T_k)}\right)\right)}\right).$$

The behavior of this term depends on the value of the parameter $\varphi$. We consider the following cases:

- if $\varphi \geq 2, 2 - \varphi \leq 0$

  Since

  $$\cos(\frac{(2(t - T_k) - 1)\pi}{2(T_{k+1} - T_k)}) \geq \cos(\frac{(2(j - T_k) - 1)\pi}{2(T_{k+1} - T_k)}) \geq 1 - \left(\frac{(2(j - T_k) - 1)\pi}{2(T_{k+1} - T_k)}\right)$$

  $$\geq 1 - \pi \left(\frac{(j - T_k)}{(T_{k+1} - T_k)}\right), \tag{24}$$

  we have

  $$\sum_{j=1+T_k}^{t} \frac{1}{2\varphi + (2-\varphi)\left(1 + \cos\left(\frac{(2(j-T_k)-1)\pi}{2(T_{k+1}-T_k)}\right)\right)}$$

  $$\leq \sum_{j=1+T_k}^{t} \frac{1}{2\varphi + (2-\varphi)\left(1 + 1 - \pi\left(\frac{(j-T_k)}{(T_{k+1}-T_k)}\right)\right)} \tag{25}$$

  $$= \sum_{j=1+T_k}^{t} \frac{1}{4 + \frac{(\varphi-2)\pi}{(T_{k+1}-T_k)}(j - T_k)}.$$

Then the equality (23) becomes

$$\exp\left(-2\lambda_i^{(k)}\eta_{\min}\left(t-T_k\right)\right)$$

$$\cdot\exp\left(-2\lambda_i^{(k)}\sum_{j=1+T_k}^{t}\left(\eta_{\max}-\eta_{\min}\right)\frac{2\left(1+\cos\left(\frac{(2(j-T_k)-1)\pi}{2(T_{k+1}-T_k)}\right)\right)}{2\varphi+(2-\varphi)\left(1+\cos\left(\frac{(2(j-T_k)-1)\pi}{2(T_{k+1}-T_k)}\right)\right)}\right)$$

$$\leq\exp\left(-2\lambda_i^{(k)}\eta_{\min}(t-T_k)\right)\exp\left[-4\lambda_i^{(k)}(\eta_{\max}-\eta_{\min})\left(1+\cos\left(\frac{(2(t-T_k)-1)\pi}{2(T_{k+1}-T_k)}\right)\right)\right.$$

$$\left.\sum_{j=1+T_k}^{t}\frac{1}{2\varphi+(2-\varphi)\left(1+\cos\left(\frac{(2(j-T_k)-1)\pi}{2(T_{k+1}-T_k)}\right)\right)}\right] \tag{26}$$

$$\leq\exp\left(-2\lambda_i^{(k)}\eta_{\min}\left(t-T_k\right)\right)\exp\left[-4\lambda_i^{(k)}\left(\eta_{\max}-\eta_{\min}\right)\left[1+\cos\left(\frac{(2(t-T_k)-1)\pi}{2(T_{k+1}-T_k)}\right)\right]\right.$$

$$\left.\sum_{j=1+T_k}^{t}\frac{1}{4+\frac{(\varphi-2)\pi}{(T_{k+1}-T_k)}(j-T_k)}\right].$$

Note that the function $h(j):=\frac{1}{4+\frac{(\varphi-2)\pi}{T_{k+1}-T_k}(j-T_k)}$ is monotone decreasing over $j\in[1+T_k,t]$, so we can bound the summation from below by the integral:

$$\sum_{j=1+T_k}^{t}\frac{1}{4+\frac{(\varphi-2)\pi}{T_{k+1}-T_k}(j-T_k)}$$

$$\geq\int_{1+T_k}^{t+1}\frac{1}{4+\frac{(\varphi-2)\pi}{T_{k+1}-T_k}(j-T_k)}dj \tag{27}$$

$$\geq\int_{1+T_k}^{t}\frac{1}{4+\frac{(\varphi-2)\pi}{T_{k+1}-T_k}(j-T_k)}dj$$

$$=\frac{(T_{k+1}-T_k)}{(\varphi-2)\pi}\ln\left(\frac{4(T_{k+1}-T_k)+(\varphi-2)\pi(t-T_k)}{4(T_{k+1}-T_k)+(\varphi-2)\pi}\right).$$

Thus the inequality (26) becomes

$$\exp\left(-2\lambda_i^{(k)}\eta_{\min}\left(t-T_k\right)\right)\exp\left[-4\lambda_i^{(k)}\left(\eta_{\max}-\eta_{\min}\right)\left[1+\cos\left(\frac{(2(t-T_k)-1)\pi}{2(T_{k+1}-T_k)}\right)\right]\right.$$

$$\left.\sum_{j=1+T_k}^{t}\frac{1}{4+\frac{(\varphi-2)\pi}{(T_{k+1}-T_k)}(j-T_k)}\right]$$

$$\leq\exp\left(-2\lambda_i^{(k)}\eta_{\min}(t-T_k)\right)\exp\left[4\lambda_i^{(k)}(\eta_{\max}-\eta_{\min})\left(1+\cos\left(\frac{(2(t-T_k)-1)\pi}{2(T_{k+1}-T_k)}\right)\right)\right.$$

$$\left.\frac{(T_{k+1}-T_k)}{(\varphi-2)\pi}\ln\left(\frac{4(T_{k+1}-T_k)+(\varphi-2)\pi}{4(T_{k+1}-T_k)+(\varphi-2)\pi(t-T_k)}\right)\right] \tag{28}$$

$$=\exp\left(-2\lambda_i^{(k)}\eta_{\min}(t-T_k)\right)\left(\frac{4(T_{k+1}-T_k)+(\varphi-2)\pi}{4(T_{k+1}-T_k)+(\varphi-2)\pi(t-T_k)}\right)^{\tau}$$

$$\leq\exp\left(-2\lambda_i^{(k)}\eta_{\min}(t-T_k)\right)\left(\frac{4(T_{k+1}-T_k)+(\varphi-2)\pi}{4(T_{k+1}-T_k)+(\varphi-2)\pi(t-T_k)}\right)^{\tau}.$$

The last inequality is because $\lambda_l^{(k)}\leq\lambda_i^{(k)}$ and $\frac{4(T_{k+1}-T_k)+(\varphi-2)\pi}{4(T_{k+1}-T_k)+(\varphi-2)\pi(t-T_k)}\leq 1$.

- if $\varphi \leq 2$, $2 - \varphi \geq 0$ Since

$$1 + \cos\left(\frac{(2(j - T_k) - 1)\pi}{2(T_{k+1} - T_k)}\right) \leq \pi - \frac{(2(j - T_k) - 1)\pi}{2(T_{k+1} - T_k)}. \tag{29}$$

We obtain

$$
\begin{aligned}
&\exp\left(-2\lambda_i^{(k)} \eta_{\min} (t - T_k)\right) \\
&\cdot \exp\left(-2\lambda_i^{(k)} \sum_{j=1+T_k}^{t} (\eta_{\max} - \eta_{\min}) \frac{2\left(1 + \cos\left(\frac{(2(j-T_k)-1)\pi}{2(T_{k+1}-T_k)}\right)\right)}{2\varphi + (2 - \varphi)\left(1 + \cos\left(\frac{(2(j-T_k)-1)\pi}{2(T_{k+1}-T_k)}\right)\right)}\right) \\
&\leq \exp\left(-2\lambda_i^{(k)} \eta_{\min}(t - T_k)\right) \exp\left[-4\lambda_i^{(k)} (\eta_{\max} - \eta_{\min})\left(1 + \cos\left(\frac{(2(t - T_k) - 1)\pi}{2(T_{k+1} - T_k)}\right)\right)\right. \\
&\left. \sum_{j=1+T_k}^{t} \frac{1}{2\varphi + (2 - \varphi)(1 + \cos\left(\frac{(2(j-T_k)-1)\pi}{2(T_{k+1}-T_k)}\right))}\right] \tag{30} \\
&\leq \exp\left(-2\lambda_i^{(k)} \eta_{\min} (t - T_k)\right) \exp\left[-4\lambda_i^{(k)} (\eta_{\max} - \eta_{\min})\left[1 + \cos\left(\frac{(2(t - T_k) - 1)\pi}{2(T_{k+1} - T_k)}\right)\right]\right. \\
&\left. \sum_{j=1+T_k}^{t}\left(\frac{1}{(2\varphi + 2\pi - \varphi\pi) - \frac{(2-\varphi)\pi}{(T_{k+1}-T_k)}(j - T_k - 0.5)}\right)\right].
\end{aligned}
$$

Because function $h(j) := \frac{1}{(2\varphi + 2\pi - \varphi\pi) - \frac{(2-\varphi)\pi}{(T_{k+1}-T_k)}(j - T_k - 0.5)}$ is monotone increasing in the range $[1 + T_k, +\infty)$, then it holds that

$$
\begin{aligned}
&\sum_{j=1+T_k}^{t} \frac{1}{(2\varphi + 2\pi - \varphi\pi) - \frac{(2-\varphi)\pi}{(T_{k+1}-T_k)}(j - T_k - 0.5)} \\
&\geq \int_{T_k}^{t} \frac{1}{(2\varphi + 2\pi - \varphi\pi) - \frac{(2-\varphi)\pi}{(T_{k+1}-T_k)}(j - T_k - 0.5)} dj \tag{31} \\
&= -\frac{(T_{k+1} - T_k)}{(2 - \varphi)\pi} \ln\left(\frac{(2\varphi + 2\pi - \varphi\pi) - \frac{(2-\varphi)\pi}{(T_{k+1}-T_k)}(t - T_k - 0.5)}{(2\varphi + 2\pi - \varphi\pi) + \frac{(2-\varphi)\pi}{2(T_{k+1}-T_k)}}\right)
\end{aligned}
$$

Thus, we have

$$\exp\left(-2\lambda_i^{(k)}\eta_{\min}(t-T_k)\right)\exp\left[-4\lambda_i^{(k)}\left(\eta_{\max}-\eta_{\min}\right)\left[1+\cos\left(\frac{(2(t-T_k)-1)\pi}{2(T_{k+1}-T_k)}\right)\right]\right.$$

$$\left.\sum_{j=1+T_k}^{t}\left(\frac{1}{(2\varphi+2\pi-\varphi\pi)-\frac{(2-\varphi)\pi}{(T_{k+1}-T_k)}(j-T_k-0.5)}\right)\right]$$

$$\leq\exp\left(-2\lambda_i^{(k)}\eta_{\min}(t-T_k)\right)\exp\left[4\lambda_i^{(k)}(\eta_{\max}-\eta_{\min})\left(1+\cos\left(\frac{(2(t-T_k)-1)\pi}{2(T_{k+1}-T_k)}\right)\right)\right.$$

$$\left.\frac{(T_{k+1}-T_k)}{(2-\varphi)\pi}\ln\left(\frac{(2\varphi+2\pi-\varphi\pi)-\frac{(2-\varphi)\pi}{(T_{k+1}-T_k)}(t-T_k-0.5)}{(2\varphi+2\pi-\varphi\pi)+\frac{(2-\varphi)\pi}{2(T_{k+1}-T_k)}}\right)\right]$$

(32)

$$\leq\exp\left(-2\lambda_i^{(k)}\eta_{\min}(t-T_k)\right)$$

$$\cdot\left(\frac{(2\varphi+2\pi-\varphi\pi)-\frac{(2-\varphi)\pi}{(T_{k+1}-T_k)}(t-T_k-0.5)}{(2\varphi+2\pi-\varphi\pi)+\frac{(2-\varphi)\pi}{2(T_{k+1}-T_k)}}\right)^{-\tau}$$

$$\leq\exp\left(-2\lambda_i^{(k)}\eta_{\min}(t-T_k)\right)$$

$$\cdot\left(\frac{(2\varphi+2\pi-\varphi\pi)-\frac{(2-\varphi)\pi}{(T_{k+1}-T_k)}(t-T_k-0.5)}{(2\varphi+2\pi-\varphi\pi)+\frac{(2-\varphi)\pi}{2(T_{k+1}-T_k)}}\right)^{-\tau}.$$

The last inequality is because $\lambda_l^{(k)}\leq\lambda_i^{(k)}$ and $\frac{(2\varphi+2\pi-\varphi\pi)-\frac{(2-\varphi)\pi}{(T_{k+1}-T_k)}(t-T_k-0.5)}{(2\varphi+2\pi-\varphi\pi)+\frac{(2-\varphi)\pi}{2(T_{k+1}-T_k)}}\leq 1.$

$\square$

**Theorem 1** *Let $n_t = H_f^{(k)}\left(W_t-\bar{W}^{(k)}\right)-H_f^{(k)}(\xi)\left(W_t-\bar{W}^{(k)}\right)$ be the stochastic curvature noise introduced by sampling at iteration t. Assume that the sampling Hessian satisfies: $\mathbb{E}_\xi\left[n_t n_t^\top\right]\preceq \sigma^2 H_f^{(k)}$ for some constant $\sigma$. Denote $\tau := \frac{4\lambda_l^{(k)}(\eta_{\max}-\eta_{\min})\left[1+\cos\left(\frac{[2(t-T_k)-1]\pi}{2(T_{k+1}-T_k)}\right)\right](T_{k+1}-T_k)}{(\varphi-2)\pi}$. If we set the learning rate as the proposed form (5), the loss uncertainty introduced by stochastic gradient method within the interval $t\in[1+T_k,T_{k+1}]$ can be bounded by two terms,*
**For $\varphi > 2$:**

$$\mathbb{E}\left[\mathcal{L}(f(W_t,\xi))-\mathcal{L}(f(\bar{W}^{(k)},\xi))\right]$$

$$\leq\left[\frac{4(T_{k+1}-T_k)+(\varphi-2)\pi}{4(T_{k+1}-T_k)+(\varphi-2)\pi(t-T_k)}\right]^\tau\cdot\exp\left(-2\lambda_i^{(k)}\eta_{\min}(t-T_k)\right)\lambda_u^{(k)}\|W_{1+T_k}-\bar{W}^{(k)}\|_2^2$$

$$+\sigma^2\sum_{i=1+T_k}^{t}\eta_i^2\sum_{j=1}^{N}(\lambda_j^{(k)})^2\exp\left(-2\lambda_j^{(k)}\eta_{\min}(t-i)\right)\cdot\left[\frac{4(T_{k+1}-T_k)+(\varphi-2)\pi(i-T_k)}{4(T_{k+1}-T_k)+(\varphi-2)\pi(t-T_k)}\right]^\tau.$$

(33)

**For $\varphi < 2$:**

$$\mathbb{E}\left[\mathcal{L}(f(W_t,\xi))-\mathcal{L}(f(\bar{W}^{(k)},\xi))\right]$$

$$\leq\left[\frac{(2\varphi+2\pi-\varphi\pi)-\frac{(2-\varphi)\pi}{(T_{k+1}-T_k)}(t-T_k-0.5)}{(2\varphi+2\pi-\varphi\pi)+\frac{(2-\varphi)\pi}{2(T_{k+1}-T_k)}}\right]^{-\tau}\cdot\exp\left(-2\lambda_i^{(k)}\eta_{\min}(t-T_k)\right)\lambda_u^{(k)}\|W_{1+T_k}-\bar{W}^{(k)}\|_2^2$$

(34)

$$+\sigma^2\sum_{i=1+T_k}^{t}\eta_i^2\sum_{j=1}^{N}\left(\lambda_j^{(k)}\right)^2\exp\left(-2\lambda_j^{(k)}\eta_{\min}(t-i)\right)\cdot\left[\frac{(2\varphi+2\pi-\varphi\pi)-\frac{(2-\varphi)\pi}{(T_{k+1}-T_k)}(t-T_k-0.5)}{(2\varphi+2\pi-\varphi\pi)-\frac{(2-\varphi)\pi}{(T_{k+1}-T_k)}(i-T_k-0.5)}\right]^{-\tau}.$$

*Proof.* Let $H_f^{(k)}$ denote full batch Hessian at $k$-th minimum, such that $\mathbb{E}(H_f^{(k)}(\xi)) = H_f^{(k)}$, and denote $n_t = H_f^{(k)}\left(W_t - \bar{W}^{(k)}\right) - H_f^{(k)}(\xi)\left(W_t - \bar{W}^{(k)}\right)$, which indicates the bias. The uncertainty brought by stochastic gradient method is measured as follows,

$$
\begin{aligned}
W_{t+1} - \bar{W}^{(k)} &= W_t - \bar{W}^{(k)} - \eta_t H_f^{(k)}(\xi)\left(W_t - \bar{W}^{(k)}\right) \\
&= W_t - \bar{W}^{(k)} - \eta_t H_f^{(k)}\left(W_t - \bar{W}^{(k)}\right) \\
&\quad + \eta_t H_f^{(k)}\left(W_t - \bar{W}^{(k)}\right) - \eta_t H_f^{(k)}(\xi)\left(W_t - \bar{W}^{(k)}\right) \\
&= (I - \eta_t H_f^{(k)})\left(W_t - \bar{W}^{(k)}\right) + \eta_t n_t \\
&= (I - \eta_t H_f^{(k)})(I - \eta_{t-1}H_f^{(k)})\cdots(I - \eta_{t-q}H_f^{(k)})(W_{t-q} - \bar{W}^{(k)}) \\
&\quad + \sum_{i=t-q}^{t}(I - \eta_t H_f^{(k)})(I - \eta_{t-1}H_f^{(k)})\cdots(I - \eta_{i+1}H_f^{(k)})\eta_i n_i \\
&:= P_t^{(k)}P_{t-1}^{(k)}\cdots P_{t-q}^{(k)}(W_{t-q} - \bar{W}^{(k)}) + \sum_{i=t-q}^{t}P_t^{(k)}P_{t-1}^{(k)}\cdots P_{i+1}^{(k)}\eta_i n_i,
\end{aligned}
\tag{35}
$$

where $P_i^{(k)} := (I - \eta_i H_f^{(k)})$.

According to the approximation approach of the objective function (1) in Section 3.1 , i.e. $\mathcal{L}(f(W,\xi)) \approx \mathcal{L}(f(\bar{W}^{(k)},\xi)) + \frac{1}{2}(W - \bar{W}^{(k)})^\top H_f^{(k)}(\xi)(W - \bar{W}^{(k)})$, the loss uncertainty introduced by stochastic gradient method within the interval $t \in [1 + T_k, T_{k+1}]$ is quantified as $\mathbb{E}[(W_{T_{k+1}} - \bar{W}^{(k)})^\top H_f^{(k)}\left(W_{T_{k+1}} - \bar{W}^{(k)}\right)]$, where the loss function is approximated by leveraging the full batch Hessian.

$$
\begin{aligned}
&\mathbb{E}\left[\mathcal{L}(f(W_t,\xi)) - \mathcal{L}(f(\bar{W}^{(k)},\xi))\right] = \mathbb{E}\left[(W_t - \bar{W}^{(k)})^\top H_f^{(k)}(W_t - \bar{W}^{(k)})\right] \\
&= \mathbb{E}\left[\left(P_t^{(k)}P_{t-1}^{(k)}\cdots P_{1+T_k}^{(k)}(W_{1+T_k} - \bar{W}^{(k)}) + \sum_{i=1+T_k}^{t}P_t^{(k)}P_{T_{k+1}-1}^{(k)}\cdots P_{i+1}^{(k)}\eta_i n_i\right)^\top\right. \\
&\quad H_f^{(k)}\left.\left(P_t^{(k)}P_{t-1}^{(k)}\cdots P_{1+T_k}^{(k)}(W_{1+T_k} - \bar{W}^{(k)}) + \sum_{i=1+T_k}^{t}P_t^{(k)}P_{t-1}^{(k)}\cdots P_{i+1}^{(k)}\eta_i n_i\right)\right] \\
&= \mathbb{E}\left[(W_t - \bar{W}^{(k)})^\top P_{1+T_k}^{(k)}\cdots P_t^{(k)}H_f^{(k)}P_t^{(k)}\cdots P_{1+T_k}^{(k)}(W_t - \bar{W}^{(k)})\right] \\
&\quad + 2\mathbb{E}\left[(W_t - \bar{W}^{(k)})^\top P_{1+T_k}^{(k)}\cdots P_t^{(k)}H_f^{(k)}\left(\sum_{i=1+T_k}^{t}P_t^{(k)}\cdots P_{i+1}^{(k)}\eta_i n_i\right)\right] \\
&\quad + \mathbb{E}\left(\sum_{i=1+T_k}^{t}P_t^{(k)}\cdots P_{i+1}^{(k)}\eta_i n_i\right)^\top H_f^{(k)}\left(\sum_{i=1+T_k}^{t}P_t^{(k)}\cdots P_{i+1}^{(k)}\eta_i n_i\right).
\end{aligned}
\tag{36}
$$

Since $\mathbb{E}[n_t] = 0$, $\mathbb{E}\left[(W_t - \bar{W}^{(k)})^\top P_{1+T_k}^{(k)}\cdots P_t^{(k)}H_f^{(k)}\left(\sum_{i=1+T_k}^{t}P_t^{(k)}\cdots P_{i+1}^{(k)}\eta_i n_i\right)\right] = 0$. Meanwhile, we notice the data samplings between different iteration are independent, thus $\mathbb{E}\left[(n_i)^\top n_j\right] =$

$0 \quad (i \neq j)$. We obtain

$$
\mathbb{E}\left(\sum_{i=1+T_k}^{t} P_t^{(k)} \cdots P_{i+1}^{(k)} \eta_i n_i\right)^\top H_f^{(k)} \left(\sum_{i=1+T_k}^{t} P_t^{(k)} \cdots P_{i+1}^{(k)} \eta_i n_i\right)
$$

$$
= \mathbb{E}\left(\sum_{i,j=1+T_k}^{t} \eta_i n_i^\top P_{i+1}^{(k)} \cdots P_t^{(k)} H_f^{(k)} P_t^{(k)} \cdots P_{j+1}^{(k)} \eta_j n_j\right) \tag{37}
$$

$$
= \sum_{i=1+T_k}^{t} \mathbb{E}\left(\eta_i n_i^\top P_{i+1}^{(k)} \cdots P_t^{(k)} H_f^{(k)} P_t^{(k)} \cdots P_{i+1}^{(k)} \eta_i n_i\right).
$$

Therefore, Eq.(36) can be reformulated as follows,

$$
\mathbb{E}\left(\sum_{i=1+T_k}^{t} P_t^{(k)} \cdots P_{i+1}^{(k)} \eta_i n_i\right)^\top H_f^{(k)} \left(\sum_{i=1+T_k}^{t} P_t^{(k)} \cdots P_{i+1}^{(k)} \eta_i n_i\right)
$$

$$
= \mathbb{E}\left[(W_t - \bar{W}^{(k)})^\top P_{1+T_k}^{(k)} \cdots P_t^{(k)} H_f^{(k)} P_t^{(k)} \cdots P_{1+T_k}^{(k)} (W_t - \bar{W}^{(k)})\right] \tag{38}
$$

$$
+ \sum_{i=1+T_k}^{t} \mathbb{E}\left(\eta_i n_i^\top P_{i+1}^{(k)} \cdots P_t^{(k)} H_f^{(k)} P_t^{(k)} \cdots P_{i+1}^{(k)} \eta_i n_i\right)
$$

$$
:= B + V,
$$

where we denote $B := \mathbb{E}\left[(W_t - \bar{W}^{(k)})^\top P_{1+T_k}^{(k)} \cdots P_t^{(k)} H_f^{(k)} P_t^{(k)} \cdots P_{1+T_k}^{(k)} (W_t - \bar{W}^{(k)})\right]$ bias

term and $V := \sum\limits_{i=1+T_k}^{t} \mathbb{E}\left(\eta_i n_i^\top P_{i+1}^{(k)} \cdots P_t^{(k)} H_f^{(k)} P_t^{(k)} \cdots P_{i+1}^{(k)} \eta_i n_i\right)$ variance term.

Denote $u_i^{(k)} \quad i \in \{1, 2, \cdots, N\}$ are linearly independent eigenvectors, which can form the basis for the $N$ dimension vector space. Then we have $H_f^{(k)}(\xi) u_i^{(k)} = \lambda_i^{(k)} u_i^{(k)} \quad (i = 1, 2, \cdots, N)$ and the initial deviation from the optimal solution can be expressed as $W_t - \bar{W}^{(k)} = \sum\limits_{i=1}^{N} s_i^{(k)} u_i^{(k)}$ where $s_i^{(k)}$ are the coefficients corresponding to the eigenvector components. Combined with Lemma 1, we obtain

**Case 1 ($\varphi > 2$):**

$$
B = (W_t - \bar{W}^{(k)})^\top P_{1+T_k}^{(k)} \cdots P_t^{(k)} H_f^{(k)} P_t^{(k)} \cdots P_{1+T_k}^{(k)} (W_t - \bar{W}^{(k)})
$$

$$
= \left[(I - \eta_t H_f^{(k)}) \cdots (I - \eta_{1+T_k} H_f^{(k)})(W_{1+T_k} - \bar{W}^{(k)})\right]^\top H_f^{(k)}
$$

$$
\cdot (I - \eta_t H_f^{(k)}) \cdots (I - \eta_{1+T_k} H_f^{(k)})(W_{1+T_k} - \bar{W}^{(k)})
$$

$$
= \sum_{i=1}^{N} (s_i^{(k)})^2 \|u_i^{(k)}\|_2^2 \lambda_i^{(k)} \prod_{j=1+T_k}^{t} \left[(1 - \eta_j \lambda_i^{(k)})\right]^2 \tag{39}
$$

$$
\leq \sum_{i=1}^{N} (s_i^{(k)})^2 \|u_i^{(k)}\|_2^2 \lambda_i^{(k)} \exp\left(-2\lambda_i^{(k)} \eta_{\min}(t - T_k)\right)
$$

$$
\cdot \left(\frac{4(T_{k+1} - T_k) + (\varphi - 2)\pi}{4(T_{k+1} - T_k) + (\varphi - 2)\pi(t - T_k)}\right)^\tau
$$

$$
\leq \left(\frac{4(T_{k+1} - T_k) + (\varphi - 2)\pi}{4(T_{k+1} - T_k) + (\varphi - 2)\pi(t - T_k)}\right)^\tau
$$

$$
\cdot \exp\left(-2\lambda_i^{(k)} \eta_{\min}(t - T_k)\right) \lambda_u^{(k)} \|W_{1+T_k} - \bar{W}^{(k)}\|_2^2.
$$

In addition,

$$
\begin{aligned}
V &= \sum_{i=1+T_k}^{t} \mathbb{E}\left(\eta_i n_i^\top P_{i+1}^{(k)} \cdots P_t^{(k)} H_f^{(k)} P_t^{(k)} \cdots P_{i+1}^{(k)} \eta_i n_i\right)\\
&= \sum_{i=1+T_k}^{t} \eta_i^2 \mathbb{E}\left[\operatorname{tr}\left(n_i^\top P_{i+1}^{(k)} \cdots P_t^{(k)} H_f^{(k)} P_t^{(k)} \cdots P_{i+1}^{(k)} n_i\right)\right]\\
&= \sum_{i=1+T_k}^{t} \eta_i^2 \mathbb{E}\left[\operatorname{tr}\left(P_{i+1}^{(k)} \cdots P_t^{(k)} H_f^{(k)} P_t^{(k)} \cdots P_{i+1}^{(k)} n_i n_i^\top\right)\right]\\
&= \sum_{i=1+T_k}^{t} \eta_i^2 \operatorname{tr}\left[P_{i+1}^{(k)} \cdots P_t^{(k)} H_f^{(k)} P_t^{(k)} \cdots P_{i+1}^{(k)} \mathbb{E}\left(n_i n_i^\top\right)\right]\\
&\le \sigma^2 \sum_{i=1+T_k}^{t} \eta_i^2 \operatorname{tr}\left[P_{i+1}^{(k)} \cdots P_t^{(k)} H_f^{(k)} P_t^{(k)} \cdots P_{i+1}^{(k)} H_f^{(k)}\right]\\
&= \sigma^2 \sum_{i=1+T_k}^{t} \eta_i^2 \sum_{j=1}^{N} (\lambda_j^{(k)})^2 \prod_{q=i+1}^{t} \left(1 - \eta_q \lambda_j^{(k)}\right)^2\\
&\le \sigma^2 \sum_{i=1+T_k}^{t} \eta_i^2 \sum_{j=1}^{N} (\lambda_j^{(k)})^2 \exp\left(-2\lambda_j^{(k)} \eta_{\min}(t-i)\right)\\
&\quad \cdot \left(\frac{4(T_{k+1}-T_k) + (\varphi-2)\pi(i-T_k)}{4(T_{k+1}-T_k) + (\varphi-2)\pi(t-T_k)}\right)^\tau,
\end{aligned}
\tag{40}
$$

where the second equality comes from cyclic property of trace and the first inequality comes from $\mathbb{E}_\xi\left[n_t n_t^\top\right] \preceq \sigma^2 H_f^{(k)}$.

**Case 2 ($\varphi < 2$):**

$$
\begin{aligned}
B &= (W_t - \bar{W}^{(k)})^\top P_{1+T_k}^{(k)} \cdots P_t^{(k)} H_f^{(k)} P_t^{(k)} \cdots P_{1+T_k}^{(k)} (W_t - \bar{W}^{(k)})\\
&= \left[(I - \eta_t H_f^{(k)}) \cdots (I - \eta_{1+T_k} H_f^{(k)})(W_{1+T_k} - \bar{W}^{(k)})\right]^\top\\
&\quad \cdot H_f^{(k)} (I - \eta_t H_f^{(k)}) \cdots (I - \eta_{1+T_k} H_f^{(k)})(W_{1+T_k} - \bar{W}^{(k)})\\
&= \sum_{i=1}^{N} (s_i^{(k)})^2 \|u_i^{(k)}\|_2^2 \lambda_i^{(k)} \prod_{j=1+T_k}^{t} \left[(1 - \eta_j \lambda_i^{(k)})\right]^2\\
&\le \sum_{i=1}^{N} (s_i^{(k)})^2 \|u_i^{(k)}\|_2^2 \lambda_i^{(k)} \exp\left(-2\lambda_i^{(k)} \eta_{\min}(t-T_k)\right)\\
&\quad \cdot \left(\frac{(2\varphi + 2\pi - \varphi\pi) - \frac{(2-\varphi)\pi}{(T_{k+1}-T_k)}(t - T_k - 0.5)}{(2\varphi + 2\pi - \varphi\pi) + \frac{(2-\varphi)\pi}{2(T_{k+1}-T_k)}}\right)^{-\tau}\\
&\le \left(\frac{(2\varphi + 2\pi - \varphi\pi) - \frac{(2-\varphi)\pi}{(T_{k+1}-T_k)}(t - T_k - 0.5)}{(2\varphi + 2\pi - \varphi\pi) + \frac{(2-\varphi)\pi}{2(T_{k+1}-T_k)}}\right)^{-\tau}\\
&\quad \cdot \exp\left(-2\lambda_i^{(k)} \eta_{\min}(t-T_k)\right) \lambda_u^{(k)} \|W_{1+T_k} - \bar{W}^{(k)}\|_2^2.
\end{aligned}
\tag{41}
$$

In addition,

$$
\begin{aligned}
V &= \sum_{i=1+T_k}^{t} \mathbb{E}\left( \eta_i n_i^\top P_{i+1}^{(k)} \cdots P_t^{(k)} H_f^{(k)} P_t^{(k)} \cdots P_{i+1}^{(k)} \eta_i n_i \right) \\
&= \sum_{i=1+T_k}^{t} \eta_i^2 \mathbb{E}\left[ \mathrm{tr}\left( n_i^\top P_{i+1}^{(k)} \cdots P_t^{(k)} H_f^{(k)} P_t^{(k)} \cdots P_{i+1}^{(k)} n_i \right) \right] \\
&= \sum_{i=1+T_k}^{t} \eta_i^2 \mathbb{E}\left[ \mathrm{tr}\left( P_{i+1}^{(k)} \cdots P_t^{(k)} H_f^{(k)} P_t^{(k)} \cdots P_{i+1}^{(k)} n_i n_i^\top \right) \right] \\
&= \sum_{i=1+T_k}^{t} \eta_i^2 \mathrm{tr}\left[ P_{i+1}^{(k)} \cdots P_t^{(k)} H_f^{(k)} P_t^{(k)} \cdots P_{i+1}^{(k)} \mathbb{E}\left( n_i n_i^\top \right) \right] \\
&\leq \sigma^2 \sum_{i=1+T_k}^{t} \eta_i^2 \mathrm{tr}\left[ P_{i+1}^{(k)} \cdots P_t^{(k)} H_f^{(k)} P_t^{(k)} \cdots P_{i+1}^{(k)} H_f^{(k)} \right] \\
&= \sigma^2 \sum_{i=1+T_k}^{t} \eta_i^2 \sum_{j=1}^{N} (\lambda_j^{(k)})^2 \prod_{q=i+1}^{t} \left( 1 - \eta_q \lambda_j^{(k)} \right)^2 \\
&\leq \sigma^2 \sum_{i=1+T_k}^{t} \eta_i^2 \sum_{j=1}^{N} (\lambda_j^{(k)})^2 \exp\left( -2\lambda_j^{(k)} \eta_{\min}(t-i) \right) \\
&\quad \cdot \left( \frac{(2\varphi + 2\pi - \varphi\pi) - \frac{(2-\varphi)\pi}{(T_{k+1}-T_k)}(t - T_k - 0.5)}{(2\varphi + 2\pi - \varphi\pi) - \frac{(2-\varphi)\pi}{(T_{k+1}-T_k)}(i - T_k - 0.5)} \right)^{-\tau},
\end{aligned}
\tag{42}
$$

where the second equality comes from cyclic property of trace and the first inequality comes from $\mathbb{E}_\xi\left[ n_t n_t^\top \right] \preceq \sigma^2 H_f^{(k)}$.

$\square$

In this proof, we employ assumption $\mathbb{E}_\xi\left[ n_t n_t^\top \right] \preceq \sigma^2 H_f^{(k)}$ used in the work [16, 36]. Moreover, we acknowledge that the idea of the proof is inspired by the approach introduced in [16, 36]

# E Experimental details

## E.1 Baselines

We detail the baselines as follows,

- **Step schedule(SS)** reduces the learning rate in a piecewise manner. It implements discrete learning rate drops at predefined epoch intervals following schedule $g(t) = d_{\lfloor t/t_s \rfloor}$, where $t_s$ is the predefined decaying step and $d_i(d_i \geq d_{i+1})$ is predefined decaying scalar [16]. A typical implementation decreases the learning rate by a factor of 0.1 after 50% of the epochs and further by a factor of 0.01 after 75% of the epochs [21]. However, it requires careful tuning of step timing since abrupt changes may destabilize optimization. In this paper, we employ such setting of step schedule ($\times$ 0.1 after 50% and $\times$ 0.01 after 75%) for all our experiments.

- **Cosine schedule(CS)** controls the $t$-th iteration learning rate $\eta_t$ following the mathematical formulation as $\eta_t = \eta_{min} + \frac{1}{2}(\eta_0 - \eta_{min})(1 + \cos(\frac{t\pi}{T}))$ where $T$ is total iterations. It provides smooth transition between learning rates, shown to improve final model accuracy step schedule. Moreover, it is proposed to mitigate abrupt decreases in the learning rate that could hinder models from escaping local minima. This approach has demonstrated effectiveness, particularly in transformer training [32]. Besides, it includes restart mechanisms where $T$ becomes the cyclic period.

- **Cyclical Learning Rates (CLR)** oscillates between base learning rate $\eta_{min}$ and maximal learning rate $\eta_{max}$ with triangular and triangular2 policies. The cyclic property help traverse loss landscape barrier, so that it enables neural networks to train significantly faster, often by an order of magnitude, compared to standard training methods [43, 44]. Besides, the commonly-used OneCycle learning rate schedule can be seen as the one period version of Cyclical Learning Rates. In our experiments, this baseline **CLR** includes Cyclical and OneCycle schedules(**1C**). For CLR, we set linear mode for each period and period is two.

- **Budgeted training(BT)** is conducted by the proposed budget-aware setting of existing learning rate schedule under epoch budget [30]. It reformulates standard schedules such as linear, step and cosine schedule as functions of remaining budget. The classic linear decay version is formulated as $\eta_t = \eta_0 \times (1 - t/T)$ where $T$ is total iterations. It is particularly effective for hyper-parameter tuning.

- **Reflected Exponential (REX)** controls the $t$-th iteration learning rate $\eta_t$ following the mathematical formulation as $\eta_t = \eta_0 \left( \frac{1 - t/T}{0.5 + 0.5 \times (1 - t/T)} \right)$. It is inspired by the performance of delayed linear schedules, aggressively reducing the learning rate near the end of training and reflecting an exponential decay [5].

## E.2 Evaluation benchmarks of language task

We evaluate models on a diverse set of open benchmarks. The benchmarks are listed in Table 5. For language models evaluation, we adopt the [15] for standardized performance comparisons.

## E.3 Implementation Details

Consistent with CIFAR, ImageNet and OLMo practices, we use step-wise scheduling for all training procedures.

**Vision tasks on CIFAR** CIFAR10/100 are simple benchmark datasets that are widely used for quick and efficient evaluation of deep learning tasks. CIFAR10 consists of a training split of 50000 examples and a test split of 10000 examples. Each example is a 32×32 color image in 10 classes (airplane, automobile, bird, cat, deer, dog, frog, horse, ship, truck). CIFAR100 comprises 60000 32×32 pixels color images which are divided into 100 classes. Each class contains 600 images and the classes are grouped into 20 super-classes. Each image comes with a 'fine' label (the class to which it belongs) and a 'coarse' label (the super-class to which it belongs). We use the `torch.optim.SGD` and `torch.optim.AdamW` API to configure the optimizer. For vision tasks in Section 4.1, we all adopt the SGD optimizer. For ablation studies in Section 4.3, we adopt the AdamW optimizer. We

Table 5: Evaluation benchmarks of language task

| Datasets name | abbreviation |
|---|---|
| ARC-Easy [10] | ARC-E |
| ARC-Challenge [10] | ARC-C |
| HellaSwag [55] | HSWAG |
| PIQA [3] | PIQA |
| WinoGrande [39] | WG |
| OpenbookQA [35] | OBQA |
| BoolQ [9] | BoolQ |
| SciQ [50] | SciQ |
| COPA [18] | COPA |
| CommonsenseQA [46] | CSQA |
| SocialIQA [40] | SIQA |
| MMLU stem [22] | STEM |
| MMLU humanities [22] | HUM |
| MMLU social sci [22] | SOC |
| MMLU other [22] | OTH |

initialize the learning rate to 1e-1, set the batch size to 128. Each epoch consists of 50000/128 + 1 = 391 steps. We leave the weight decay value as 5e-4, and complete the training task on PyTorch 2.4.1+cu118 with RTX 5880 and RTX 3090.

**Vision tasks on ImageNet** To evaluate the scalability of our schedule on larger-scale dataset, we conduct experiments on the ImageNet dataset, a dataset that more closely reflects real-world visual recognition challenges. ImageNet consists of approximately 1.28 million training images and 50000 validation images across 1000 classes, which also come with an official dataset split. This benchmark, particularly with the ResNet-50 architecture, has been extensively studied and is widely regarded as a standard evaluation setup. We adopt the common ResNet50 architecture, and separately train it with different learning rate schedulers. To provide a comprehensive evaluation, we conduct experiments using both the classical SGD optimizer and the popularized AdamW optimizer. For the experiments with SGD, we set the peak learning rate to 0.5 and apply the weight decay factor of 1e-4. For the experiments using AdamW, the peak learning rate is set to 3e-3, with a decoupled weight decay of 0.02. Following default configuration which requires learning rate warmup, we integrate 10% warmup phases of the total training tokens into all schedules. For schedules that inherently include a warmup-like behavior, such as the 1C schedule, where the learning rate initially rises from zero to a target value, we retain their original settings during the ascending phase. We set the batch size to 2048 and complete these training tasks with 8 NVIDIA A100 GPUs.

**Language tasks** For language tasks, we use the OLMo model. OLMo [19] is a decoder-only transformer-based language model that builds upon the vanilla Transformer architecture [49]. It incorporates several key improvements inspired by recent large language models (LLMs) such as PaLM [8], the LLaMA family [48], OpenLM and Falcon [1]. Notably, OLMo stands out as a truly open language model, offering full transparency through its open training data, released training/evaluation code, intermediate model checkpoints, and comprehensive training logs, making it a valuable resource for reproducibility and further research.

For the training hyper-parameters on OLMo, we primarily adopt the configuration outlined in OLMo [19]. We conduct experiment on H100 and A100. Following default configuration which requires learning rate warmup, we integrate 10% warmup phases of the total training tokens into all schedules. For schedules that inherently include a warmup-like behavior, such as the 1C schedule, where the learning rate initially rises from zero to a target value, we retain their original settings during the ascending phase. Besides, large language models commonly provide multiple scales to accommodate different compute constraints. We adjust both model size and training steps to explore budgets. To ensure robust conclusions, we maintain a data-to-parameters ratio (D/N) of 500 across scales. The specific configurations are as follows, 36M-parameter model is trained on 50 billion tokens, 73M-parameter model is trained on 50 billion tokens, 151M-parameter model is trained on 75 billion tokens, 300M-parameter model is trained on 150 billion tokens. This scaling strategy allows us

Table 6: Configurations of OLMo

| Model scale | 36M | 73M | 151M | 300M |
|---|---|---|---|---|
| token number | 50B | 50B | 75B | 150B |
| Layers ($L$) | 12 | 16 | 20 | 24 |
| Hidden dim ($D$) | 512 | 640 | 768 | 1024 |
| Attention heads | 8 | 10 | 8 | 8 |
| Inner dim ($H$) | 1280 | 1536 | 2240 | 2688 |
| Vocab size ($V$) | | 100,352 | | |

to systematically study the relationship between model size, training data, and performance while controlling for the D/N ratio. The OLMo model is trained using AdamW optimizer. The detailed configuration are shown in Table 6.

Our implementation strictly follows the PyTorch official API, i.e. it is inherited from the base class

```
torch.optim.lr_scheduler.LRScheduler
```

. It maintains full compatibility with all

```
torch.optim
```

optimizers. The complete production-ready code is publicly available.

The code of UBA schedule is available at `https://github.com/Ttt-answer/UBA.git`.

### E.4 Overall experiment results for Vision Classification Tasks

Overall results for the vision classification tasks are presented in Table 7, depicting validation accuracy on vision benchmarks (e.g., CIFAR10/100 and ImageNet) using different architectures ( ResNet18, ResNet34, ResNet50). We conduct several independent runs for error bars. Due to computational constraints, we use two independent experimental results for ImageNet-ResNet50. It is obvious that our proposed UBA schedule outperform baselines, achieving the highest validation accuracy over both small-scale and large-scale benchmarks across all training budgets.

Our analysis reveals the following key findings. (i) On CIFAR10-ResNet18 and CIFAR100-ResNet34 task, UBA shows the strongest performance across all training budgets. On the large-scale ImageNet benchmark with ResNet50, UBA maintains consistent superiority. The consistent improvements on both small-scale and large-scale benchmarks highlight UBA's **generalizability across scales and data regimes**. (ii) UBA outperforms baselines not only at 100% training budget but also at 25% and 50% iteration budgets, demonstrating its **budget efficiency**, i.e.,effectiveness in computation-constrained scenarios. (iii)Notably, UBA shows robust performance even while the second-best schedule varies depending on both the datasets, architectures and training budgets. This instability among competing schedules suggests their performance is highly sensitive to datasets-architectures characteristics and training dynamics at different learning phases. Therefore, for practitioners seeking reliable schedules without extensive method selection, UBA provides a default-strong choice. UBA eliminates the need for case-by-case baseline comparison and delivers **stable superiority and reliability** regardless of datasets, architectures, training budgets and scales.

### E.5 Overall experiment results for language models

For most tasks, we conduct a comprehensive evaluation across six representative baselines to ensure broad coverage. For large model with 300M parameters, we focus our comparison on the top three performing baselines due to the computational tractability.

The overall performance of the OLMo models is presented in Table 8. We also plot training curves in Figures 7,8, 9 and 10, showing training and validation losses along with downstream evaluation results across model sizes ranging from 36M to 300M parameters and training data scales from 50B to 150B tokens. As shown, UBA schedule consistently achieves lower training and validation losses while delivering superior downstream performance.

Table 7: Generalization accuracy.

| Dataset | Network | Method | training budget (epoch(%)) | | |
|---------|---------|--------|---------------------|-----------|------------|
| | | | 75 (25%) | 150(50%) | 300(100%) |
| CIFAR10 | ResNet18 | SS | 92.71±0.4243 | 93.45±0.4822 | 93.61±0.3035 |
| | | CS | 94.21±0.5116 | 94.24±0.2419 | 95.52±0.2433 |
| | | CLR | 92.13±0.2610 | 92.99±0.5459 | 94.92±0.3811 |
| | | 1C | 94.23±0.1838 | 95.03±0.1353 | 95.35±0.2984 |
| | | BT | 94.18±0.3546 | 94.79±0.4605 | **95.68±0.1168** |
| | | REX | 94.49±0.3225 | 95.10±0.1929 | 95.45±0.1350 |
| | | UBA(ours) | **94.57±0.3281** | **95.26±0.2150** | 95.65±0.1589 |
| | | | 75 (25%) | 150(50%) | 300(100%) |
| CIFAR100 | ResNet34 | SS | 71.09±0.2676 | 75.34±0.1802 | 77.24±0.1808 |
| | | CS | 64.12±0.1808 | 69.84±0.7580 | 74.29±0.5958 |
| | | CLR | 73.33±0.5635 | 73.41±0.2835 | 74.74±0.2266 |
| | | 1C | 73.38±0.3227 | 75.04±0.6382 | 77.86±0.2239 |
| | | BT | 72.75±0.2611 | 75.35±0.2423 | 78.58±0.2705 |
| | | REX | 73.64 ±0.5635 | 75.14±0.3110 | 78.04±0.1430 |
| | | UBA(ours) | **74.60±0.1508** | **76.63±0.2678** | **78.95±0.2818** |
| | | | 75 (25%) | 150(50%) | 300(100%) |
| ImageNet | ResNet50 | SS | 74.74±0.1386 | 77.24±0.3861 | 78.56±0.6435 |
| | | CS | 75.84±0.2220 | 77.95±0.2164 | 79.07±0.0438 |
| | | CLR | 73.90±1.4609 | 76.04±1.6419 | 77.84±1.3053 |
| | | 1C | 75.83±0.7792 | 77.74±0.5218 | 78.90±0.2956 |
| | | BT | 75.90 ±0.2786 | 77.81±0.4624 | 78.86± 0.2814 |
| | | REX | 75.85±0.8047 | 77.66 ±0.8853 | 78.66 ±0.2828 |
| | | UBA(ours) | **76.29±0.4031** | **78.26± 0.1640** | **79.39±0.0170** |

From Table 8, UBA achieves state-of-the-art performance on approximately 50% of the benchmarks across all scales while the second-best schedule varies. Furthermore, it achieves *consistent superior average performance* among all baselines. It highlights the stable superiority and reliability of UBA, providing a *default-strong choice*. Moreover, it demonstrates significant improvements in SciQ-73M(+1.7) and ARC-E-300M(+2.63), highlighting its ability to enhance generalization across diverse benchmarks. Besides, in Figure 2, UBA consistently achieves lower training loss and validation loss throughout the training, indicating the *efficient training ability* and downstream performance enhancement. Notably, while task-specific tuning of $\varphi$ can further improve model performance, we intentionally fix its value across all tasks and architectures to ensure fair evaluation. Remarkably, even with this universal $\varphi$ setting, UBA consistently outperforms baselines on diverse benchmarks, demonstrating *inherent robustness* to varying benchmarks and model scales.

### E.6 Performance across different optimizers

Our scheduling strategy originates from the solution to a optimization problem under gradient descent dynamics. While modern optimizer (e.g., AdamW [33]) introduce additional momentum and adaptive mechanisms, they maintain the fundamental property of performing gradient-based updates. To verify the performance of schedule, we conduct cross-optimizer ablation studies. We evaluate the model on CIFAR100 and ImageNet datasets using AdamW optimizer. For AdamW, we use a learning rate of 0.003 and weight decay of 0.0001. We report the validation accuracy for each configuration. The results are summarized in Table 9.

As shown in Table 9, our UBA schedule achieves state-of-the-art validation accuracy on both SGD and AdamW optimizers, consistently outperforming baselines across CIFAR100-ResNet34 and ImageNet-ResNet50 benchmarks. This demonstrates that although our schedule is theoretically derived from standard gradient descent dynamics, it generalizes effectively to modern optimizers like AdamW — despite their additional momentum and adaptive mechanisms — highlighting its broad applicability.

Table 8: Performance comparison on OLMo model

| Size | Sched. | PIQA | HSWAG | OBQA | SciQ | ARC-E | ARC-C | COPA | SIQA | SOC | OTH | Avg. |
|---|---|---|---|---|---|---|---|---|---|---|---|---|
| | | | | | | Benchmark(accuracy %) | | | | | | |
| 36M | CS | 59.74 | 27.36 | 26.80 | 66.50 | 44.74 | 21.74 | 60.00 | 40.58 | 24.33 | 29.09 | 40.09 |
| | BT | 59.41 | 27.33 | 27.40 | 67.20 | 43.68 | 21.40 | 59.00 | 39.82 | 23.48 | 29.13 | 39.79 |
| | SS | 60.12 | 27.63 | 27.20 | 65.30 | 45.26 | 22.41 | 63.00 | 40.89 | 23.87 | 29.62 | 40.53 |
| | REX | 60.17 | 27.82 | **27.80** | 68.10 | 45.09 | 22.41 | 62.00 | 40.63 | **25.18** | 30.02 | 40.92 |
| | CLR | **61.15** | 27.21 | 27.60 | 67.50 | 42.11 | 22.07 | 62.00 | **41.45** | 24.29 | 29.38 | 40.48 |
| | 1C | 60.17 | 27.91 | 25.80 | 67.40 | 43.68 | **23.41** | 55.00 | 40.02 | 22.26 | 29.86 | 39.55 |
| | UBA | 60.39 | **27.98** | 27.40 | **68.20** | **45.79** | 21.74 | **63.00** | 40.69 | 24.33 | **30.14** | **40.97** |
| 73M | CS | 63.06 | 29.68 | 28.80 | 72.60 | 45.09 | 25.08 | 65.00 | 41.56 | 26.24 | 28.17 | 42.53 |
| | BT | 61.70 | 29.79 | 28.00 | 71.40 | 47.54 | 23.41 | **66.00** | **41.97** | 25.05 | 26.48 | 42.13 |
| | SS | 61.53 | 29.30 | 28.20 | 69.40 | 47.19 | 24.41 | 65.00 | 41.76 | 26.32 | 28.33 | 42.14 |
| | REX | 62.46 | **30.22** | 27.60 | 72.20 | 45.09 | 26.09 | 65.00 | 41.81 | 25.52 | **29.34** | 42.53 |
| | CLR | 62.30 | 29.38 | 27.80 | 70.40 | 45.44 | 24.08 | 62.00 | 42.02 | 25.05 | 29.05 | 41.75 |
| | 1C | 61.81 | 29.90 | 27.80 | 71.10 | **47.54** | **26.42** | 63.00 | 40.79 | 24.37 | 28.81 | 42.15 |
| | UBA | **63.17** | 30.09 | **28.80** | **74.30** | 45.79 | 22.74 | 65.00 | 41.45 | **27.13** | 28.85 | **42.73** |
| 150M | CS | **66.00** | 35.19 | 32.00 | 76.60 | 49.82 | 25.42 | 67.00 | 42.84 | 26.24 | 32.27 | 45.34 |
| | BT | 64.85 | 34.95 | 29.80 | 78.20 | 48.95 | 26.42 | 67.00 | 42.99 | 26.83 | 31.87 | 45.19 |
| | SS | 65.45 | 34.76 | 30.60 | 77.10 | 49.65 | 24.75 | 68.00 | 42.37 | 26.58 | 31.59 | 45.09 |
| | REX | 65.56 | **35.72** | 29.40 | 78.30 | 52.46 | 25.08 | **69.00** | 43.30 | 27.68 | 32.64 | 45.91 |
| | CLR | 65.07 | 34.76 | 31.60 | 77.20 | 50.70 | 25.08 | 68.00 | **43.71** | 27.30 | 31.23 | 45.47 |
| | 1C | 65.29 | 35.16 | **32.80** | 76.30 | **53.16** | 25.75 | 67.00 | 43.45 | 27.64 | 30.62 | 45.72 |
| | UBA | 65.23 | 35.50 | 29.80 | **78.30** | 50.35 | **27.42** | 67.00 | 43.50 | **29.29** | 32.72 | **45.91** |
| 300M | CS | 68.88 | 45.32 | 33.40 | 83.20 | 57.19 | 27.76 | 69.00 | **44.58** | 29.25 | 34.57 | 49.32 |
| | BT | 69.64 | 45.21 | 30.80 | 83.70 | 57.37 | 26.42 | 72.00 | 43.96 | 28.66 | 35.49 | 49.33 |
| | REX | **70.18** | 46.30 | 33.00 | **84.40** | 57.72 | 28.43 | 67.00 | 43.71 | **29.50** | **36.74** | 49.70 |
| | UBA | 69.48 | **46.44** | **34.60** | 83.90 | **60.35** | **29.10** | **72.00** | 44.42 | 28.53 | 35.41 | **50.42** |

## E.7 Parameter analysis of $\varphi$

In this subsection, we perform a sensitivity analysis of the key parameter $\varphi$ in equation (5) , which controls the variation speed of learning rate. Our experiments systematically evaluate $\varphi \in \{0.25, 0.5, 1.0, 2.5, 5, 10\}$ across different optimization scenarios. The results are plotted in Figure 11 and listed in Table 10).

**Observations** From the experimental results, we find an interesting dichotomy of $\varphi$. Phenomenon(a): By choose proper $\varphi$, UBA demonstrates consistent performance gains over all baselines across the full spectrum of training budgets (25%, 50% and 100%), irrespective of optimizer choice (SGD/AdamW). This suggests the effectiveness of UBA is orthogonal to specific optimization strategies. Besides, with optimizer changed, UBA maintains comparable accuracy on the same dataset-architecture, which shows that UBA can exhaustively exploit the model's capacity regardless of optimizer variants. Phenomenon(b): As shown in Figure 11 and Table 10, on CIFAR100 dataset, the best performance for AdamW is achieved with $\varphi = 1$, whereas SGD performs optimally with $\varphi = 2.5$. On ImageNet dataset, the best performance for AdamW is achieved with $\varphi = 0.5$, whereas SGD performs optimally with $\varphi = 10$. Our experiments reveal a notable trend: when using AdamW, a smaller $\varphi$ yields better performance, whereas a larger $\varphi$ is preferred for SGD.

**Analysis** Furthermore, we intend to explain the dual behavior of $\varphi$ through theoretical analyses (Proposition 1 and Theorem 1). We attribute Phenomenon (a) to UBA's optimal scheduling dynamics, governed by the $\varphi$ value. Moreover, Phenomenon (b) reveals a fundamental dichotomy in $\varphi$ value selection between optimizers. We attribute this phenomenon(b) to the preconditioning effect of AdamW. Unlike SGD, AdamW adapts the learning rate by scaling gradients with the square root of the second moment estimate $\sqrt{v_t}$. In regions where gradients $g_t$ are large, the surrounding landscapes are steep. The denominator $\sqrt{v_t}$ is also large, effectively reducing the learning rate. This adaptive

Table 9: Cross-optimizer ablation studies. We list validation accuracy for vision classification tasks across SGD and AdamW.

| Dataset -network | Optimizer | Schedule | training budget (epoch(%)) | | |
|---|---|---|---|---|---|
| | | | 30 (25%) | 60(50%) | 120(100%) |
| CIFAR100 -ResNet34 | SGD | SS | 70.85 | *75.58* | 77.28 |
| | | CS | 63.88 | 68.84 | 70.43 |
| | | CLR | 72.63 | 73.81 | 74.80 |
| | | 1C | *73.72* | 75.53 | 77.90 |
| | | BT | 72.53 | 75.40 | 78.49 |
| | | REX | 73.12 | 75.48 | *77.99* |
| | | UBA(ours)$\varphi = 5$ | **74.57** | **76.68** | **78.97** |
| | AdamW | SS | 64.01 | 70.93 | 73.65 |
| | | CS | 62.56 | 68.91 | 72.98 |
| | | CLR | 62.87 | 70.13 | 73.22 |
| | | 1C | 64.27 | 71.60 | 74.06 |
| | | BT | 64.01 | 71.44 | *74.34* |
| | | REX | **65.19** | *71.72* | 74.08 |
| | | UBA(ours)$\varphi = 0.5$ | *64.44* | **72.10** | **74.48** |
| | | | 75 (25%) | 150(50%) | 300(100%) |
| ImageNet -ResNet50 | SGD | SS | 74.84 | 76.96 | 78.10 |
| | | CS | *75.99* | *77.79* | *79.10* |
| | | CLR | 72.86 | 74.88 | 76.91 |
| | | 1C | 75.28 | 77.37 | 78.79 |
| | | BT | 75.71 | 77.48 | 78.66 |
| | | REX | 75.28 | 77.03 | 78.46 |
| | | UBA(ours)$\varphi = 5$ | **76.00** | **77.99** | **79.32** |
| | AdamW | SS | 74.64 | 77.51 | 79.01 |
| | | CS | 75.68 | 78.10 | 79.04 |
| | | CLR | 74.93 | 77.20 | 78.76 |
| | | 1C | 76.38 | 78.11 | 79.21 |
| | | BT | 76.10 | 78.14 | *79.05* |
| | | REX | *76.42* | *78.28* | 78.86 |
| | | UBA(ours) $\varphi = 0.5$ | **76.57** | **78.37** | **79.38** |

Table 10: Cross-$\varphi$ ablation studies.

| Dataset | Network | Epoch | Optimizer | UBA | | | | | |
|---|---|---|---|---|---|---|---|---|---|
| | | | | $\varphi=0.25$ | $\varphi=0.5$ | $\varphi=1$ | $\varphi=2.5$ | $\varphi=5$ | $\varphi=10$ |
| CIFAR100 | ResNet34 | 60 | SGD | 75.10 | 75.89 | 76.31 | 76.40 | **76.68** | 75.64 |
| | | | AdamW | **72.10** | 71.34 | 71.54 | 71.21 | 70.40 | 69.01 |
| ImageNet | ResNet50 | 300 | SGD | 78.35 | 78.75 | 78.95 | 79.24 | 79.32 | **79.40** |
| | | | AdamW | 79.26 | **79.38** | 79.12 | 79.10 | 78.72 | 78.47 |

behavior mimics preconditioning, implicitly lowering the condition number of the optimization landscape.

Our theoretical framework (Proposition 1) establishes a connection between parameter $\varphi$ and the condition number of the landscape around local minima: smaller $\varphi$ is favored for well-conditioned problems (low condition number), while larger $\varphi$ is beneficial for large condition number scenarios. Since preconditioning effect of AdamW naturally mitigates ill-conditioning, the schedule with smaller $\varphi$ is preferred. In contrast, SGD lacks such adaptive mechanisms, thus a larger $\varphi$ is more suitable for this situation. This alignment between theory and experiment underscores the importance of tailoring $\varphi$ to the optimizer's characteristics. *Thus, we recommend selecting higher values of $\varphi$ when*

Table 11: Performance across different initial learning rates.

| Dataset | Network | Schedule | Validation Accuracy | | |
|---------|---------|----------|-----------|------------|-------------|
| | | | lr=0.1 | lr=0.01 | lr=0.001 |
| CIFAR100 | ResNet34 | SS | 77.28±0.2401 | 73.48±0.4267 | 67.98±0.1473 |
| | | CS | 74.11±0.7690 | 74.66±0.5888 | 69.18±0.3736 |
| | | CLR | 74.77±0.1861 | 74.26±0.3523 | 68.84±0.4613 |
| | | 1C | 77.73±0.1868 | 75.80±0.1701 | 69.20±0.2152 |
| | | BT | 78.53±0.0751 | 77.11±0.1026 | 68.84±0.4613 |
| | | REX | 78.11±0.1124 | 77.40±0.4477 | 69.88±0.1253 |
| | | UBA(ours) | 79.02±0.2838 | 77.59±0.3522 | 70.50±0.0862 |

*facing challenging optimization difficulty, while employing lower $\varphi$ values in scenarios with smoother optimization difficulty.*

## E.8 Performance across different initial learning rates

Since each schedule will have a different optimal learning rate in practice, to thoroughly address this concern, we conducted additional experiments to evaluate the performance of all schedules with different learning rates.

We conducted auxiliary experiments on CIFAR100-ResNet34 for 3 runs. For computational efficiency, these experiments were run for slightly fewer epochs than the main experiments, with results reported over three independent runs. Through current observation and search, we identified 0.1 as the proper initial learning rate for most schedules. We use 0.01 and 0.001 to confirm this choice maximizes validation accuracy across methods. Additional smaller or larger values such as 0.3 or 0.0001 were explored in preliminary studies, showing consistent trends but lower peak accuracy. Thus we do not choose them for comparison.

The experimental results are listed in the following table 11. We report results of different base learning rate and compare the performance with optimal learning rate of each schedule.

## E.9 Performance across different periods

Our schedule has a periodic phase-based learning rate adjustment setting, where the learning rate at the $k$-th phase is dynamically determined by the $k$-th local minimum of the loss landscape. In the original conception, this design captures the optimization dynamics around successive local minima, with the hyper-parameter $\varphi$ related to the difficulty of each phase. Specifically, $\varphi$ controls the trade-off between exploration (large steps) and exploitation (small steps) within each phase.

To validate our scheduling strategy, we conduct experiments by varying $K$. We observe that when roughly setting $\varphi$ as a constant in each phase, performance degrades as training progresses. This suggests that a fixed $\varphi$ strategy cannot adapt to the changing landscape of multi-phase scheduling. Since the optimization difficulty should decrease during training process, we drop $\varphi$ as phase increases. By finely evaluate $\varphi$ in each phase, it achieves performance improvement, confirming the need for dynamic control. It makes sense since multi-phase captures the dynamic features of the loss surface more finely, it needs careful selection for $\varphi$, where $\varphi$ reflects optimization difficulty. These results highlight a fundamental trade-off: when $\varphi$ values per phase for multi-phase scheduling can be finely evaluated, multi-phase scheduling better captures the loss landscape's non-stationary behavior than single-phase scheduling. However, selecting optimal $\varphi$ values per phase for multi-phase remains non-trivial, which motivates future work on automated landscape-aware $\varphi$ tuning.

Table 12: Performance on multi-phase version UBA.

| Dataset | Network | $\varphi$ | Original | Multi-phase | | |
|---|---|---|---|---|---|---|
| | | | | $K=4$ | $K=6$ | $K=8$ |
| CIFAR100 | ResNet34 | $5$ | 76.68 | 76.54 | 75.37 | 75.05 |
| | | $5 \times 0.8^k$ | | 76.31 | 76.60 | 76.45 |

## F   Related work

**Budgeted training**    Researchers face significant challenges in achieving optimal model performance under fixed hardware and limited time. To address this, budgeted training has emerged as a broad research domain focusing on optimizing performance while minimizing resource usage. This research spans multiple aspects, including computation efficiency, model compression, training stability, convergence improvement, and optimization [41]. By strategically integrating efficient training approaches, budgeted training advances cost-effective model development, offering pathways to achieve high performance under realistic constraints.

Budgeted training can be divided from three main perspectives: data, model, and optimization. From the data perspective, budgeted training emphasizes the allocation of resources between the dataset and the algorithms that process it, such as the balance between the model size and the amount of data [2, 6, 24, 28]. From the model perspective, budgeted training searches the optimal number of model parameters within the given compute budget [17, 26, 31, 38]. Beyond data and model perspective, optimization based methods guide budgeted training based on learning rate schedules [5, 30, 43, 44], batch size [17] and other weight averaging method [26, 27].

Among these three aspects, optimization-based methods are particularly aligned with the objectives of budgeted-iteration training. In this domain, learning rate scheduling based methods are particularly aligned with the objectives of budgeted-iteration training. Smith et al. [43] propose a new learning rate schedule, named cyclical learning rates (CLR). It improves accuracy in fewer iterations without tuning and is relevant to super-convergence phenomenon [44]. Li et al. [30] introduce an alternative setting of existing learning rate schedules for budgeted training. Chen et al. [5] propose the reflected exponential schedule (REX) via a profile and sampling fashion. Learning rate based approaches achieve robust and high-performing results under various lengths of training iterations, which corresponds to our purpose in this work, i.e. budgeted-iteration training. Besides, this approach is plug-and-play, requiring no substantial alterations to the underlying model structure, making it readily adaptable to various deep network frameworks. Therefore, we explore the proper learning rate schedule to achieve budgeted-iteration training.

**Learning rate schedule**    The learning rate plays a pivotal role in controlling the non-convex optimization process during network training. Generally, the learning rate is gradually adjusted progressively alongside the iterations, which can be represented as $\eta = g(t)\eta_0$, where $g(t)$ is the schedule function which control the variation of learning rate, $t$ is the current iteration and $\eta_0$ is the initial learning rate.

Learning rate schedules have been extensively studied to improve training efficiency and model performance. The common scheme is the step decay schedule. Its schedule function can be represented as $g(t) = d_{\lfloor t/t_s \rfloor}$, where $t_s$ is the predefined decaying step and $d_i(d_i \geq d_{i+1})$ is predefined decaying scalar. A typical instance decreases the learning rate by a decaying scalar $0.1$ after $50\%$ epochs and by a decaying scalar $0.01$ after $75\%$ epochs [21]. Then Loshchilov et al. [32] observe that sharp decreases may prevent models from escaping local minima. Thus they propose cosine schedule function as $\frac{1}{2}(1 + \cos(\frac{t\pi}{t_s}))$, where $t_s$ is the predefined stage to restart the learning rate schedule, which has proven effective in transformer training. In addition, cosine schedule is a commonly-used schedule nowadays.

Pan et al. [36] propose Eigencurve, and demonstrate that minimax-optimal convergence can be achieved for quadratic objectives when the Hessian's eigenvalue distribution exhibits substantial skewness. Pan et al. [37] further introduce a learning rate schedule called elastic step decay as $\eta_t = \eta_0/2^k$, if $t \in [(1 - r^k)T, (1 - r^{k+1})T)$ to accelerate the pre-training for Berts, where $r$ and $k$ are hyper-parameters that control learning rate interval. Defazio et al. [11] developed the theoretically-

grounded framework for learning rate scheduling, establishing optimal linear decay schedules and refinements through rigorous mathematical derivation. Their work additionally provides the most extensive empirical evaluation of scheduling approaches to date. Hu et al. [25] propose a Warmup-Stable-Decay(WSD) for large model pretraining and promoting continuous training. WSD consists of three consecutive phases: warmup, stable constant, and decay. Luo et al. [34] discover a new schedule that outperforms the cosine schedule, resembling WSD but with lower final loss.

**Adaptive learning rate**   Compared to schedule methods, adaptive learning rate based methods focus on the learning rate adaptation based on local loss landscape statistics. Typical methods includes AdaGrad [13], Adadelta [54], RMSprop [23] and Adam [29]. These methods have been shown stabilizing the training process while effectively optimizing learning rate. Then Loshchilov et al. [33] propose AdamW optimizer which improves the performance in the transformer training. Recently, Chen et al. [7] discovers Lion optimizer for more memory-efficiency. Xie et al. [52] propose the Adan optimizer which implements 50% faster training than Adam and AdamW on commonly-used networks. Despite these improvements, some studies suggest that adaptive methods may underperform momentum SGD in terms of generalization [30, 51], a critical factor for budgeted-iteration training.

Learning rate design is not only significant in general training, but also critical in budgeted-iteration training which still remains a topic of debate. Some analyses advocate for small, constant learning rates to ensure stability and convergence [12]. On the contrast, one prevailing hypothesis suggests that large learning rates may facilitate crossing over sharp local minima in the optimization landscape [56]. Despite the lack of comprehensive theoretical explanations, a range of learning rate schedules inspired by the above analyses as heuristic guidelines has been widely adopted in practice, using variable learning rates to budgeted-iteration training [5, 30]. In this work, we explore learning rate schedule from optimization problem tailored to budgeted-iteration training, aiming to balance iteration budget constraints and generalization.

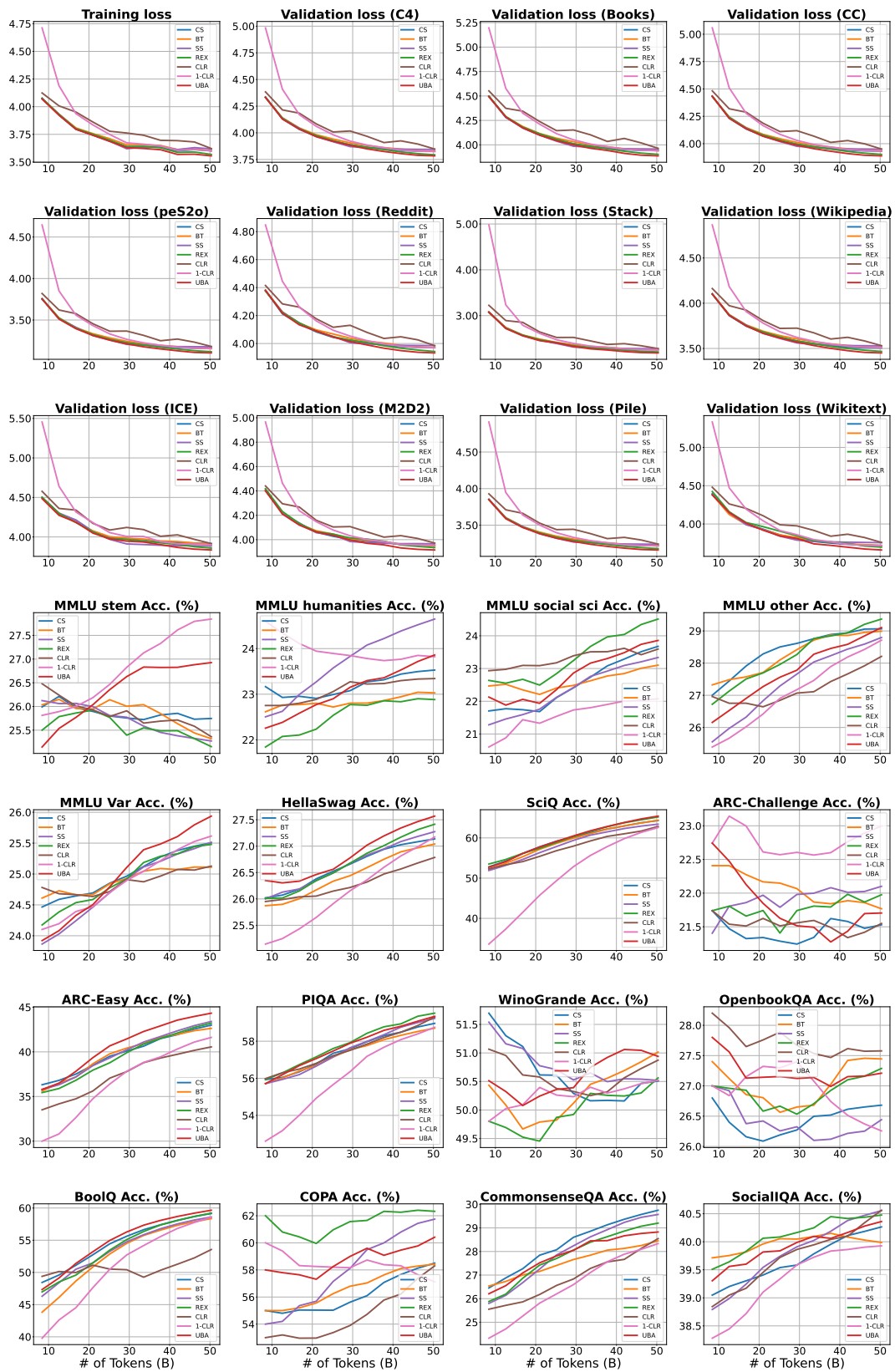

Figure 7: Overall performance for language tasks on OLMo-36M.

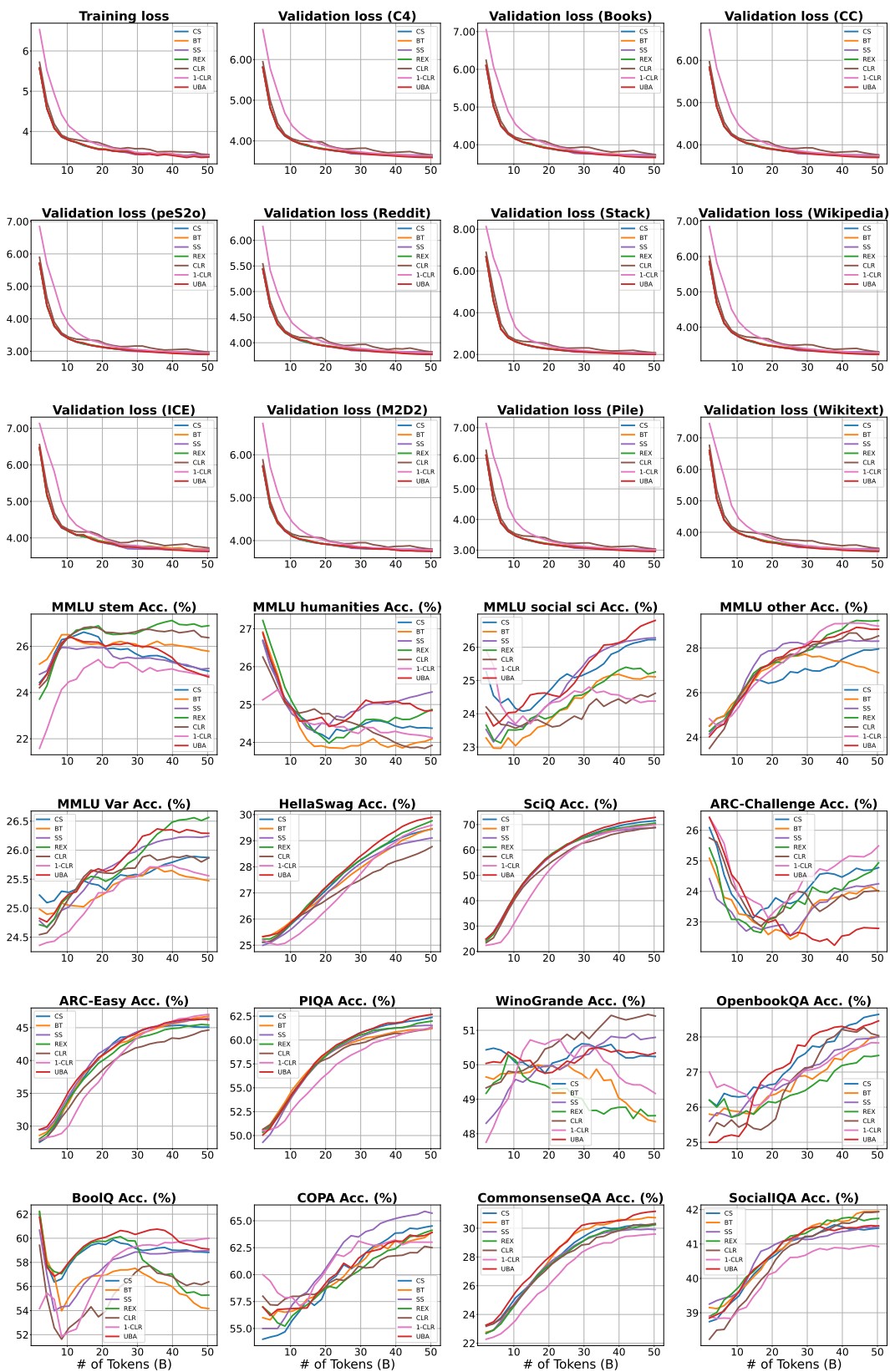

Figure 8: Overall performance for language tasks on OLMo-73M.

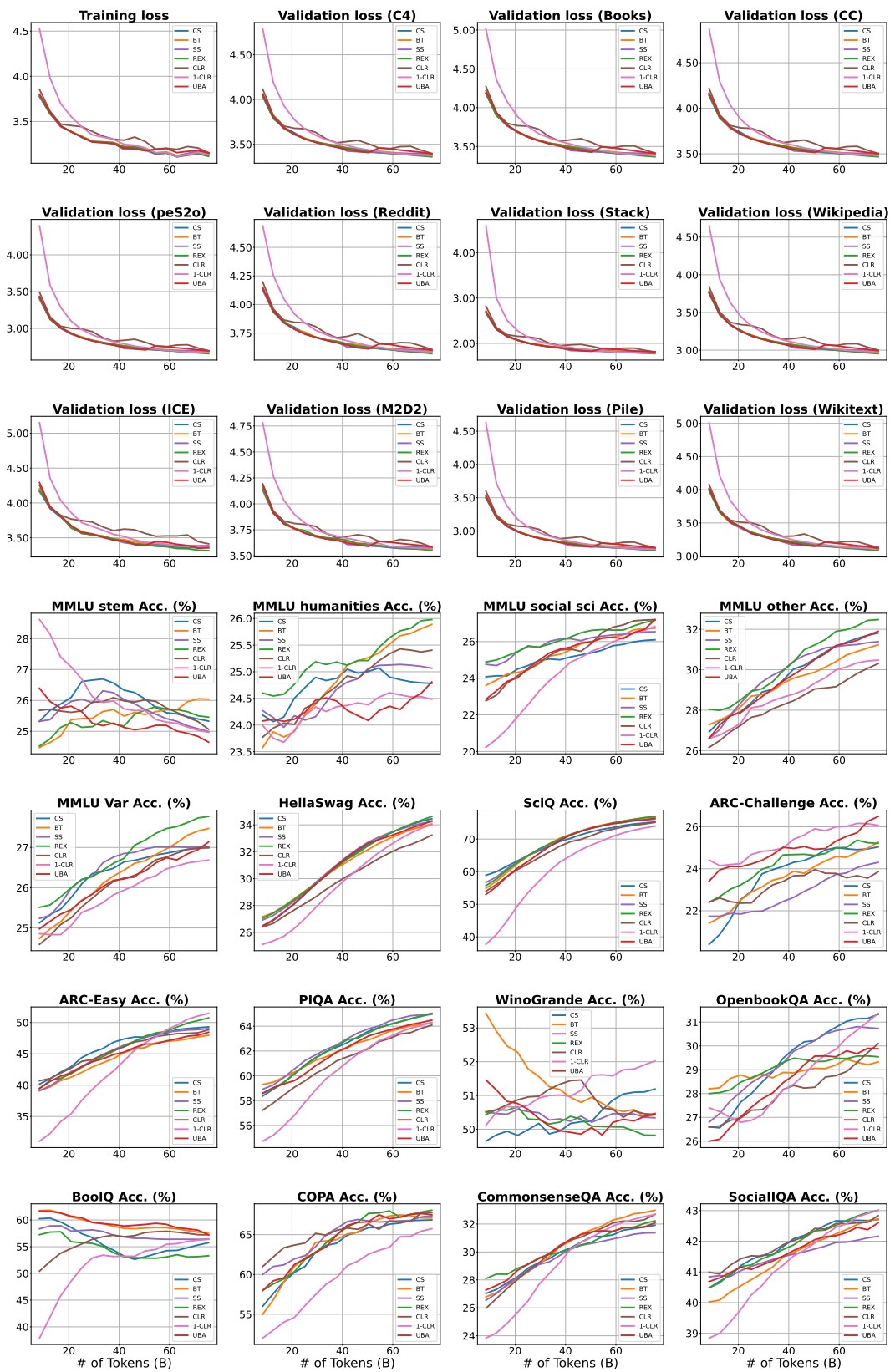

Figure 9: Overall performance for language tasks on OLMo-150M.

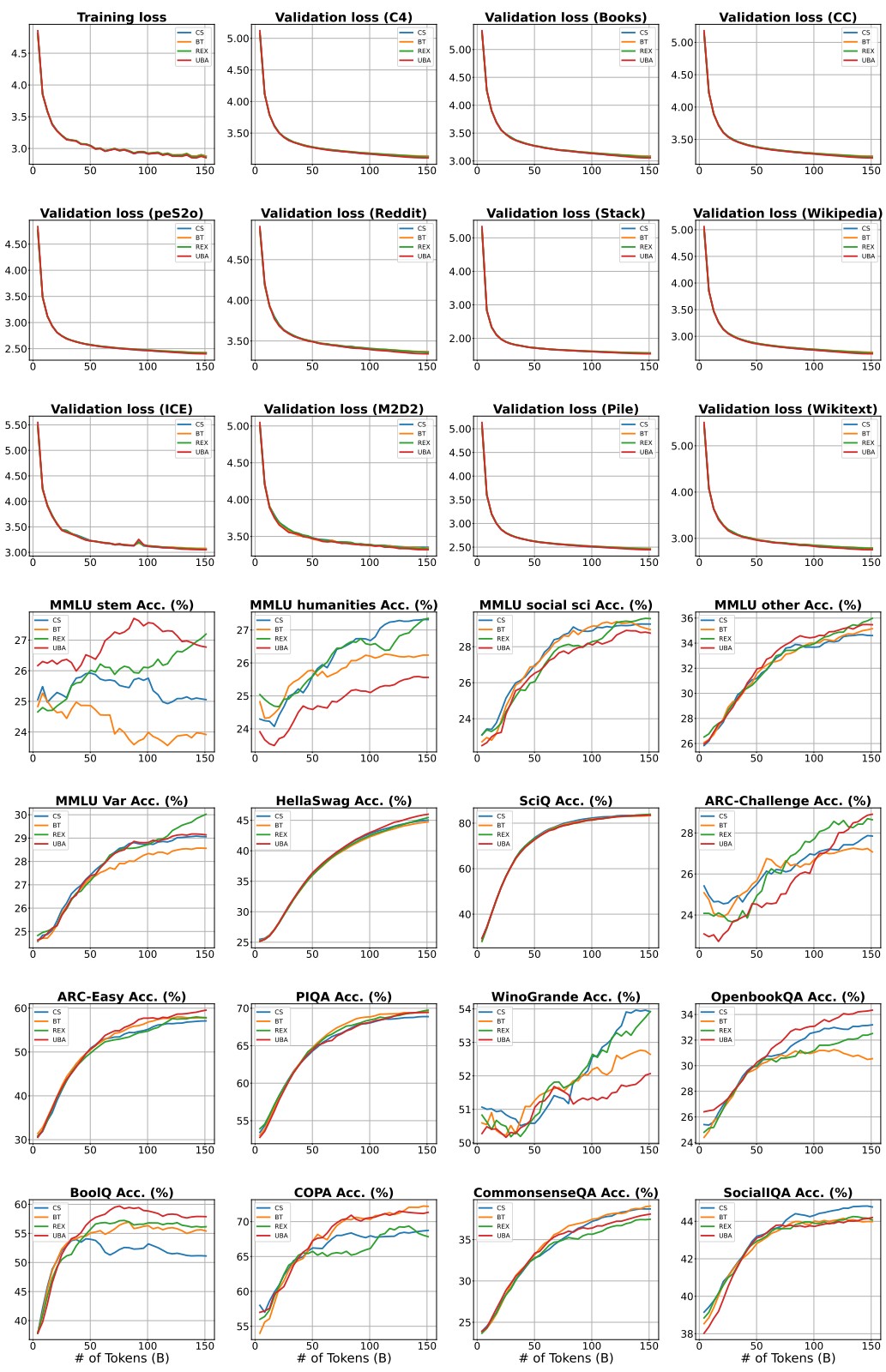

Figure 10: Overall performance for language tasks on OLMo-300M.

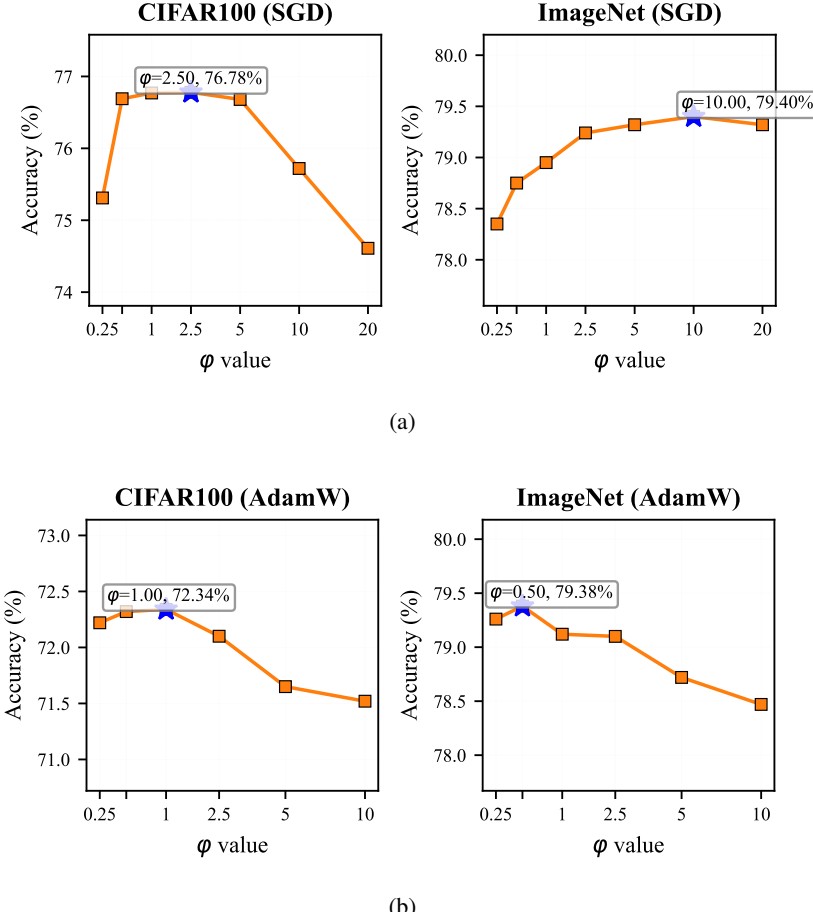

(a)

(b)

Figure 11: Visualization of cross-$\varphi$ performance.

