# OpenReview forum: "Stepsize anything: A unified learning rate schedule for budgeted-iteration training"
_NeurIPS.cc/2025/Conference — NeurIPS 2025 poster_

### Official Review · Reviewer_a6Zw · 2025-06-09

**Clarity:** 2
**Significance:** 3
**Originality:** 3
**Rating:** 5
**Confidence:** 3

**Summary:**

This paper proposes a theoretical framework for a unified learning rate schedule that is budget-iteration aware. The resulting schedule adapts to the loss landscape of the model being trained and better leverages the available training budget across different training conditions. This is done by addressing the worst-case loss function and approximating the optimization trajectory across the loss landscape by approximating it with a sequence of local optima. The authors include a comprehensive theoretical analysis of the proposed framework and the related learning rate schedule. The experimental section extensively explores a wide range of datasets and domains.

**Questions:**

See weaknesses.

**Ethical Concerns:**

["NO or VERY MINOR ethics concerns only"]

**Final Justification:**

I increased my score. The author's rebuttal addresses the main concerns raised by my review, and the response to the other reviewers is also sound.

**Limitations:**

As listed in the weaknesses, the authors do not analyze the computational overhead from the proposed approach.

**Paper Formatting Concerns:**

Please review the grammar and typos across the paper. The following are only some examples:
- Typo L24: across a wide range of...
- Typo L137: we aim to guarantee
- L140: what is $\xi$ ?
- L194: missing Appendix reference
- Typo L195: for the following reasons.

**Quality:**

3

**Strengths And Weaknesses:**

Strengths:
- The proposed method and framework are well motivated and based on a strong theoretical analysis and an intuitive concept.
- The paper has extensive theoretical and experimental justifications and analysis of the proposed approach.

Weaknesses:
- The practicality of the proposed method is not explored or discussed. The authors mention that practitioners can easily leverage this method, but do not provide a discussion about the implementation.
- The paper focuses on training efficiency, but the authors do not discuss the additional computational cost that the proposed learning rate schedule brings.
- The abstract and the introduction of the paper address the work from a high-level perspective and include scarce details of the proposed approach. The readers would benefit from more introductory details of the proposed theoretical framework and the proposed learning rate schedule in the initial parts of the paper.
- The text should be reviewed for typos and grammar. See the formatting section for more details on this.

---

> ### Author Rebuttal · Authors · 2025-07-29
>
> *$\mathbf{Weakness\enspace 1}$:The practicality of the proposed method is not explored or discussed. The authors mention that practitioners can easily leverage this method, but do not provide a discussion about implementation.*
>
> **Answer:** We thank the reviewer for raising this important point. **The implementation practicality has been thoroughly addressed in our submission**:
>
> - Our implementation **strictly follows the PyTorch official API**, i.e. it is inherited from the base class `torch.optim.lr_scheduler.LRScheduler` (Appendix E.3). It maintains full compatibility with all `torch.optim` optimizers. **The complete production-ready code is publicly available (see Appendix E.3 for link).**
> - We conduct parameter analysis for $\varphi$. It provides guidance for $\varphi$ selection, which makes practitioners leverage this method more easily.
> To further clarify, we give an example snippet of implementing UBA schedule:
>
> `import torch`
>
> `from custom_schedulers import UBA`
>
> `scheduler = UBA(optimizer,  num_warmup_steps,    num_training_steps,    varphi,    max_lr=0.1,    min_lr=0)`
>
> `scheduler.step()`
>
> In the revised manuscript, we will expand Appendix E.3 with more implementation details of UBA.
>
> *$\mathbf{Weakness\enspace 2}$:   The paper focuses on training efficiency, but the authors do not discuss the additional computational cost that the proposed learning rate schedule brings.*
>
> **Answer:** We appreciate the reviewer’s attention to computational efficiency. The proposed UBA schedule is designed to be **negligible computational overhead**.  Our **performance improvements come with negligible computational overhead**, which constitutes one of its key advantages:
>
> UBA introduces **only 6-15 scalar arithmetic operations per update**, which is compared to standard learning rate schedules (4-8 ops per iteration in standard schedulers like cosine annealing).
> Since *the operations of learning rate scheduling are **orders of magnitude smaller** than the FLOPs required for forward/backward passes in typical neural networks*, these extra operations for UBA is negligible computational overhead.
>
> Moreover, UBA requires minimal additional memory allocations or gradient computations required.
>
> We will include a dedicated subsection in revised manuscript on Computational Efficiency.
>
> *$\mathbf{Weakness\enspace 3}$:  The abstract and the introduction of the paper address the work from a high-level perspective and include scarce details of the proposed approach. The readers would benefit from more introductory details of the proposed theoretical framework and the proposed learning rate schedule in the initial parts of the paper.*
>
> **Answer:** We sincerely appreciate this constructive suggestion to improve the paper's accessibility. In the revised manuscript, we will make the following enhancements to the abstract and introduction:
>
> - The  Unified Budget-Aware schedule framework unifies robustness to curvature variations caused by data distribution shifts, mini-batch sampling noise, architecture-induced landscape properties and optimization dynamics. To achieve this, we **introduce an implicit random variable to model uncertainty in the objective function**. Moreover, we **theoretically transform the problem into an eigenvalue formulation**.
> - The framework also employs a minimax optimization approach to guarantee performance under worst-case conditions, since **the minimization of learning rate is operated on an upper bound.**
> - The optimization
> \begin{aligned}
> \min_{\eta_{1+T_{k}}, \eta_{2+T_{k}},..., \eta_{T_{k+1}}}
> &\max_{\lambda_{l}^{(k)} \leq \lambda_i^{(k)} \leq \lambda_{u}^{(k)}}
> \prod_{t=1+T_{k}}^{T_{k+1}} \left[(1-\eta_{t}\lambda_i^{(k)}) \right]^2 \\
> \text{s.t.} \quad & \eta_{1+T_{k}}, \eta_{2+T_{k}},..., \eta_{T_{k+1}} \in [\eta_{\min},\eta_{\max}], \\
> & k=1,2,\dots K
> \end{aligned}
> is solved by gradient projection method. Multiple initializations are used to explore the solution space. Despite varying initial conditions, numerical solutions exhibit consistent patterns. **These patterns are captured through a parametric approximation.**
> - By approximating numerical solutions via **a universal parametric function** (with single hyperparameter $\varphi$ linked to condition numbers), UBA eliminates per-network retraining while guaranteeing convergence.
> - The hyperparameter **$\varphi$ governs the functional form of the scheduling trajectory.** Through $\varphi$ modulation, the learning rate schedule can exhibit: **(a) Early slow decay → rapid descent → asymptotic convergence (b) Initial rapid drop → gradual decline → terminal smoothing.** Moreover, $\varphi$ directly controls the **transition rates between these phases**, enabling precise adaptation to optimization dynamics.
>
>
> *$\mathbf{Weakness\enspace 4}$:  The text should be reviewed for typos and grammar. See the formatting section for more details on this.*
>
> **Answer:** We sincerely appreciate the reviewer's careful reading and valuable feedback. We have thoroughly addressed the concerns regarding text polishing:
>
> - We corrected all identified typos. Parts of changes are listed as follows.
>
> L24:  ...across various across a wide range of ... → ...across a wide range of ...
>
> L137: we aim to guarantees ... → we aim to guarantee ...
>
> L140: $\xi$ denotes the data sampling on the dataset $D$→ Let $\xi$ denote the data sampling on the dataset $D$.
>
> L194: compared in Appendix. → compared in Appendix E.8
>
> L195: for reasons. →for the following reasons.
>
> L227: is not large. → is manageable.
>
> L301: Experiments 11→ Figure 11.
>
> - We have implemented all corrections in the revised manuscript, which has been proofread by two  colleagues.

---

### Official Review · Reviewer_LjTY · 2025-06-13

**Clarity:** 2
**Significance:** 3
**Originality:** 2
**Rating:** 5
**Confidence:** 4

**Summary:**

This paper proposes a quadratic framework for deriving a family of learning rates schedules parameterized by a $\varphi$ parameter. The schedules show strong results for ResNet training.

**Questions:**

N/A

**Ethical Concerns:**

["NO or VERY MINOR ethics concerns only"]

**Final Justification:**

I have increased my score. I hope that the authors can run more seeds for the final camera ready to get comprehensive error bars for the larger imagenet runs.
Also, a finer granularity learning rate sweep would be nice for cifar10.

**Limitations:**

Introducing an additional tunable parameter into your learning rate schedule is a clear limitation. I think it's discussed in reasonable depth.

**Paper Formatting Concerns:**

Some important plots are scaled too small (e.g. figure 1)

**Quality:**

2

**Strengths And Weaknesses:**

This paper has some interesting ideas but is very limited by a compressed presentation and the lack of proper citations to relevant work.

The idea of deriving a learning rate schedule numerically optimized for a quadratic has been explored before by Pan et. al (2021) in their Eigencurve paper. This Eigencurve paper is cited in the appendix, so the authors are aware of it. It should be in the body of the paper, and given the extreme similarity in the development framework and concept, the Eigencurve work needs to be properly credited. They also show that cosine-like schedule shapes arise from their framework.
Defazio et. al. (2023)'s work on theoretically motivated linear decay schedules should also be mentioned, as it's the only other completely theory driven framework for learning rate scheduling, and is currently the largest comparison of learning rate scheduling approaches.

The experimental results in this paper are strong and encouraging, but it's hard to trust the results given the following limitations:
 - No error bars or standard errors are reported. ResNet experiments are famously noisy, and the differences between different schedules are often at the same level as the noise. Highlighting UBA as the winning method for 75 epoch ResNet-50 training with accuracy 76.00 compared to cosines schedule's 75.99 is not appropriate as the run-to-run variability is more than 0.2%!
 - The use of a fixed learning rate for all schedules doesn't give fair comparisons. This is a major limitation. Generally each schedule will have a different optimal learning rate in practice. The choice of this learning rate is not motivated, and may favor certain schedules over others.
- Is the budgeted iteration method listed actually the linear decay schedule? this should be labelled as linear decay if so.
- The introduction is too long and and largely restates the abstract. It could be shortened significantly to make more room for discussion of related work (I see you have a longer related work section in the Appendix that you could use some of).

Overall I like the ideas in this paper and I think it's a good extension of the work of Pan et. al, and I would generally support acceptance if the authors commit to the improvements mentioned above, and a proper phrasing of their work within the context of the existing literature.

---

> ### Author Rebuttal · Authors · 2025-07-29
>
> *$\mathbf{Comment\enspace 1}$: ... Pan et. al (2021) in their Eigencurve paper. ... Defazio et. al. (2023)'s work...*
>
> **Answer:** We sincerely appreciate the reviewer's insightful comments regarding prior theoretical works on learning rate scheduling. We fully acknowledge that both Pan et al. (2021)'s Eigencurve and Defazio et al. (2023)'s linear decay framework represents seminal contributions to theory-driven schedule.
> **In the revision, we will cite the Eigencurve in the body of the paper. Moreover, we elevate discussion of Pan et. al (2021) Eigencurve and  Defazio et. al. (2023)'s work.** The revision is listed as follows.
>
> "Pan et al. propose Eigencurve,  and demonstrate that minimax-optimal convergence can be achieved for quadratic objectives when the Hessian's eigenvalue distribution exhibits substantial skewness. Pan et al. propose Extremebert, and further introduce a learning rate schedule called elastic step decay to accelerate the pre-training for Berts. Defazio et al. developed the theoretically-grounded framework for learning rate scheduling, establishing optimal linear decay schedules and refinements through rigorous mathematical derivation. Their work additionally provides the most extensive empirical evaluation of scheduling approaches to date.
>
> While Eigencurve and our work both employ quadratic optimization, our approach differs in: (a) incorporating robust min-max optimization, (b) constructing an adaptive learning rate schedule controlled by a hyper-parameter \varphi, (c) establishing the connection between \varphi and optimization difficulty. "
>
> We will clarify these points in the revision.
>
> *$\mathbf{Comment\enspace 2}$: No error bars or standard errors are reported......*
>
> **Answer:** We sincerely appreciate the reviewer's important concern regarding experimental reliability. We provide statistical validation (mean ± std over multiple runs) across tasks:
>
> On vision tasks. We conduct 5 independent runs for cifar10/100. Due to computational constraints and limited time within the rebuttal period, we use two independent experimental results for ImageNet-ResNet50 . We report the validation accuracy by mean accuracy ± standard deviation. Due to character limit, part of results are listed in the following table. The results demonstrate consistent performance of UBA. Full results will be tabulated in revised manuscript.
>
> | Dataset   | Network  | Method | 25% Budget   | 50% Budget  | 100% Budget  |
> |-|-|-|-|-|-|
> | CIFAR100|ResNet34|SS|71.092 ± 0.2676| 75.348 ± 0.1802|77.244±0.1808|
> |||CS|64.124 ± 0.1808|69.840 ± 0.7580|74.295±0.5958|
> |||CLR|73.336 ± 0.5635|73.412 ± 0.2835|74.748±0.2266|
> |||1C|73.384 ± 0.3227|75.036 ± 0.6382|77.860±0.2239 |
> |||BT|72.750 ± 0.2611|75.346 ± 0.2423|78.578±0.2705 |
> |||REX|73.642 ± 0.5635|75.136 ± 0.3110|78.040±0.1430  |
> |||**UBA**| **74.604 ± 0.1508**|**76.626 ± 0.2678**|**78.948 ± 0.2818**|
> | ImageNet|ResNet50|SS|74.738 ± 0.1386|77.235 ± 0.3861|78.559 ± 0.6435|
> |||CS|75.839 ± 0.2220|77.945 ± 0.2164|79.067 ± 0.0438|
> |||CLR|73.899 ± 1.4609|76.037 ± 1.6419|77.837 ± 1.3053|
> |||1C|75.831 ± 0.7792|77.743 ± 0.5218|78.997 ± 0.2956|
> |||BT|75.907 ± 0.2786|77.809 ± 0.4624|78.855 ± 0.2814|
> |||REX|75.853 ± 0.8047|77.658 ± 0.8853|78.662 ± 0.2828|
> |||**UBA**| **76.289 ± 0.4031**|**78.258 ± 0.1640**|**79.388 ± 0.0170**|
>
> On LLM training.
> - **Computational Reality:** Given the exorbitant costs with single  run  for LLMs,  multiple experimental trials is often impractical and rarely adopted in practice. **However, our study encompasses 40-50 experimental runs, cumulatively consuming 30,000 A100 GPU-hours.** It is a substantial investment for a research paper (for comparison, DeepSeek-V3 utilized 3M H800 GPU-hours). This scale validates robustness and scalability.
> - **Empirical Reliability Criteria:** The reliability of such experiments can be effectively assessed through the data-parameters ratio (D/N ratio). As Chinchilla scaling laws saying, a D/N ratio of 20 is already sufficient. Notably, seminal works like LLaMA-2 and OLMo omit multi-trial reporting when operating at D/N ratios >200.
> **In our LLM experiments, the D/N ratios exceed this threshold** (50B/73M ≈ 685, 75B/151M ≈ 497, 150B/300M ≈ 500), indicating that each parameter has been allocated ample data. **This high D/N regime ensures: (a) stable final performance, not just early convergence, and  (b) representative single-run results with minimal fluctuation.** This practice aligns with contemporary  LLM training paradigms.
>
> *$\mathbf{Comment\enspace 3}$: The use of a fixed learning rate for all schedules doesn't give fair comparisons. ...*
>
> **Answer:** We sincerely appreciate the reviewer's insightful critique regarding learning rate selection. We address this concern through **methodological clarification** and **additional experimental validation**:
>
> **Methodological clarification**:
> Our original design rationale follows two points to ensure fairness:
> - According to Learning rate scaling law, the optimal learning rate scales as a power-law with the size of model, batch size and data. Thus, the predicted optimal learning rates fall within a narrow range and the peak performance differences with this narrow range may vary slightly.
> - To ensure rigorous experimental control, all schedules in one task use the same base learning rate. However, this rate was determined through multiple-run search to enable peak performance for most schedules. Such standardization isolates the scheduler's effect from other variables.
>
> However, **we fully agree with your insightful observation that each schedule will have a different optimal learning rate in practice.** Therefore, to thoroughly address this concern, we conducted additional experiments to evaluate the performance of all schedules with different learning rates.
>
> **Extended validation**:
> We operate experiments on CIFAR100-ResNet34 for 3 runs.
> Through current observation and search, we identified 0.1 as the proper initial learning rate for most schedules. We use  0.01 and 0.001 to confirm this choice maximizes validation accuracy across methods. Additional smaller or larger values such as 0.3 or 0.0001 were explored in preliminary studies, showing consistent trends but lower peak accuracy. Thus we do not choose them for comparison.
>
> The experimental results are listed in the following table. We report results of different base learning rate and compare the performance with optimal learning rate of each schedule.
>
> | Dataset | Network | Method| lr=0.1| lr=0.01| lr=0.001 | Performance on optimal lr |
> |-|-|-|-|-|-|-|
> | CIFAR100| ResNet-34(100% budget) | SS|77.28±0.2401|73.48±0.4267|67.98±0.1473| 77.28±0.2401 |
> ||| CS| 74.11±0.7690| 74.66±0.5888| 69.18±0.3736| 74.11±0.7690|
> ||| CLR| 74.77±0.1861|74.26±0.3523|68.84±0.4613|74.77±0.1861|
> |||1C| 77.73±0.1868| 75.80±0.1701| 69.20±0.2152| 77.73±0.1868|
> |||BT (linear decay) |78.53±0.0751| 77.11±0.1026| 68.84±0.4613|78.53±0.0751|
> |||REX| 78.11±0.1124|77.40±0.4477| 69.88±0.1253|78.11±0.1124|
> |||**UBA**| **79.02±0.2838**| **77.59±0.3522**| **70.50±0.0862**| **79.02±0.2838**|
>
> From the table, we can see that compared with schedule performance on corresponding optimal learning rate, **UBA also achieve the leading place, demonstrating the superiority regardless of the base learning rate.**
>
> In the revision, we will detail this analysis of varying base learning rate with the results table.
>
> *$\mathbf{Comment\enspace 4}$: Is BT actually the linear decay?...*
>
> **Answer:** We thank the reviewer for careful reading. The "BT" method is mathematically equivalent to linear decay scheduling. The original "budgeted training" naming reflected its application context, but we will standardize to "Linear Decay" in the revised manuscript to avoid confusion.
>
> *$\mathbf{Comment\enspace 5}$: The introduction is too long ...*
>
> **Answer:** Thank you for your suggestion. **We will shorten the introduction and incorporate relevant material from the Appendix' to strengthen the discussion of related work.** The main points of revision is listed as follows.
>
> "Pan et al. propose Eigencurve,  and demonstrate that minimax-optimal convergence can be achieved for quadratic objectives......Their work additionally provides the most extensive empirical evaluation of scheduling approaches to date.
>
> While Eigencurve and our work...... optimization difficulty. "
> (We respectfully note that **the complete revision details appear in our response to Comment 1**. We have omitted the detailed revision content here due to character limit while ensuring all substantive changes are documented.)
>
> *$\mathbf{Comment\enspace 6}$: Overall I like the ideas in this paper...I would generally support acceptance if the authors commit to the improvements mentioned above, and a proper phrasing of their work within the context of the existing literature.*
>
> **Answer:** Thank you for your positive feedback and constructive suggestions. We sincerely appreciate your support and will **carefully address all the requested improvements, including refining the paper’s positioning within the existing literature.**
>
> As you recommend, we make the following improvements:
> - We have addressed all the points raised in your review, as detailed in our responses above.
> - We have revised the manuscript to better position our work within the existing literature. Specifically, we properly credit the work of Pan et. al (2021)  and  Defazio et al.(2023). We also clarify how our approach differs in handling adaptive learning rate  design and establishing the meaning of parameter.
>
> *$\mathbf{Limitations}$: ...tunable parameter...*
>
> **Answer:** Thanks for this observation. We agree with your opinion. Thus, we conduct parameter analysis, demonstrating it robustness (Sec.4.3 and Appendix E.7). Moreover, we give a guideline for parameter selection (Appendix E.7). Furthermore, we discuss the potential of $\varphi$ to achieve adaptive improvement (Sec. 5) .

---

### Official Review · Reviewer_DPSV · 2025-07-01

**Clarity:** 3
**Significance:** 2
**Originality:** 2
**Rating:** 4
**Confidence:** 3

**Summary:**

This paper proposes a unified budget-aware learning rate scheduling strategy Unified Budget-Aware schedule, which aims to solve the problem of how to efficiently train models under a fixed training iteration budget. The author constructs a theory-driven optimization framework, designs a scheduling function that only requires a hyperparameter $\varphi$ to control the worst case, and establishes the connection between it and the optimization condition number. Experiments on a large number of vision and language tasks have shown that UBA outperforms existing mainstream scheduling strategies under different models, tasks, and budgets, and has good generalization and robustness.

**Questions:**

Questions:

1: Have the authors evaluated whether the local quadratic approximation still applies in the early stages of training?

2: $\varphi$ is fixed and related to the condition number. Have the authors considered adapting $\varphi$ based on the optimization state?

3: Although the authors have verified that it is effective on both SGD and AdamW, the scheduling strategy itself is currently independent of the optimizer. Have the authors considered combining the $\varphi$ or $η_t$ design with the state term in adaptive gradient methods?

**Ethical Concerns:**

["NO or VERY MINOR ethics concerns only"]

**Final Justification:**

The authors provided solid experimental validation addressing the applicability of their method in early training stages, and offered a clear sensitivity analysis for the key parameter. I also appreciate the preliminary work on adaptive scheduling based on optimizer states. These responses address my main concerns, and I am raising my score to 4.

**Limitations:**

Partially. While the limitations are addressed in the conclusion, more discussion on the applicability of the quadratic approximation would improve the clarity and transparency of the work. No societal risks are apparent.

**Quality:**

3

**Strengths And Weaknesses:**

Strengths:

1: This paper derives a UBA schedule based on a well-motivated min-max optimization framework with a rigorous approach.

2: This paper proves the proposed scheduler is an exact solution to a specific min-max problem and analyzes its convergence in noisy environments.

3: Experiments in vision and language tasks show that the method in this paper outperforms baseline models under various settings.

Weaknesses:

1: The method relies on a local quadratic approximation of the loss surface, which is valid for models with smooth structures, but its applicability is still questionable in transformers with small batch sizes in the early training stage.

2: Although the authors achieved good results by fixing $\varphi$, there is still a lack of robustness analysis on how to adaptively select $\varphi$ or when $\varphi$ is selected incorrectly.

---

> ### Author Rebuttal · Authors · 2025-07-27
>
> *$\mathbf{Weakness\enspace 1}$:  The method relies on a local quadratic approximation of the loss surface, which is valid for models with smooth structures, but its applicability is still questionable in transformers with small batch sizes in the early training stage.*
>
> **Answer:** We sincerely appreciate the reviewer's insightful comment regarding the validity of our method  when the loss landscape may exhibit less smooth characteristics. We fully agree that this requires careful analysis. However, our empirical results demonstrate that the applicability of our proposed schedule remains robust despite these potential challenges. Below we provide detailed explanations:
>
> - To prevent the approximation error from compromising practical applicability, we employ robust optimization design. Our approach employs minimax optimization to determine learning rates, optimizing for worst-case scenarios within the iteration process. First, this formulation mitigates early training approximation errors by operating on an upper bound. Second, the explicit solution avoids direct network value computation or local quadratic approximations.
> - We incorporate standard warmup phases (linear warmup from 0 to maximum LR). This industry-standard practice  effectively circumvents early-stage approximation issues before applying UBA. Empirical results on OLMo demonstrate UBA's effectiveness in early training stages.
> - Due to the complexity of NN loss landscapes (affected by architecture, data distribution, etc.), direct evaluation is intractable. We use our experiment results to demonstrate its applicability in Transformers in early training. Comprehensive LLM experiments validate UBA's performance across model sizes (36M-300M). The training curves in both **main text (Figure 2) and Appendix (Figure 7,8,9,10) provide visual confirmation of UBA's robust behavior on transformer based model in early training**. We list the following table to present quantitative comparisons with  baseline scheduling methods, showing consistent stability of our approach and competitive performance throughout early training phases.
>
> | Training | Size | Scheduler | HSWAG | ARC-C | COPA |
> |-|-|-|-|-|-|
> | 10%|36M  | BT|25.8  | 22.4  | 55.0 |
> |||CS    | 26.0  | 21.8  | 55.0 |
> |||UBA       | **26.4**  | **22.8**  | **58.0** |
> ||73M|BT |**25.9**  | 23.3  | **56.0** |
> |||CS|**25.9**| 23.7  | 55.0 |
> |||UBA|**25.9**  | **24.4**  | **56.0** |
> ||150M  | BT| 27.0  | 21.6  | 56.0 |
> |||CS| **27.5**  | 21.0  | 57.0 |
> |||UBA| 27.0  | **24.0** | **58.0** |
> ||300M  |BT| 25.3  | 25.2  | 55.0 |
> |||CS| **25.4**  | **25.3**  | **57.0** |
> ||| UBA  | 25.2  | 23.0  | **57.0** |
>
> - In the future, we will enhance UBA through $\varphi$ universal flexibility for non-smooth adaptation. Dynamic adjustment for high-curvature regions will enable adaptive optimization "modes".
> - The revised manuscript will include  discussion on method applicability.
>
> *$\mathbf{Weakness\enspace 2}$:    Although the authors achieved good results by fixing $\varphi$ , there is still a lack of robustness analysis on how to adaptively select $\varphi$  or when $\varphi$ is selected incorrectly.*
>
> **Answer:** We sincerely appreciate your valuable suggestions, which we have addressed from both **experimental** and **methodological** perspectives.
>
> **Experimental validation:**
> - Our original manuscript **already includes a comprehensive sensitivity analysis of the key parameter $\varphi$**. Our experiments systematically evaluate $\varphi \in $ {0.25, 0.5, 1.0, 2.5,5,10} across different optimization scenarios.  **The results are plotted in Figure 11 and listed in Table 11 in Appendix E.7. In addition, Sec. 4.3 provides the analysis of $\varphi$'s impact.**
> - **The experimental results demonstrate that, UBA consistently achieves optimal performance across different tasks (CIFAR100, ImageNet) and optimizers (AdamW, SGD)**. Moreover, from our parameter robustness analysis, we recommend selecting higher values of $\varphi$ when facing challenging optimization difficulty, while employing lower $\varphi$ values in scenarios with smoother optimization difficulty.
> - Notably, **even with suboptimal $\varphi$ selection,** CIFAR100-SGD: Accuracy remains **above 76.3** (best 76.68) for $\varphi \in$  {1.0, 2.5,5} . ImageNet-AdamW: Accuracy stays **above 79.10** (best 79.38) for $\varphi \in$ {0.25, 0.5, 1.0, 2.5}. This confirms our method's **robustness to parameter selection, especially in the situation that when $\varphi$ is selected incorrectly**, as you mentioned.
>
> **Methodological development:**
> - Regarding **adaptive $\varphi$ selection** during training (as mentioned in Sec. 5), We've made **recent progress** by incorporating first and second-order moment estimates into the UBA optimization framework. Preliminary experiments show modest improvements over fixed-$\varphi$ approaches. The results are listed in the following table.
>
> ||| Epoch| Optimizer  | $\eta_0$   | Original | Adaptive $\varphi$ |
> |-|-|-|-|-|-|-|
> |||| SGD| 0.1  | 79.0| 79.15|
> | CIFAR100  | ResNet34 | 300   | AdamW     | 0.1| 75.94 | 77.42|
> ||||| 0.01| 73.98| 75.98|
> ||||| 0.001| 74.27| 75.16|
>
> The results show that, although adaptive $\varphi$ improves performance marginally under SGD optimizer, the results under AdamW show **promising potential of adaptive $\varphi$ usage on UBA**. Moreover, current implementation requires further refinement, which we're actively pursuing.
>
> In the  revision, we will highlight the parameter robustness results more prominently. Moreover, we  will include discussion of dynamic $\varphi$ selection in the revised manuscript.
>
> *$\mathbf{Question\enspace 1}$: Have the authors evaluated whether the local quadratic approximation still applies in the early stages of training?*
>
> Answer:  We sincerely appreciate the reviewer's insightful comment.
>
> - Methodological Robustness: Our UBA learning rate is derived as an explicit solution from the optimization problem, avoiding direct computation of network-dependent values or local quadratic approximations in early training. Moreover, the robust min-max framework mitigates approximation errors of local quadratic approximation.
> - Empirical Validation: Due to the complexity of NN loss landscapes (affected by architecture, data distribution, etc.), direct evaluation is intractable. Instead, **we empirically validate UBA’s applicability in early training using OLMo**, a state-of-the-art Transformer model sensitive to optimization dynamics. **We plot training curves of validation losses and accuracy on OLMo between different sizes(36M, 73M, 150M and 300M)**. The training curves in both **main text (Figure 2) and Appendix (Figure 7,8,9,10)** provide visual confirmation of UBA's robust behavior on Transformer based models in early training . We list the validation accuracy in the 10% training stage as an example **in the table of answer to "Weakness 1", showing consistent stability of our approach and competitive performance throughout early training phases**.
>
> To further strengthen our claims, we will highlight these results in the revised manuscript.
>
> *$\mathbf{Question\enspace 2}$: $\varphi$ is fixed and related to condition number. Have the authors considered adapting $\varphi$ based on the optimization state?*
>
> **Answer:** We sincerely appreciate your insightful suggestions, which resonate strongly with our planned future work directions. Given that this study establishes $\varphi$ as an indicator of optimization difficulty, a natural extension is to investigate difficulty representations derived from optimizer states, including gradients, momentum terms, second-order moments, and temporal progression
> - Our current exploration focuses on two methodological advancements. First, **we develop a mechanism that incorporates time steps information into min-max optimization**. The numerical results generate a characteristic warmup-stable-decay pattern that we integrate with the UBA schedule. Initial validation shows  performance gains across some tasks.
> - Second, we introduce first and second-order moment estimators into the UBA framework. Preliminary experiments demonstrate improvements over fixed $\varphi$ baselines. **Current  performance gain demonstrates promising potential of adaptive $\varphi$ design**.
> The findings will be will be included in the discussion section in our upcoming revision.
>
> *$\mathbf{Question\enspace 3}$: Although the authors have verified that it is effective on both SGD and AdamW, the scheduling strategy itself is currently independent of the optimizer. Have the authors considered combining the $\varphi$ or $\eta_t$ design with the state term in adaptive gradient methods?*
>
> **Answer:** We sincerely appreciate your valuable suggestions, which closely align with our research directions.
> - Building on UBA's empirical validation, we're extending the framework to incorporate adaptive gradient methods as additional constraints in our optimization framework.
> - Regarding dynamic $\varphi$ selection during training (as mentioned in our Sec.5), We've made recent progress by incorporating first and second-order moment estimates into the UBA optimization framework. Preliminary experiments show modest improvements over fixed-$\varphi$ approaches. The results are listed in the table (see the second table, in answer of "Weakness 2").
> - Adaptive $\varphi$ demonstrates stronger synergy with AdamW than SGD, suggesting better compatibility with adaptive optimizers.
> - In the future, we will refine implementation for more robust gains, developing advanced moment estimation techniques and theoretical  analysis and broader validation.
>
> *$\mathbf{Limitations}$: Partially. While the limitations are addressed in the conclusion......*
>
> **Answer:** We sincerely appreciate this constructive feedback. We will strengthen our discussion including the above answers of weaknesses, such as the quadratic approximation's applicability, in the revised manuscript.

---

> > ### Comment · Reviewer_DPSV · 2025-08-05
> >
> > Thank you for the thoughtful and detailed rebuttal. I appreciate the experimental validation and the promising preliminary work on adaptive scheduling. These responses address my main concerns, and I am raising my score.

---

> > > ### Author Response · Authors · 2025-08-05
> > >
> > > We sincerely appreciate your time and constructive feedback.  We are grateful for your support in improving this work.

---

### Official Review · Reviewer_jwKN · 2025-07-03

**Clarity:** 3
**Significance:** 2
**Originality:** 3
**Rating:** 4
**Confidence:** 4

**Summary:**

This paper proposes a Unified Budget-Aware (UBA) learning rate schedule designed to address training under limited computational resources and varying model scales. UBA requires tuning only a single hyperparameter, φ, which controls the learning rate step size and balances adaptivity with simplicity. Despite its minimal tuning cost, the method consistently yields marginal but robust improvements across a range of models, tasks, and training iteration budgets.

**Questions:**

1. Given the marginal improvements reported, how does the proposed UBA schedule compare to baseline schedules under more challenging or large-scale tasks where training budgets are more constrained?

2. Can the authors provide statistical significance analyses (e.g., mean and standard deviation across multiple runs) to validate that the reported improvements are consistent and not due to random fluctuations?

3. While UBA is claimed to generalize across model scales, have the authors considered evaluating it on a broader set of architectures (e.g., ResNet101, ViT, or Transformer-based models) to better support this claim?

**Ethical Concerns:**

["NO or VERY MINOR ethics concerns only"]

**Final Justification:**

After carefully review the response provided by author, they has sufficient empirical observation on different settings and models, although the UBA still marginal improvement from previous setting, but it consist improve on different tasks. Such empirical evidence might benefit future work in community. Given that I have improve my rate to 4.

**Limitations:**

Yes

**Paper Formatting Concerns:**

No formatting concerns

**Quality:**

2

**Strengths And Weaknesses:**

Strengths

1. The paper provides a clear and elegant theoretical analysis of step size selection under finite-iteration training budgets, culminating in a unified learning rate schedule governed by a single hyperparameter, φ. This formulation is both principled and practical, offering a clean balance between theoretical rigour and ease of application.

Weaknesses

2. The performance improvements over baseline learning rate schedules are marginal across tasks and architectures, raising questions about the practical impact of the proposed method.

3. Experimental results are reported using single-value metrics without accompanying variance or standard deviation. Given the small performance gains, statistical measures (e.g., mean ± std over multiple runs) are necessary to assess whether improvements are significant and reproducible.

4. While the authors claim UBA generalises across model scales, the experimental validation spans a relatively narrow range—from ResNet-18 to ResNet-50—limiting the strength of the claim. Broader evaluation on larger or more diverse model families would strengthen this argument.

---

> ### Author Rebuttal · Authors · 2025-07-27
>
> *$\mathbf{Weakness\enspace 1}$:The performance improvements over baseline learning rate schedules are marginal across tasks and architectures, raising questions about the practical impact of the proposed method.*
>
> *$\mathbf{Question\enspace 1}$ Given the marginal improvements reported, how does the proposed UBA schedule compare to baseline schedules under more challenging or large-scale tasks where training budgets are more constrained?*
>
> **Answer:** We appreciate the reviewer’s observation. While the gains may appear modest at first glance, we argue that UBA’s consistent improvements with negligible computational costs across all tested scenarios demonstrate its significant practical value, particularly in large-scale applications.
> Below, we clarify five key aspects supporting this claim:
>
> - **Unified improvements across all settings.** Unlike task-specific optimizations, UBA consistently outperforms baseline schedules  without any per-task tuning. Moreover, UBA reliably improves performance in every tested scenario across  **diverse architectures (CNNs, Transformers), varying model scales (from 36M to 300M parameters), multiple benchmarks (vision, language)**. This reliability aligns perfectly with our core thesis: UBA provides a universal and task-agnostic solution for training dynamics optimization.
>
> - **Extensive validation on LLM (OLMo).** Our experiments on OLMo-36M, OLMo-73M, OLMo-150M, OLMo-300M (Sec. 4.2)  address the scenario of constrained training budgets through four key dimensions: (1) The OLMo family is a **modern Transformer architecture**. Moreover, Transformer-based models are known to be sensitive to optimization dynamics, making consistent improvements more meaningful. It is representative of **"challenging or large-scale tasks"**.  (2)  The progressive scaling from **36M to 300M** parameters systematically tests the method's robustness across different budget constraints. (3) We conduct sufficient overtrain experiments. All models are trained with **D/N ratios >500×**, far above Chinchilla's 20× sufficiency threshold. It ensures stable final performance, not just early convergence. (4) **Our study encompasses 40-50 experimental runs, cumulatively consuming 30,000 A100 GPU-hours—a substantial investment for a research paper** (for comparison, DeepSeek-V3 utilized 3M H800 GPU-hours). This scale validates robustness and scalability.
> *Overall, UBA  consistently improves final accuracy across all OLMo models (see Sec. 4.2 and Appendix E.5).  As a result, this systematic evaluation on this modern Transformer architecture provides particularly compelling evidence for UBA's practical utility and practical impact.*
>
> - **Significant improvements in LLM training scenarios.** Although the performance gain of UBA may appear numerically modest on some tasks at first glance, our method achieves significant improvements across multiple benchmarks. For instance, it can be see in Figure 2 in the current paper, UBA matches baseline performance with 25B fewer tokens (125B vs. 150B) on HellaSwag Benchmark, representing a **20% reduction in training cost**. Moreover, the same target accuracy is achieved using 33% less data (100B vs. 150B tokens) on ARC-E Benchmark, while maintaining identical model configurations , representing a **33% reduction in training cost**. These improvements become particularly impactful when considering scaling laws, i.e. the logarithmic scaling of LLM performance with data volume.
>
> - **Performance gains v.s. Scale.** The significance of performance gains must be assessed relative to both model parameter scale and training token volume. What appears as a marginal improvement at small scales can represent substantial absolute gains when scaled to foundation model sizes. Although some metric improvements appear numerically modest in absolute terms, their scaled impact becomes substantial given the massive parameter counts and token volumes characteristic of modern LLMs.
>
> - **Negligible additional costs.**  Our improvements come with negligible trade-offs in computational overhead and memory footprint while maintaining training stability.
>
> We will clarify these points in the revision to better highlight UBA’s generalization and practical utility.
>
> *$\mathbf{Weakness\enspace 2}$: Experimental results are reported using single-value metrics without accompanying variance or standard deviation. Given the small performance gains, statistical measures (e.g., mean ± std over multiple runs) are necessary to assess whether improvements are significant and reproducible.*
>
> *$\mathbf{Question\enspace 2}$: Can the authors provide statistical significance analyses (e.g., mean and standard deviation across multiple runs) to validate that the reported improvements are consistent and not due to random fluctuations?*
>
> **Answer:** We thank the reviewer for this important suggestion. To rigorously address the concern about random fluctuations, we provide the following statistical evidence from our experiments:
>
> **On vision tasks**: We conduct 5 independent runs for cifar10-ResNet18 and cifar100-ResNet34. Due to computational constraints and limited time within the rebuttal period, we use two independent experimental results for ImageNet-ResNet50.  We report the validation accuracy by mean accuracy ± standard deviation. Due to character limit, part of results are listed in the following table, demonstrating consistent performance. We will include these statistical validation in the next version of paper.
>
> | Dataset   | Network  | Method | 25% Budget   | 50% Budget  | 100% Budget  |
> |-|-|-|-|-|-|
> | CIFAR100|ResNet34|SS|71.092 ± 0.2676| 75.348 ± 0.1802|77.244±0.1808|
> |||CS|64.124 ± 0.1808|69.840 ± 0.7580|74.295±0.5958|
> |||CLR|73.336 ± 0.5635|73.412 ± 0.2835|74.748±0.2266|
> |||1C|73.384 ± 0.3227|75.036 ± 0.6382|77.860±0.2239 |
> |||BT|72.750 ± 0.2611|75.346 ± 0.2423|78.578±0.2705 |
> |||REX|73.642 ± 0.5635|75.136 ± 0.3110|78.040±0.1430  |
> |||**UBA**| **74.604 ± 0.1508**|**76.626 ± 0.2678**|**78.948 ± 0.2818**|
> | ImageNet|ResNet50|SS|74.738 ± 0.1386|77.235 ± 0.3861|78.559 ± 0.6435|
> |||CS|75.839 ± 0.2220|77.945 ± 0.2164|79.067 ± 0.0438|
> |||CLR|73.899 ± 1.4609|76.037 ± 1.6419|77.837 ± 1.3053|
> |||1C|75.831 ± 0.7792|77.743 ± 0.5218|78.997 ± 0.2956|
> |||BT|75.907 ± 0.2786|77.809 ± 0.4624|78.855 ± 0.2814|
> |||REX|75.853 ± 0.8047|77.658 ± 0.8853|78.662 ± 0.2828|
> |||**UBA**| **76.289 ± 0.4031**|**78.258 ± 0.1640**|**79.388 ± 0.0170**|
>
> **On LLM training**: Given the high cost of training large language models (LLMs), running multiple experimental trials is often impractical and rarely adopted in practice. **Our study encompasses 40-50 experimental runs, cumulatively consuming 30,000 A100 GPU-hours, which is a substantial investment for a research paper** (for comparison, DeepSeek-V3 utilized 3M H800 GPU-hours). This scale validates robustness and scalability.
>
> More critically, experiment reliability can be assessed through the data-to-parameters ratio (D/N ratio). As Chinchilla scaling laws show, a D/N ratio of 20 is already sufficient for stable convergence.
> **Our LLM experiments far exceed this threshold** (e.g., 50B/36M ≈ 1389, 50B/73M ≈ 685, 75B/151M ≈ 497, 150B/300M ≈ 500), indicating that each parameter receives ample training data. This high D/N regime guarantees: (a) stable convergence, not just early convergence, and  (b) representative single-run results with minimal variance.
> This approach follows modern LLM training standards. Seminal works like LLaMA-2 and OLMo similarly omit multi-trial reporting when D/N ratios exceed 200, as their publications show.
>
> *$\mathbf{Weakness\enspace 3}$: While the authors claim UBA generalizes across model scales, the experimental validation spans a relatively narrow range—from ResNet-18 to ResNet-50—limiting the strength of the claim. Broader evaluation on larger or more diverse model families would strengthen this argument.*
>
> *$\mathbf{Question\enspace 3}$: While UBA is claimed to generalize across model scales, have the authors considered evaluating it on a broader set of architectures (e.g., ResNet101, ViT, or Transformer-based models) to better support this claim?*
>
> **Answer:** We appreciate the reviewer's valuable feedback regarding the generalization across model scales.
> We would like to clarify that the current manuscript **already includes comprehensive experiments on Transformer architectures (OLMo family)**. In detail, we have already conducted experiments on VGG16, ResNet-18, ResNet-34, ResNet-50 ( CNN architectures), **OLMo-36M, OLMo-73M, OLMo-150M, OLMo-300M (Transformer architectures with varying token number and depth/width configurations)**,  in the current version of paper.
>
> The OLMo experiments provide particularly compelling evidence because they represent modern Transformer architectures widely used in practice. Moreover, the 8× parameter range (from 36M to 300M) exceeds the scale variation in our vision experiments. Furthermore, Transformers' known sensitivity to optimization makes consistent improvements more meaningful, and the consistent leading results on all OLMo sizes test true generalization.
>
> **Our study encompasses 40-50 experimental runs, cumulatively consuming 30,000 A100 GPU-hours—a substantial investment for a research paper** (for comparison, DeepSeek-V3 utilized 3M H800 GPU-hours). This scale validates robustness and scalability. Moreover, we conduct sufficient overtrain experiments. All models are trained with **D/N ratios >500×**, far above Chinchilla's 20× sufficiency threshold. It ensures stable final performance, not just early convergence.
>
> *Overall, UBA  consistently improves final accuracy across all OLMo models (see Sec. 4.2 and Appendix E.5).  As a result, this systematic evaluation on this modern Transformer architecture provides particularly compelling evidence for UBA's practical utility and practical impact.*
>
> To further strengthen our claims, we will highlight these points in the revised manuscript.

---

### Note · Authors · 2025-08-11

Dear Reviewers and Area Chairs,

We are truly grateful for thorough feedback and valuable guidance. Below we highlight our key contributions and revisions:

**Main Contributions**

We propose a theoretically-grounded LR schedule using a single optimization-encoding parameter, balancing mathematical rigor with implementation simplicity while guaranteeing convergence.
Key advantages include:
- Minimal Overhead: Negligible computational/memory cost.
- Universal Adaptation: Consistent gains across architectures, tasks, and scales.
- Theoretical Guarantees: Proven $\varphi$-optimization condition relationship with convergence guarantees.
- Scalability and Stability: Validated on LLMs with large D/N ratios.
- Notable Gains:  Significant training cost reduction in some scenarios.

**Reviewers positively acknowledge these contributions as：**
- "...clear and elegant theoretical analysis...both principled and practical...clean balance between theoretical rigour and ease of application...appreciate the consistency and extensive empirical study..." (jwKN).
- "...well-motivated...rigorous approach...outperforms baseline models under various settings...appreciate the experimental validation and the promising..."(DPSV).
- "...strong results...I like the ideas...good extension...I would generally support acceptance if the authors commit to the improvements ..." (LjTY).
- "...comprehensive theoretical analysis...well motivated...strong theoretical analysis and an intuitive concept...extensive theoretical and experimental justifications and analysis..." (a6Zw).

**Revisions**

Addressing common concern: We conduct additional runs to provide error bars to confirm consistent improvements.

Addressing individual concerns:

- jwKN: We clarify that our experiments include Transformer architectures. Error bars and multi-dimensional analyses confirm robust gains.
- DPSV: We address approximation error concerns via method safeguards and experimental validation. We also include parameter analysis and $\varphi$ -adapted scheduling analysis.
- LjTY: We discuss and credit the citations. We conduct additional experiments of all schedules with different learning rates.
- a6Zw: We clarify that our implementation adheres to PyTorch's official API, with production-ready code. Moreover, it requires negligible computational/memory cost.

We have carefully addressed all key concerns. We extend our heartfelt thanks to the reviewers and area chairs for the feedback and guidance.

Best regards

---

### Decision · Program_Chairs · 2025-09-17

**Decision:**

Accept (poster)

**Comment:**

This work introduces an original and novel approach for selecting and tuning hyperparameters for training neural networks under a constrained budget. The method is elegant, theoretically grounded, and demonstrates strong practical performance - which are important strengths of the paper.

Concerns were raised regarding the scale of the experiments, but in my view, an ImageNet evaluation is sufficient for a novel idea of this kind. More computationally intensive studies can naturally be deferred to followup work. The rebuttal effectively addressed the reviewers’ questions, and all acknowledged the paper’s merits: it is original, well-motivated, and empirically validated that should bring some attention at NeurIPS.